# Determinants of functional synaptic connectivity among amygdala-projecting prefrontal cortical neurons in male mice

Yoav Printz [1], Pritish Patil [1], Mathias Mahn [1], Asaf Benjamin [1], Anna Litvin [1], Rivka Levy[1], Max Bringmann[1] & Ofer Yizhar [1] ✉

The medial prefrontal cortex (mPFC) mediates a variety of complex cognitive functions via its vast and diverse connections with cortical and subcortical structures. Understanding the patterns of synaptic connectivity that comprise the mPFC local network is crucial for deciphering how this circuit processes information and relays it to downstream structures. To elucidate the synaptic organization of the mPFC, we developed a high-throughput optogenetic method for mapping large-scale functional synaptic connectivity in acute brain slices. We show that in male mice, mPFC neurons that project to the basolateral amygdala (BLA) display unique spatial patterns of local-circuit synaptic connectivity, which distinguish them from the general mPFC cell population. When considering synaptic connections between pairs of mPFC neurons, the intrinsic properties of the postsynaptic cell and the anatomical positions of both cells jointly account for ~7.5% of the variation in the probability of connection. Moreover, anatomical distance and laminar position explain most of this fraction in variation. Our findings reveal the factors determining connectivity in the mPFC and delineate the architecture of synaptic connections in the BLA-projecting subnetwork.

The computational power of the neocortex is thought to be derived from the complexity and plasticity of connectivity patterns in cortical neuronal networks. Understanding these connections and their dynamics is therefore crucial to deciphering the principles of neuronal computation. Electrophysiological recordings from pairs or groups of neurons have revealed many of the factors which determine the probabilities and properties of cortical synaptic connections. The majority of this work has focused on primary sensory and motor cortical regions, aiming to delineate the streams of information that support sensory processing and motor control[1–3]. These studies have established that the pattern of synaptic connectivity among pairs of cortical pyramidal neurons is not homogeneous[4] but rather depends, among other factors, on the pre- and postsynaptic cell types[5,6], their intracortical laminar source of input[5,7–9], and the long-range projection target of each of the neurons[10–17]. Moreover, the anatomical axodendritic overlap alone cannot account for the probability of connection[11,18–20], indicating specific selection of synaptic partners. These findings suggest that cortical regions consist of interdigitated functional subnetworks of preferentially interconnected neurons[2]. In the primary visual cortex, pyramidal neurons which respond to similar visual stimuli are more likely to be synaptically connected[18,21]. Remarkably, connections between neurons sharing similar stimulus tuning are also the strongest[22], emphasizing the preferential connectivity between cells that share a common role in the circuit.

Despite this body of knowledge, little is known about the synaptic organization of associative cortical structures such as the medial prefrontal cortex (mPFC)[23]. In line with the complex morphology of its pyramidal cells[24], the mPFC connects with numerous cortical and subcortical regions[25,26] and plays a role in multiple cognitive functions and complex behaviors[23,27–38]. We set out to test whether similar principles of functional-subnetwork organization can be applied to the mPFC. We focused on the population of mPFC cells extending long-

[1]Department of Brain Sciences, Weizmann Institute of Science, Rehovot 76100, Israel. ✉e-mail: ofer.yizhar@weizmann.ac.il

range axonal projections to the basolateral amygdala (mPFC-BLA cells) in order to test our hypothesis that their known involvement in associative fear learning[30,39–43] is associated with a unique connectivity pattern that distinguishes them from other neuron populations in the mPFC, similar to principles found in the primary visual cortex[11,18,21,22]. For this purpose, we developed an optogenetic approach for large-scale, unbiased mapping of functional synaptic connections at the level of specific neuron populations. Our approach is based on co-expression of the channelrhodopsin stCoChR[44] with the calcium indicator GCaMP6s[45] in a targeted cell population. Under this configuration, we conducted whole-cell patch-clamp recordings from single neurons in the expressing region while semi-automatically detecting and stimulating other cells in their vicinity in three dimensions, in order to evaluate the input from each stimulated cell onto the recorded postsynaptic cell. We utilized the overlapping excitation spectra of stCoChR and GCaMP6s to perform simultaneous optogenetic stimulation and calcium recording using a single femtosecond laser source. This allows validation of spiking in stimulated cells, thereby providing information on the spatial selectivity of synaptic connections as well as accurate measures of connection probabilities. With this method, we mapped the functional connections among mPFC-BLA cells and among randomly labeled mPFC cells as reference. Our results reveal the detailed layer and projection target selectivity in the connectivity patterns of mPFC pyramidal cells. We further used the comprehensive connectivity maps to quantify the contribution of various anatomical and physiological features to the probability of connection between mPFC neurons.

## Results

### An optogenetic strategy for simultaneous two-photon stimulation and calcium recording of neurons in three dimensions using a single laser source

To achieve reliable, single-cell-targeted optogenetic stimulation of pyramidal cells in the mPFC, we used the recently published soma-targeted channelrhodopsin variant stCoChR, which allows highly efficient two-photon stimulation[44]. Since both stCoChR and GCaMP6s can be efficiently excited at $\lambda = 940$ nm, their co-expression allows simultaneous photostimulation and fluorescence-based activity readout using one wavelength. To target mPFC-BLA cells, we first injected a Cre-expressing rAAV2-retro vector[46] into the BLA of the reporter mouse line Ai9 (ref. [47]) and characterized its retrograde coverage in the mPFC (Fig. S1; see also Methods, under *Specificity of mPFC-BLA cell labeling*). We next co-expressed stCoChR and GCaMP6s in mPFC-BLA cells by injecting rAAV2-retro-Cre into the BLA of wildtype mice, and Cre-dependent AAV vectors expressing stCoChR and GCaMP6s into their mPFC (Fig. 1a, b and Fig. S2a, b). In these two experiments we observed that in the dorsal mPFC (prelimbic and cingulate cortices), BLA-projecting cells were largely absent from layer 3, whereas in the ventral infralimbic and dorsal peduncular cortices, BLA-projecting cells distributed densely across all layers (Fig. 1b, Fig. S1a, and Fig. S6).

In order to validate spiking in response to two-photon stimulation, we performed cell-attached recordings from mPFC-BLA cells expressing stCoChR and GCaMP6s in acute slices while scanning spiral patterns over the soma[48] (Fig. 1c). Two-photon spiral patterns were scanned at 10 Hz (spiral duration: 7.1 ms), and GCaMP6s fluorescence was recorded only during scan periods (Fig. 1d). We found that spiral patterns scanned at 10 Hz evoked reliable spiking as well as an increase in GCaMP6s fluorescence over the spiral train (Fig. 1d–f). Application of TTX (1 μM) blocked spiking and abolished the increase in GCaMP6s fluorescence (Fig. 1d–h), indicating that GCaMP6s fluorescence can be used as a proxy for spiking activity in these conditions. Furthermore, reducing the duration of each spiral from 7.1 ms to 3.6 ms such that some spirals fail to evoke a spike, we found that GCaMP6s fluorescence increased only following successful spiral stimulations (Fig. S3a). Since the rise and decay kinetics of GCaMP6s fluorescence are slower than

the duration of a spiral and the inter-spiral interval, respectively (Fig. 1i and ref. [45]), the relative GCaMP6s fluorescence during each spiral in a train reports spiking in response to previous spirals in the train. We therefore used the raw GCaMP6s fluorescence slope to determine whether spiking occurred during a spiral train (Fig. 1g, bottom; Fig. 1h, right; and Fig. S3b; see Methods).

We next surveyed the space of two-photon scan parameters in order to optimize the GCaMP6s-based spike readout and the time precision of spiking. We performed cell-attached recordings from mPFC-BLA cells expressing stCoChR and GCaMP6s and measured spiking and relative GCaMP6s fluorescence while scanning trains of 10 spirals over each cell (Fig. 1i and Fig. S3c). We modified the diameter of the spirals, their duration (by concatenating multiple spirals and keeping the dwell time constant), their frequency in the train, and the light power on the cell. GCaMP6s $\Delta F/F_0$ was higher for smaller spirals (10 and 15 μm; Fig. 1i), mainly due to increased contribution from off-cell noise in larger spirals (20 μm; see Methods). Spike latency and jitter were lower for larger spirals (Fig. S3c), leading us to proceed with 15 μm spirals for our experiments. We chose a spiral duration of 7.2 ms (a two-spiral sequence), stimulation frequency of 10 Hz, and light power of 10 mW on cell, since these parameters provided high GCaMP6s $\Delta F/F_0$ and high spike probability while maintaining low spike latency, jitter and number of spikes per spiral (Fig. 1i and Fig. S3c). Under these conditions, the spatial specificity for evoking spikes, measured as the full width at half maximum (FWHM), was $55.9 \pm 16.4$ μm in the axial (z) axis and $24.2 \pm 7.2$ μm in the radial (xy) plane (Fig. S4a, b). Based on this spatial specificity and on the density of expressing cells (Fig. S4c–e), we estimate the potential bias in measurement of probability of synaptic connection to be 11.5% (see Methods). The spatial specificity curve for GCaMP6s $\Delta F/F_0$ tended to be narrower than that of spiking (Fig. S4a, b; FWHM for $\Delta F/F_0$: $23.9 \pm 4.1$ μm in z and $11.3 \pm 1.8$ μm in xy; see Fig. S4f–h for the microscope's point spread function), potentially as a result of the high light-sensitivity of stCoChR compared with the light power needed for imaging GCaMP6s.

We next applied our method for combined stimulation and imaging to a population of cells in three dimensions (Fig. 1j). For this, we used an algorithm for automated detection of fluorescently labeled cell bodies in image-stack volumes[49]. Detection was based on mScarlet[50] and nuclear dTomato[51] co-expressed with stCoChR and GCaMP6s, respectively (since these two markers could not be distinguished, co-expression of both stCoChR and GCaMP6s was validated later in analysis based on GCaMP6s fluorescence in response to stimulation; see below and Methods). Running the algorithm on $n = 5$ scanned volumes from two mice (volume size ~$420 \times 420 \times 300$ μm³; $n = 209.2 \pm 8.9$ detected cells per volume) and observing the detections, we found that $2.5 \pm 0.4\%$ of detections were false and $5.4 \pm 0.5\%$ were double (i.e., coordinates point to the center of two adjacent cells; Fig. S2d). We tested the algorithm on additional $n = 4$ scanned volumes from four mice, where we manually registered the coordinates of all the cells in the volume, independent of and blind to the automated detection ($n = 316.5 \pm 45.8$ manually registered cells per volume). The automated detection was biased to cells with higher fluorescence intensities compared with our manual cell registration (Fig. S2e). In contrast, the distribution of cell depths was similar for automatically and manually detected cells (Fig. S2f). Finally, we transformed the coordinates of the automatically detected cells into standardized brain-reference anatomical positions using anatomical landmarks (Fig. 1j; see Methods).

### Analysis of functional connectivity

In order to measure functional synaptic connectivity among mPFC-BLA cells, we co-expressed stCoChR and GCaMP6s in mPFC-BLA cells (Fig. 1a). In each experiment, we obtained a whole-cell recording from one mPFC-BLA cell in the acute slice. We then acquired a series of Z-sections covering the entire depth of the slice ($\lambda = 1040$ nm),

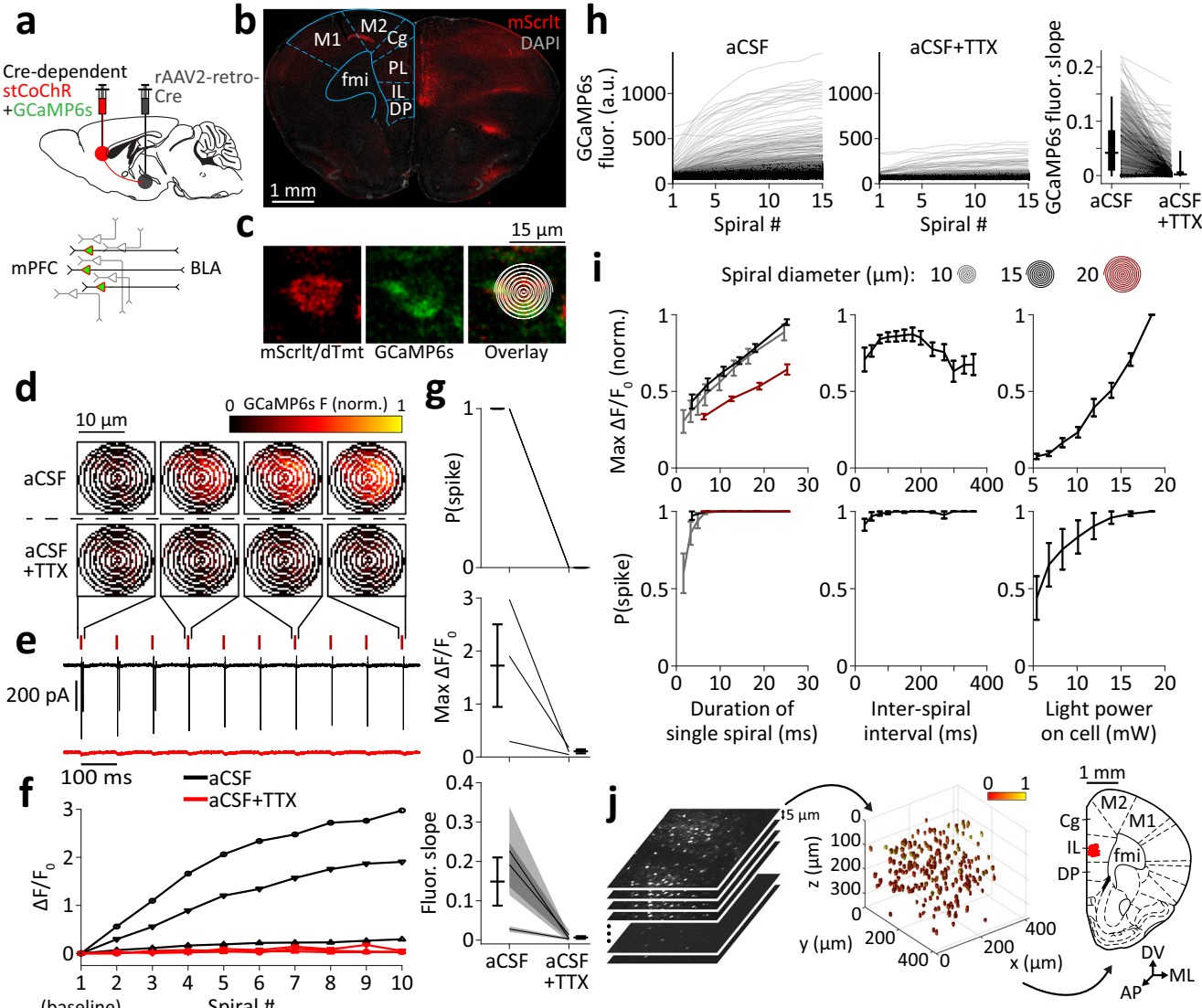

**Fig. 1 | Simultaneous optogenetic stimulation and GCaMP6s-based spike readout using a single laser source. a** Top, Intersectional viral strategy for expressing stCoChR (co-expressed with mScarlet) and GCaMP6s (co-expressed with nuclear dTomato) in mPFC cells projecting to the BLA (mPFC-BLA cells). Sagittal atlas illustration was adapted from ref. [107]. Bottom, Schematic of expression of stCoChR and GCaMP6s exclusively in mPFC-BLA cells. **b** Representative confocal image of a coronal section showing expression of stCoChR and GCaMP6s in mPFC-BLA cells. Anatomical region outline is adapted from ref. [54]. **c** Representative two-photon images of a cell targeted for spiral scanning, with the spiral-scan pattern overlaid (right). **d, e** Cell-attached recordings from an mPFC-BLA cell expressing stCoChR and GCaMP6s. Trains of 10 spiral patterns (7.1 ms each) were scanned at 10 Hz over the soma as in **c** to excite stCoChR and GCaMP6s, and GCaMP6s fluorescence was recorded during the scan of each spiral. **d** Raw GCaMP6s fluorescence (normalized to the maximum) during four selected spirals in absence of TTX (top, aCSF) and after application of TTX to block spiking (bottom, aCSF+TTX). **e** Cell-attached recording traces from the same cell shown in **d**, in absence (black) and presence (red) of TTX. Red ticks denote spiral-scan times. **f, g** Cell-attached recordings from three mPFC-BLA cells during spiral scanning as in **d, e. f** GCaMP6s ΔF/F$_0$ without TTX (black) and with TTX (red) over 10 spirals for each cell. Plotted symbols indicate individual cells. **g** Probability for at least one spike per spiral (top), maximal GCaMP6s ΔF/F$_0$ (middle), and slope of the linear fit of the raw GCaMP6s fluorescence trace (bottom), with and without TTX. Here and elsewhere, error bars indicate mean ± SEM, unless indicated otherwise. Lines represent individual cells, and shaded regions in the fluorescence slope (bottom) indicate the 95% confidence interval of the slope for each cell. **h** Raw GCaMP6s fluorescence of automatically

detected mPFC-BLA cells across 15 spirals scanned at 10 Hz, in absence (left) and presence (middle) of TTX (see **j** for the soma-detection process). Right, Slope of GCaMP6s fluorescence traces for the same cells. Vertical lines inside boxes indicate median, boxes indicate 25th and 75th percentiles, and whiskers represent the 5th and 95th percentiles. n = 645 cells from six slices in two mice. See also Fig. S3b. **i** Effects of scan parameters on maximal GCaMP6s ΔF/F$_0$ (top; normalized per cell to the maximal value across conditions) and on probability for at least one spike per spiral (bottom). Cell-attached recordings were acquired from mPFC-BLA cells expressing stCoChR and GCaMP6s, while the cells were scanned with trains of 10 spirals. The following scan parameters were varied: diameter and duration of each spiral (left; n = 11 cells; light power on cell = 13.9 mW, inter-spiral interval = 100 ms), time interval between consecutive spirals (middle; n = 11 cells; light power on cell = 13.9 mW, spiral duration = 3.6 ms), and light power on cell (right; n = 10 cells; spiral duration = 3.6 ms, inter-spiral interval = 100 ms). See Fig. S3c for additional measurements. **j** Process of targeting a group of cells for photostimulation and imaging. First, a two-photon image stack is acquired (left). An algorithm then automatically detects the cell bodies within the acquired image volume (middle; color scale indicates relative fluorescence intensity) and targets the detected cells for consecutive individual stimulation and imaging using spiral scans. Finally, the positions of the cells are transformed into brain-reference anatomical coordinates (right; the same cells are shown in red overlaid on a reference coronal atlas image adapted from ref. [54]). Cortical region abbreviations: Cg cingulate, PL prelimbic, IL infra-limbic, DP dorsal peduncular, M1 primary motor, M2 secondary motor. fmi forceps minor of the corpus callosum. DV dorsoventral axis, ML mediolateral axis, AP anteroposterior axis. Source data are provided as a Source Data file.

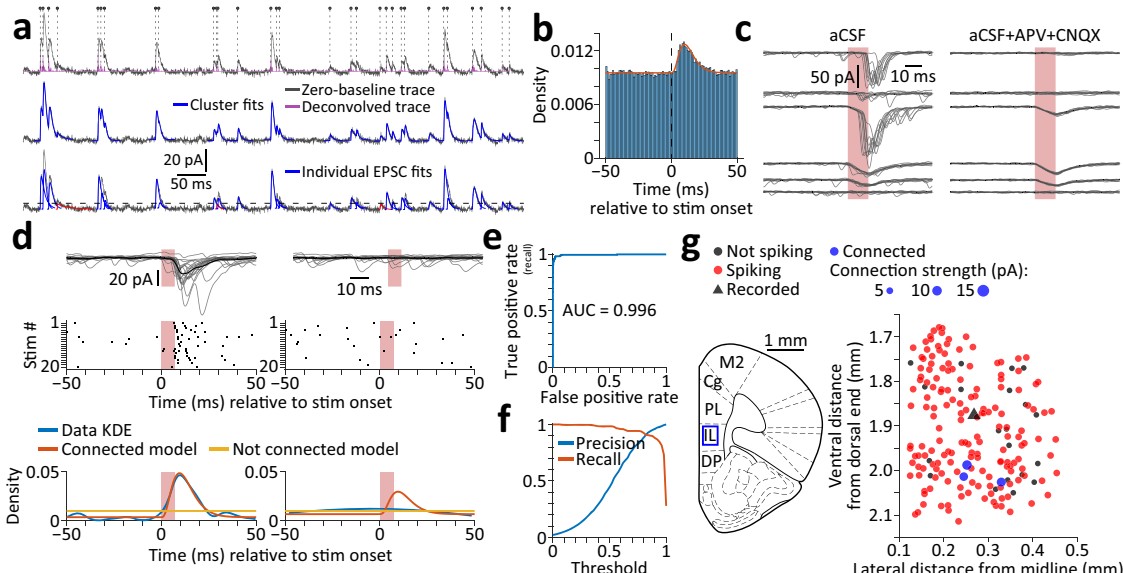

**Fig. 2 | Analysis and modeling of synaptic connections. a**, Detection and modeling of excitatory postsynaptic currents (EPSCs) in continuous voltage-clamp recording traces obtained during stimulation of candidate presynaptic cells. Top, The zero-baseline version of the inverted recording trace (gray) was deconvolved with a synaptic-event waveform kernel using OASIS (purple trace). Putative EPSCs were detected based on peaks in the deconvolved trace. Dots and vertical dashed lines mark the fitted onset times of detected events (see Bottom). Middle, Putative events were then clustered by temporal proximity, and each cluster was fit with a model which assumes linear summation of EPSCs (blue traces). Bottom, Clustered events were decomposed and their kinetic features were extracted. EPSCs with amplitude below a 2.5 $\widehat{SD}$ threshold (dashed line) were discarded (red traces). **b** Peristimulus time histogram of EPSCs ($n = 151797$ EPSCs recorded during stimulation of $n = 10445$ cells, bin size = 1 ms), overlaid with a fitted mixture model of gamma and uniform distributions. Parameter fits [95% confidence interval]: Gamma shape $k = 4.1896$ [3.7063, 4.6729]; gamma scale $\theta = 3.0942$ [2.6602, 3.5283]; weight $w = 0.0455$ [0.0421, 0.0489]. Only stimulated cells that did not evoke any direct photocurrent in the recorded cell were considered for this distribution. **c** Example traces obtained during stimulation of six mPFC-BLA cells while recording from another mPFC-BLA cell (left), and during repeated stimulation in the presence of

glutamate-receptor blockers (right). The top three cells were synaptically connected to the recorded cell, and the bottom three were not. Note the compound photocurrent+EPSC response in the third cell (see Methods, under *Subtracting evoked photocurrents*). **d** Identification of synaptic connections based on EPSC distribution. Top, Overlaid traces recorded during stimulation of a synaptically connected cell (left) and a non-connected cell (right). Red shaded area denotes stimulation period, shaded lines are individual traces ($n = 20$ per cell), and dark lines indicate mean of all traces. Middle, Raster plots of the EPSCs. Bottom, Estimated probability density functions of the EPSC times, using either a model that assumes synaptic connection or a model that assumes no connection. The kernel density estimation (KDE) of the true EPSC times is presented as reference. **e, f** Connectivity model performance, based on $n = 10445$ cell pairs (see Methods), of which 243 are synaptically connected. **e** Receiver operating characteristic curve. **f** Precision and recall as a function of the model score threshold. **g** A representative synaptic connectivity map describing the functional inputs onto a recorded mPFC-BLA cell from neighboring mPFC-BLA cells. Blue box on Atlas image (left, adapted from ref. [54]) marks the anatomical location of the map. Source data are provided as a Source Data file.

detected the labeled mPFC-BLA cells within the scanned volume (Fig. 1j), and stimulated them sequentially as described above while recording both their GCaMP6s fluorescence and the synaptic currents in the recorded postsynaptic cell. Cells whose GCaMP6s fluorescence indicated they did not spike in response to stimulation (see Methods) were excluded from analysis.

In order to determine which of the stimulated cells is connected to the recorded cell, we first used a template deconvolution-based method[52] with a synaptic event waveform kernel[53] to identify all excitatory postsynaptic currents (EPSCs) recorded during the period of sequential stimulation (see Methods). We clustered the detected EPSCs (both spontaneous and evoked) based on temporal proximity and fit EPSC clusters with a sum of functions describing the waveform of a synaptic current, such that compound events could be decomposed to determine the kinetics of each of the underlying events (Fig. 2a).

We next constructed a model to determine whether each stimulated cell is connected to the corresponding recorded cell. We used the rate and the stimulation-aligned time distribution of all EPSCs (Fig. 2b and Table S5) to predict the EPSC distribution around the stimulation of each cell. We then fitted two models for each stimulated cell, one model that assumes no synaptic connection (whereby EPSCs distribute uniformly) and another model that assumes connection (whereby the EPSC distribution contains a bump following the stimulation; see Methods). To decide whether the stimulated cell is connected to the

recorded cell, we used information criteria (see Methods) to determine which of the two models fits the EPSC distribution more accurately (Fig. 2d). To validate and quantify the performance of our model, we examined the recording traces of all cells to identify synaptic connections manually. The area under the receiver operating characteristic curve was 0.996 (Fig. 2e, f). To maximize accuracy, we relied both on the manual observation and on the model to identify synaptic connections, such that candidate connections which could not be resolved by manual observation ($n = 44$ stimulated cells) were settled by the model. Finally, we calculated the strength of synaptic connection at each stimulation as the weighted average of EPSCs within a time window following stimulation (see Methods), thus accommodating for jitter in synaptic latency and the possibility for multiple evoked EPSCs (Fig. S3c).

Overall, we generated 92 functional input maps by recording from 92 mPFC cells and stimulating a total of 10817 cells (Tables S1 and S2). Out of the stimulated cells in each input map, $78.6 \pm 1.6\%$ responded with spiking based on GCaMP6s data, giving a total of 8780 responsive cells ($95.4 \pm 5.1$ responsive cells out of $117.6 \pm 5.0$ stimulated cells per map). Of the maps, 75 were in the ventral mPFC (69 maps in the infralimbic cortex and 6 maps in the dorsal peduncular cortex) and 17 were in the dorsal mPFC (14 maps in the prelimbic cortex and 3 maps in the cingulate cortex). An example of one functional input map of an mPFC-BLA cell in the ventral mPFC is shown in Fig. 2g. The anatomical

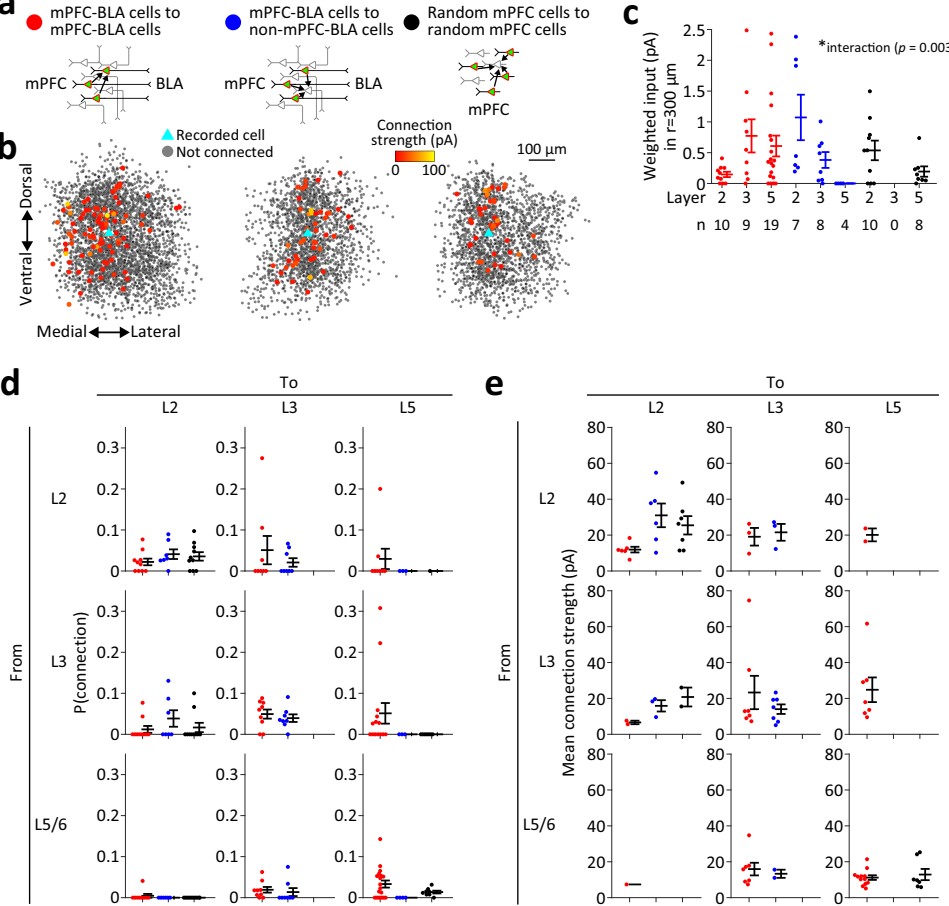

**Fig. 3 | Functional synaptic connectivity in cell populations of the ventral mPFC. a** Description of the three types of synaptic connections measured. **b** Overlay of all connectivity maps obtained for each connection type in **a**. Stimulated cells are in shaded black, and connected (presynaptic) cells are color coded for connection strength. All cell locations are relative to their corresponding recorded (postsynaptic) cell position (within the same map), which is presented as a cyan triangle at the center of the overlaid maps. Positions along the depth of the slice (anteroposterior axis) are collapsed. **c** Weighted synaptic input according to the layer of the postsynaptic cell, for each connection type. Zero weighted input indicates that none of the stimulated cells were connected to the recorded cell. * Two-way ANOVA interaction effect between layer and connection type, $F_{(4)} = 5.2$, $p$

$= 0.003$. $n$ recorded cells for each condition is indicated under the plot. In **c**–**e**, error bars indicate mean ± SEM. **d** Probabilities of cross-layer synaptic connections. Notice that stimulated (presynaptic) cells in layers 5 and 6 were pooled. $n$ cells for mPFC-BLA to mPFC-BLA cell connections: 10 L2→L2, 10 L3→L2, 9 L5/6→L2, 8 L2→L3, 9 L3→L3, 9 L5/6→L3, 8 L2→L5, 14 L3→L5, 19 L5/6→L5. $n$ cells for mPFC-BLA to non-mPFC-BLA cell connections: 7 L2→L2, 7 L3→L2, 6 L5/6→L2, 8 L2→L3, 8 L3→L3, 8 L5/6→L3, 3 L2→L5, 3 L3→L5, 4 L5/6→L5. $n$ cells for random mPFC to random mPFC cell connections: 10 L2→L2, 10 L3→L2, 10 L5/6→L2, 0 L2→L3, 0 L3→L3, 0 L5/6→L3, 1 L2→L5, 7 L3→L5, 8 L5/6→L5. **e** Strengths of cross-layer synaptic connections, presented as mean amplitude of all connections found in the corresponding layers. $n$ cells are as in **d**. Source data are provided as a Source Data file.

position of each recorded and stimulated cell was inferred using anatomical landmarks aligned with projections on reference atlas images[54] (see Methods).

### Functional synaptic connectivity in mPFC cell populations

We used our technique to analyze the properties of functional synaptic connections among mPFC-BLA cells in the ventral mPFC (Fig. 3a, left). As reference, we measured synaptic output from mPFC-BLA cells onto non-labeled cells, which likely do not extend projections to the BLA (given efficient retrograde labeling by rAAV2-retro[46] and assuming that <10% of mPFC neurons project to the BLA[55]), by recording from non-labeled mPFC pyramidal cells while exciting mPFC-BLA cells in their environment (mPFC-BLA cells to non-mPFC-BLA cells; Fig. 3a, middle). To measure the general, non-specific connectivity in the ventral mPFC, we sparsely expressed stCoChR and GCaMP6s in the mPFC (see Methods) and recorded from a randomly selected pyramidal cell while scanning over labeled cells in its environment (random mPFC cells to random mPFC cells; Fig. 3a, right). Fig. 3b shows the overlay of all maps obtained for each of the three types of synaptic connection, aligned to the position of the corresponding recorded cell. Among the 75 maps in

the ventral mPFC, six maps (8%) were in the dorsal peduncular cortex (DP). Since the DP contained cells projecting to the BLA with a laminar distribution similar to that of the infralimbic cortex (Fig. 1b, Fig. S1a, and Fig. S6; see also ref. [56]), and since the DP-BLA cells have been implicated in fear extinction, similar to infralimbic neurons[39] – we have included the DP maps in our analysis of ventral mPFC connectivity.

We first asked whether the three types of connection differ in their probabilities and strengths. To avoid a bias between maps in the distances between presynaptic cells and the recorded cell, we restricted the measurement of overall probability for synaptic connection to a 300 μm distance from the recorded cell. Overall connection probability and connection strength (which was not restricted in distance) did not differ between the three map types (mPFC-BLA cells to mPFC-BLA cells, mPFC-BLA cells to non-mPFC-BLA cells, and random mPFC cells; Fig. S5a–c, top and middle). To quantify the compound input that each recorded cell receives, we treated non-connected cells as having zero amplitude and calculated the average connection strength from all cells in the map (weighted input). Weighted input (restricted to 300 μm distance) did not differ between the three map types as well (Fig. S5a–c, bottom). Moreover, we did not find a correlation between the

connection probability and mean connection strength across recorded cells (Fig. S5d). We next divided the connectivity maps to layers, based on the laminar position of the recorded cell (Fig. S6). We found that the weighted output from mPFC-BLA cells onto other mPFC-BLA cells was stronger in deeper layers than in superficial ones, whereas their output onto non-mPFC-BLA cells had an opposite laminar pattern (Fig. 3c–e and Fig. S5e). To further examine the directionality of excitatory input in each network, we measured the weighted input that each cell receives from a given anatomical direction. We found that mPFC-BLA cells tend to receive stronger input from other mPFC-BLA cells located medially to them (Fig. S5f), yielding a superficial-to-deep flow of excitatory input. In contrast, we did not find such a pattern in output from mPFC-BLA cells onto non-mPFC-BLA cells or among random mPFC cells. Consistent with this finding, random mPFC cells in L5 did not receive input from L2 and L3 cells, and non-mPFC-BLA cells in L5 did not receive input from mPFC-BLA cells in L2 and L3 (Fig. 3d). While this stands in contrast to canonical circuits in sensory cortex, where L5/6 samples from L2/3 (ref. [6,9]), it could be explained by the small sample of L2/3-to-L5 connectivity maps, since we did find connections from L2 and L3 to L5 among mPFC-BLA cells, which are included in the random mPFC cell category. Weighted input along the dorsoventral axis showed no particular directionality (Fig. S5g). All connection types showed similar dependency of connectivity on distance (Fig. S5b, c). Finally, synapses of all types and in all layers displayed similar short-term depression upon a presynaptic stimulation train (Fig. S7; see Discussion).

### Determinants of functional synaptic connectivity in the mPFC

We next asked which parameters (of the ones recorded in our experiments) contribute to the variation in probability of connection between pairs of cells in the entire mPFC. For this purpose, in addition to the information about the anatomical locations of stimulated and recorded cells, we analyzed the intrinsic electrophysiological properties of each recorded cell, for the three classes of mPFC neuron (Fig. 4a). We found that in the ventral mPFC, mPFC-BLA cells had higher output gain compared with non-mPFC-BLA cells (Fig. 4b, d). mPFC-BLA cells showed stronger firing-rate adaptation compared with non-mPFC-BLA and with random mPFC cells (Fig. 4c, e). Other electrophysiological properties did not differ between cell classes (Fig. S8a and Table S3; see Fig. S8b for pairwise correlations between electrophysiological properties and connectivity features). Counterintuitively, cells receiving strong weighted input tended to have lower input resistance (Fig. 4f), perhaps as a homeostatic mechanism to restrain excitation. When examining the relationship between anatomical position and intrinsic properties, we found that sag ratio and spike threshold were both strongly correlated with cells' mediolateral (laminar) position, for all cell classes (Fig. S8c). Spike half-width was also correlated with mediolateral position, but not in the mPFC-BLA cell class (Fig. S8c).

To identify the factors that play a role in determining synaptic connectivity, we treated the data as pairs of stimulated (candidate presynaptic) and recorded (postsynaptic) cells, and included cell pairs in both the ventral and the dorsal mPFC ($n = 7752$ cell pairs; see Methods). We used a logistic regression model to predict whether each pair is connected (Fig. 4g). As features, we used the electrophysiological properties of the recorded cell, the anatomical positions of both cells, their distance, the connection type (as in Fig. 3a), and the anteroposterior position of the recorded slice ($n = 20$ features overall). In order to remove multicollinearity between features (Fig. S8b), we performed zero-phase components analysis (ZCA) whitening (see Methods). To find the features which non-redundantly affect connectivity, we imposed sparsity on the regression model by using Horseshoe priors[57,58] (Fig. 4h and Fig. S9). We found that among all features, the distance between cells (Euclidean and lateral), the anatomical position of the presynaptic cell (mediolateral and

dorsoventral), the bursting behavior of the postsynaptic cell, and the connection type had coefficients that center away from zero, suggesting that they contribute most to the variation in connectivity. The postsynaptic cell's input resistance and the anteroposterior position of the slice also contributed to connectivity. In order to quantify the contribution of all features to variation in connectivity, we used the unregularized regression model (Fig. 4g) and calculated the cross-entropy between predicted and true connectivity as a measure for the information that the features in the model have on connection probability (Fig. 4i; see Methods). We found that using the selected features mentioned above ($n = 8$ features) to predict connectivity reduced the cross-entropy by 7.5% compared with using shuffled connections (from 0.1161 to 0.1074). Using all features ($n = 20$) resulted in worse performance compared with using only the selected features, as demonstrated by higher cross-entropy and larger difference between the cross-validated and non-cross-validated models (Fig. 4i). This finding supports the central role of these selected anatomical and physiological features in explaining variation in connectivity. When removing individual features from the model that uses selected features, we found that the largest effect on cross-entropy arises when removing Euclidean distance and the mediolateral position of the presynaptic cell (Fig. 4i). These results suggest that anatomical position, especially along the mediolateral axis, and intersomatic distance dominate connectivity, while all features in our dataset can jointly account for ~7.5% of overall variation in connection probability.

## Discussion

Current knowledge about functional synaptic connectivity in cortical networks is derived largely from experiments using paired-patch recordings, where two or more cells are simultaneously recorded in the whole-cell patch-clamp configuration, and spikes are triggered in one cell while the others are recorded for synaptic responses[6,9,21,23,59]. These recordings are typically limited to a confined intersomatic distance and to cells located within ~60 μm of the surface of the slice. While this method provides high-precision information about the functional properties of the probed synapses, it suffers from very low yield, making it extremely difficult to detect sparse connections and to avoid bias in distances or positions of probed connections. In contrast, several recent studies have implemented recording from one cell while optogenetically stimulating a selection of cells in its vicinity. This approach allows unbiased mapping of synaptic connections in the network, thereby facilitating detection of sparse connections, but is restricted to measuring only unidirectional (in-degree) connectivity from the stimulated cells onto the recorded cell[60–64]. Here we presented a large-scale implementation of such an approach, combining it with calcium-based readout of activity using a single laser source. We utilized the overlapping excitation spectra of stCoChR and GCaMP6s to perform simultaneous stimulation and calcium recording. Using GCaMP6s recording to monitor the activity of each stimulated cell allowed us to accurately reconstruct the spatial architecture of synaptic connections by assigning an anatomical position to each connected and non-connected cell. Our semi-automated approach for cell detection and for sequential stimulation and calcium recording allowed us to probe the input from 95.4 ± 5.1 cells, whose spiking in response to stimulation was validated using the GCaMP6s signal, onto each recorded cell in three dimensions within a volume of ~420×420×300 μm³. The use of a single light path makes this approach accessible and widely applicable using almost any commercially available two-photon microscope system. Future experiments could utilize scanless holographic illumination and temporal focusing techniques[60,65] in order to reduce the stimulation-induced spike jitter and thereby improve detection of connections.

Among all the available anatomical and cellular features in our data, several seem to dominate the probability of synaptic connection between pyramidal cells in the mPFC (Fig. 4g–i and Fig. S9). The most

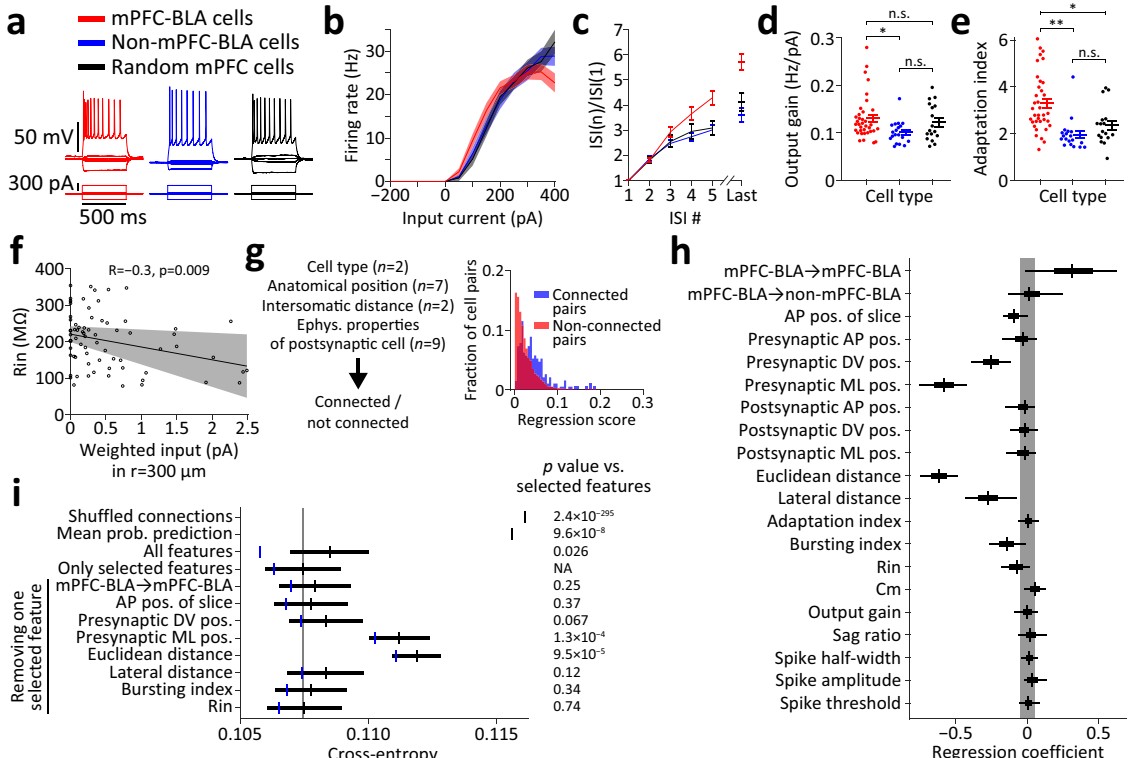

**Fig. 4 | Determinants of functional synaptic connectivity among mPFC pyr-amidal neurons. a–e** Electrophysiological properties of cell populations in the ventral mPFC ($n = 36$ mPFC-BLA cells, $n = 19$ non-mPFC-BLA cells, and $n = 17$ random cells). **a** Representative current injection traces recorded from the three cell types corresponding to the connectivity maps in Fig. 3. **b** Firing rate as function of injected current for all cell types. Shaded areas represent SEM. **c** Ratio between the n-th and the first inter-spike intervals (ISIs). ISIs are taken from the current injection traces evoking closest to 80% of the maximal firing rate per cell. In **c–e**, error bars indicate mean ± SEM. **d** Output gain for each cell type. Kruskal-Wallis test with post hoc comparisons, *$p = 0.015$; n.s. $p ≥ 0.26$. **e** Adaptation index for each cell type. Kruskal-Wallis test with post hoc comparisons, **$p = 6.0×10^{-6}$; *$p = 0.016$; n.s. $p = 0.24$. See Fig. S8a for additional electrophysiological properties and Table S3 for full statistics. **f** Correlation between weighted input and input resistance across all cell types. The line indicates a linear fit and the shaded region represents its 95% confidence interval. **g** Logistic regression model for predicting connectivity using all available features ($n = 20$ total features; $n$ for each category is mentioned in parentheses) on all cell pairs in the entire mPFC ($n = 7752$ cell pairs). Right, Model score calculated for connected cell pairs and for non-connected cell pairs. **h** Feature selection performed using Horseshoe priors on the regression model, using the same cell pairs as in **g**. For each coefficient, the median (vertical line), 25th and 75th percentiles (box), and 90% highest density interval (HDI, whiskers) of the

posterior distribution are shown. Shaded area spans ±0.05 to represent change of 5% in odds (see Methods). See Fig. S9 for the full posterior distributions. **i** Cross-entropy between predicted connection probability (calculated using logistic regression as in **g**) and true connections, using different sets of features based on their selection in **h**. Cross-entropy of shuffled connections is calculated between true connections and predictions run on shuffled connections ($n = 10,000$ shuffles). Mean probability prediction presents the cross-entropy between the true con-nectivity and a constant connection probability equal to its mean over all data. The bottom rows present cross-entropies calculated when using only selected features, while the denoted one is removed. Vertical lines present the mean and error bars present the SEM of cross-validated distributions of cross-entropies. Vertical blue lines present the cross-entropy calculated without cross-validation. Shaded vertical line indicates the mean of the cross-validated distribution of the model using selected features as reference. Note that the SEM of shuffled connections is too small to be discernible. P values on the right are based on comparison with the model that uses the selected features (see Methods), using a paired-sample Stu-dent's t-test (since the same cross-validations were used for all models), except for the shuffled connections (where a two-sample Student's t-test was used since the model using shuffled connections did not use cross validation) and the mean connection prediction (where a one-sample t-test was used). Source data are pro-vided as a Source Data file.

prominent features are intersomatic distance and the laminar depth of the presynaptic cell, consistent with previous studies of cortical synaptic connectivity[6,9]. Connection probability tends to decrease with intersomatic distance, and cell pairs whose presynaptic cell is in dee-per layers also tend to have a lower probability of connection (Fig. 4h, i). Notably, the laminar depth of the recorded (postsynaptic) cell is not correlated with its input properties (connection probability and amplitude, $p ≥ 0.12$). Other features contributing to the reduction in connection probability include the spike-bursting behavior of the postsynaptic cell (and, to a lesser extent, its input resistance), the dorsoventral anatomical position of the presynaptic cell (whereby ventral position is associated with reduction in connection prob-ability), and the anteroposterior position of both cells (whereby anterior position is weakly associated with reduction in connection probability). A connection type of mPFC-BLA cell to mPFC-BLA cell is positively associated with connection probability (Fig. 4h and Fig. S9).

Notably, the negative correlation of input resistance with connectivity (Fig. 4f) indicates that it is not an artefact of the detectability of synaptic connections (as higher resistance would facilitate detection of weaker EPSCs). Our regression analysis and cross-entropy calcula-tions (Fig. 4i) suggest that all features taken together can account for ~7.5% of variation in connectivity. This finding suggests that the probability of pyramidal cells in the mPFC to form a synaptic con-nection is determined mostly by features that were not available in our experimental paradigm. These could be the morphological features of the pre- and postsynaptic cells, the electrophysiological properties of the presynaptic cell (which are not accessible using optogenetic con-nectivity mapping)[6,12,14,23], the activity patterns of the cells[18,21], their gene-expression profile[66,67], cell lineage[67–70] (and lineage of projection-target cells[71]), age and experience of the animal[72–74], or other factors.

A previous transsynaptic tracing study has revealed that only ~20% of the synaptic inputs onto mPFC L5 cells are from local mPFC cells,

whereas in L5 cells of the barrel cortex, input from local cells amounts to ~80% of the total synaptic input[75]. This property could translate to sparser local connectivity in the mPFC compared with other cortical regions. Accordingly, the overall synaptic connection rates in our dataset (Fig. S5a–c, top) were low compared with connection rates found in the sensory cortex[6,9]. This sparse connectivity is consistent with recent findings regarding connections among pyramidal neurons in the mPFC found using paired intracellular recording[76]. However, an earlier study found similar connectivity rates in the mPFC and the visual cortex of the ferret[23]. These differences could be attributed to experimental factors of our system, such as the temporal jitter in presynaptic spiking and in EPSC latencies, and the limitation in the number of stimulations per presynaptic cell, set by the limited recording duration and the large overall number of putative connections to be probed. We also experimentally estimated that the process of slicing the brain scales down connection rates by a factor of 43.6% (see Fig. S10 and Methods), informing future studies relying on in vitro recording.

Synapses among pyramidal neurons in the ferret mPFC have been shown to be diverse in their kinetic properties, with some showing strong short-term facilitation[23] compared with the largely depressing synapses among sensory pyramidal neurons in the mouse[77] (although this may differ in vivo[78,79]). Although our connectivity-mapping protocol does not allow precise control over the number of presynaptic spikes per stimulation (Fig. S3c), we used it to estimate the short-term plasticity in probed synapses. We found no differences in short-term plasticity between cell types and layers (Fig. S7), suggesting either that this specific type of facilitating synaptic connection does not exist in the mouse mPFC, or that it was not represented in our dataset. Alternatively, the relatively high extracellular calcium concentration we used (2 mM)[80], although standard for slice electrophysiology, could have masked facilitation[81–83].

Pyramidal cells in the mPFC are highly heterogeneous in their long-range input and output profiles[25,26]. Given their common projections to the BLA and their shared involvement in associative fear learning[30,39–43], we hypothesized that mPFC-BLA cells would show unique connectivity patterns within the mPFC local circuit. Our data indicated that, at the population level, mPFC-BLA cells seem to be similar in their connectivity pattern to the general mPFC pyramidal cell population (Fig. S5a–c). However, close examination of the spatial distribution of connections revealed that mPFC-BLA cells are selective in their synaptic output along the laminar axis (Fig. 3c–e). The mPFC and the BLA have been shown to share preferentially strong reciprocal synaptic connections[55,84,85]. Axonal projections from the BLA onto the mPFC, which carry information about learned associations[86], densely innervate L2 (and also L5)[84,87], where they directly excite L2/3 pyramidal cells[55] and interneurons[84]. Our findings suggest a stream of information, whereby input from the BLA strongly excites back-projecting mPFC-BLA cells in superficial layers[55], from which information diverges to two main routes. One route is directed laterally onto other mPFC-BLA cells in deeper layers (Fig. 3c and Fig. S5f) to form a recurrent excitatory loop with the BLA. The other route spreads locally within the superficial layers onto other mPFC pyramidal cell populations (Fig. 3c). One possible target population for this route is nucleus accumbens (NAc)-projecting neurons, which are abundant in superficial mPFC layers[25,84] (as well as pyramidal neurons dually projecting to both the BLA and the NAc[84]). Therefore, this local processing of information from the BLA in the mPFC could form a basis for the processing of sensory inputs associated with negative and positive valence, to guide action selection in the face of conflicting cues[35,86].

## Methods

### AAV expression plasmids

A Cre-dependent stCoChR expression plasmid, labeled with mScarlet (pAAV-EF1α-DIO-CoChR-Kv2.1-P2A-mScarlet), was generated as described in ref. [44]. Cre-dependent GCaMP6s plasmid labeled with nuclear dTomato (pAAV-EF1α-DIO-GCaMP6s-P2A-NLS-dTomato) was acquired from Addgene (plasmid #51082). Cre expression plasmids, used for local sparse expression of stCoChR and GCaMP6s in the mPFC, were either acquired from Addgene (pAAV-EF1α-NLS-Cre-P2A, plasmid #55636) or cloned based on pAAV-EF1α-NLS-Cre-P2A using standard restriction cloning (pAAV-CaMKIIα-NLS-Cre-P2A).

### Production of recombinant AAV vectors

HEK293 cells were seeded at 25–35% confluence. The cells were transfected 24 h later with plasmids encoding AAV rep, cap of AAV1 and AAV2, and a vector plasmid for the rAAV cassette expressing the relevant DNA using the PEI method[88]. Cells and medium were harvested 72 h after transfection, pelleted by centrifugation ($300 \times g$), resuspended in lysis solution ([mM]: 150 NaCl, 50 Tris-HCl; pH 8.5 with NaOH), and lysed by three freeze–thaw cycles. The crude lysate was treated with 250 U benzonase (Sigma) per 1 ml of lysate at 37 °C for 1.5 h to degrade genomic and unpackaged AAV DNA before centrifugation at $3000 \times g$ for 15 min to pellet cell debris. The virus particles in the supernatant (crude virus) were purified using heparin-agarose columns, eluted with soluble heparin, washed with phosphate-buffered saline (PBS) and concentrated by Amicon columns. Viral suspension was aliquoted and stored at −80 °C. Viral titers were measured using real-time PCR. Retrograde AAV vectors (rAAV2-retro) were kindly provided by the Janelia Viral Tools facility (rAAV2-retro-hSyn-Cre) or generated as described in ref. [46] (rAAV2-retro-EF1α-Cre).

### Animals

All experimental procedures were approved by the Institutional Animal Care and Use Committee (IACUC) at the Weizmann Institute of Science. C57BL/6J male mice aged four weeks postnatal were obtained from Envigo and used for AAV vector injections and for recordings. Ai9 mice[47] (Cre-dependent tdTomato reporter line, used for calibrating retrograde expression from the BLA) were obtained from Jackson Laboratory and bred in-house. Up to five mice were housed in a cage in a 12 h light–dark cycle with food and water ad libitum at 22 °C and 48% humidity. Following viral injection surgery, mice were housed for at least four weeks before being recorded to allow for recovery and virus expression.

### Stereotactic injection of AAV vectors

Four- to six-week-old mice (29–46 days postnatal) were initially induced with ketamine (80 mg/kg) and xylazine (10 mg/kg) by intra-peritoneal injection and then placed into a stereotaxic frame (David Kopf Instruments) and put under isoflurane anesthesia (~0.9% in O₂, v/v). A craniotomy (~1 mm diameter) was made above each injection site. A Nanofil syringe (World Precision Instruments) with a 34G beveled needle was filled with virus suspension (or mixture of viruses, according to injection site). The needle was inserted into the injection site, bevel facing anteriorly, and left in place for 5 min, followed by slow injection of the virus mixture (10–100 nl/min). After injection, the needle was left in place for additional 10 min and then slowly withdrawn. The surgical incision was closed with tissue adhesive (3M), and buprenorphine (0.05 mg/kg) was subcutaneously injected for post-surgical analgesia. Mice were monitored daily for the first week after surgery and twice weekly afterward. Injections coordinates, in mm relative to bregma (injected volume): mPFC: 1.95 anterior, 0.3 lateral, 2.85 ventral (400–500 nl); BLA: 1.15 posterior, 3.0 lateral, 5.0 ventral (100–350 nl). AAV vectors used for intracranial injections had genomic titers ranging $5.3 \times 10^{11}$–$3.1 \times 10^{13}$ genome copies per milliliter (gc/ml, before dilution; see below). When AAV vectors were injected together for co-expression, their titers were matched by up to one order of magnitude. To achieve local sparse expression of stCoChR and GCaMP6s, Cre-dependent vectors were injected together with a titer-matched Cre-expressing vector that was first diluted in PBS by a factor of 1:100 before being mixed with the Cre-dependent vectors and

injected. This dilution factor was based on calibration injections performed using dilutions of 1:20 to 1:1000 in order to achieve cell density comparable to that of mPFC-BLA cells.

## Calibration of retrograde expression in mPFC-BLA cells

Ai9 mice were injected with varying volumes of rAAV2-retro-Cre into the BLA (100 nl at injection rate of 10 nl/min, 200 nl at 20 nl/min, or 300 at 30 nl/min). The left BLA of each mouse was injected with rAAV2-retro-EF1α-Cre at a titer of $4.1 \times 10^{12}$ gc/ml, and the right BLA was injected with the same volume of rAAV2-retro-hSyn-Cre at a titer of $2.5 \times 10^{13}$ gc/ml. At least four weeks after injection, mice were deeply anesthetized with an intraperitoneal injection of pentobarbital (400 mg/kg) and perfused transcardially with ice-cold PBS followed by a solution of 4% paraformaldehyde (PFA) in PBS (pH 7.4). Brains were removed and incubated overnight in 4% PFA at 4 °C, and then transferred to 30% sucrose in PBS for at least 24 h at 4 °C until cryosectioning. Coronal sections (40 μm thickness) were cut on a microtome (Leica Microsystems) and collected in cryoprotectant solution (25% glycerol and 30% ethylene glycol in PBS, pH 6.7). Sections were washed in PBS, stained for 3 min with DAPI (5 mg/ml solution diluted 1:30,000 prior to staining), washed again with PBS, mounted on gelatin-coated slides, dehydrated, and embedded in DABCO mounting medium (Sigma). Images of sections from each mouse, located approximately at the same anteroposterior position, were acquired using a slide-scanning microscope (VS120, Olympus, using VS-ASW 2.9 software), with acquisition settings being kept constant across all sections. Regions of interest (ROIs) for quantification of cell number and fluorescence intensity were selected based on DAPI fluorescence and on atlas reference images[54] to cover the BLA and the ventral mPFC. Cell bodies in the ventral mPFC ROIs were counted manually from z-stack images that were reacquired under a confocal microscope (Zeiss LSM 700, using ZEN 14.0 software). Cell bodies in the BLA could not be clearly discerned due to local expression of the rAAV2-retro-Cre vector and therefore were not counted. Fluorescence intensity was calculated as the mean for the entire ROI from the z-stack images acquired using the slide scanning microscope.

## Immunohistochemistry

Since both stCoChR and GCaMP6s were co-expressed with red fluorophores (cytosolic mScarlet and nuclear dTomato, respectively), we performed anti-GFP staining on fixed prefrontal slices in order to visualize GCaMP6s independent of stCoChR and quantify their colocalization. Slices were washed three times in PBS for 5 min each, then blocked for 1.5 h at room temperature (RT) in blocking solution (20% normal horse serum (NHS) and 0.5% triton in PBS). Slices were then incubated with a polyclonal rabbit anti-GFP primary antibody (ThermoFisher Scientific, catalog # A-11122) for 24 h at RT (1:300 anti-GFP antibody, 2% NHS and 0.5% triton in PBS). Slices were subsequently washed three times in PBS for 5 min each and incubated with a polyclonal goat anti-rabbit secondary antibody conjugated to Alexa Fluor 488 (Abcam, catalog # ab150077) for 2 h at RT (1:300 anti-rabbit antibody and 2% NHS in PBS). Finally, slices were washed in PBS three times, stained with DAPI for 5 min (5 mg/ml solution diluted 1:10,000 prior to staining), washed, mounted, and imaged on a confocal microscope as described above.

## Point spread function (PSF) measurement

Fluorescent microspheres (Bruker) were suspended in ethanol and thinly spread on a coverslip. After allowing the ethanol to evaporate, microspheres were imaged under 940 nm at 10 mW with pixel size of 0.1 μm in z (binned to 0.2 μm) and 0.03 μm in xy (binned to 0.12 μm).

## Acute brain slice preparation

Mice were injected intraperitoneally with pentobarbital (130 mg/kg) and perfused transcardially with carbogenated (95% $O_2$, 5% $CO_2$) ice-

cold slicing solution ([mM] 2.5 KCl, 11 glucose, 234 sucrose, 26 $NaHCO_3$, 1.25 $NaH_2PO_4$, 10 $MgSO_4$, 0.5 $CaCl_2$; 340 mOsm/kg). After decapitation, 300 μm-thick coronal mPFC slices were prepared in carbogenated ice-cold slicing solution using a vibratome (Leica VT 1200 S) and allowed to recover for 20 min at 33 °C in carbogenated high-osmolarity artificial cerebrospinal fluid (high-osmolarity aCSF; [mM] 3.21 KCl, 11.8 glucose, 131.6 NaCl, 27.8 $NaHCO_3$, 1.34 $NaH_2PO_4$, 1.07 $MgCl_2$, 2.14 $CaCl_2$; 320 mOsm/kg) followed by 25 min incubation at 33 °C in carbogenated iso-osmotic aCSF ([mM] 3 KCl, 11 glucose, 123 NaCl, 26 $NaHCO_3$, 1.25 $NaH_2PO_4$, 1 $MgCl_2$, 2 $CaCl_2$; 300 mOsm/kg). Subsequently, slices were kept at room temperature in carbogenated aCSF until use.

## Electrophysiological recording in acute brain slices

Whole-cell patch-clamp recordings were obtained under visual control using oblique illumination on a two-photon laser-scanning microscope (Ultima IV, Bruker) equipped with a femtosecond pulsed laser (Chameleon Vision II, 80 MHz repetition rate; Coherent), a 12 bit monochrome CCD camera (QImaging QIClick-R-F-M-12) and a 20×, 1.0 NA objective (Olympus XLUMPlanFL N). Borosilicate glass pipettes (Sutter Instrument BF100-58-10) with resistances ranging 3–6 MΩ were pulled using a laser micropipette puller (Sutter Instrument Model P-2000). The recording chamber was perfused with carbogenated aCSF at 2 ml/min and maintained at -26–32 °C. Pipettes were filled with K-based low-Cl solution ([mM] 130 K-gluconate, 5 KCl, 10 HEPES, 10 $Na_2$-phosphocreatine, 4 ATP-Mg, 0.3 GTP-Na; 285 mOsm/kg; pH adjusted to 7.25 with KOH) for most of the connectivity-mapping experiments, and with Cs-based intracellular solution for six of the experiments ([mM] 120 Cs-gluconate, 11 CsCl, 1 $MgCl_2$, 1 $CaCl_2$, 10 HEPES, 11 EGTA, 5 QX-314; 280 mOsm/kg; pH adjusted to 7.3 with CsOH). In cell-attached recordings, used for correlating GCaMP6s fluorescence with electrophysiological recording of the same cell, pipettes were filled with 150 mM NaCl. Alexa Fluor 350 dye (<1 mM; Thermo Fisher Scientific) was added to the intracellular solutions, as well as Neurobiotin Tracer (0.3 mg/ml; Vector Laboratories) in some of the experiments. Recordings were performed using a MultiClamp 700B amplifier, filtered online at 8–10 kHz, digitized at 20–50 kHz using a Digidata 1440A digitizer, and acquired using pClamp 10 software (Molecular Devices).

## Full-field illumination and light power calibration

Full-field illumination was performed using a 470 nm light-emitting diode (29 nm bandwidth LED; M470L2-C2; Thorlabs) delivered through the microscope illumination path including a custom dichroic in order to reflect the 470 nm activation wavelength. Light power densities were calculated by measuring the light transmitted through the objective using a power meter (Thorlabs PM100A with S142C sensor) and dividing by the illumination area, which was directly measured by placing an autofluorescent micrometer in the image plane and illuminating with the LED to measure the fluorescent area observed through the microscope eyepiece.

## Sequential two-photon spiral scanning of candidate presynaptic cells

A region in an acute mPFC slice containing cells expressing stCoChR and GCaMP6s was selected using brief wide-field green illuminations so as to minimize activation of stCoChR in the slice. A cell (either expressing stCoChR and GCaMP6s or non-expressing) was patch-clamped and sections spanning the entire depth of the slice were scanned (5 μm interval, 1040 nm, -9–55 mW under objective) using Prairie View software (Bruker) to obtain a volume of ~420 × 420 × 300 μm³ containing the recorded cell. Cell bodies were detected in the volume using a custom script written in Matlab (MathWorks)[49]. Coordinates for two-photon spiral scanning of each of the cells (15 μm diameter, 1 μm revolution distance, scanning inward and outward for 7.16 ms) were

generated using a custom-written Matlab script. An additional Matlab script was generated for sequential scanning of all the cells while adjusting focus and light power between cells. Detected cells were sequentially scanned at 940 nm using Prairie View while the patched cell was continuously recorded in voltage-clamp mode at −70 mV. The GCaMP6s fluorescence signal was recorded through a GaAsP PMT (Bruker) during scan periods. Each cell was scanned with 10 or 15 spirals delivered at 10 Hz. Light power on each cell was 10 mW (after adjustment for attenuation through the tissue; see below). If the recorded cell remained viable after all putative presynaptic cells were stimulated, up to three repetitions were executed with the same set of scan patterns. At the beginning and at the end of each protocol repetition, a series of 5 mV square hyperpolarizing voltage pulses were delivered in order to monitor the recorded cell's input resistance, membrane capacitance, and access resistance. At the beginning of each recording (prior to the presynaptic cell scanning protocol), full-field light pulses were delivered to verify stCoChR expression, or to look for active synaptic contacts across the entire field of illumination in case the recorded cell was non-expressing.

### Detection and modeling of synaptic events

The process of detecting and modeling EPSCs in a voltage-clamp recording trace was composed of six steps.

(1) Detrending. To standardize event detection, we inverted the recording trace (multiplied by −1) and then downsampled it to 5000 Hz using polyphase resampling. To compensate for drifts in the holding current and other slow fluctuations in the signal, we estimated a baseline for the recording trace by applying a 10th percentile filter of 50 ms width, followed by smoothing, which was done by downsampling to 200 Hz, median filtering with a 15 ms window, and upsampling back to 5000 Hz. The baseline was then subtracted from the recording trace to yield a "zero-baseline" trace $S$ (shifted to have a zero median) which was used for all subsequent analyses. The noise in the trace was characterized by an estimated standard deviation $\widehat{SD}$ of $1.4826 \times$ MAD, where MAD is the median absolute deviation around the median.

(2) Deconvolution. To detect the EPSCs in the zero-baseline trace, we deconvolved it with a kernel similar to the waveform of a typical EPSC:[53,89]

$$k(t) = \exp\left(-t/\tau_{decay}\right) \times \left(1 - \exp(-t/\tau_{rise})\right) \quad (1)$$

We used the OASIS implementation of this deconvolution[52] originally intended for fast and accurate detection of spikes from calcium signals. We used a kernel with time constants $\tau_{decay} = 3.5$ ms and $\tau_{rise} = 0.7$ ms and ran the OASIS algorithm for up to 10 iterations with a sparsity-imposing L1 penalty and noise level of $0.8\ \widehat{SD}$. For subsequent analyses we used two outputs from the OASIS algorithm: the deconvolved trace and the denoised trace (convolution of the deconvolved trace with the kernel).

(3) Detection. Detection of EPSCs was performed by finding peaks in the deconvolved trace after filtering it with a triangular kernel of 2.5 ms width. We only used peaks with height and prominence of at least $0.5\ \widehat{SD}$ and a minimum distance of 2.0 ms between them. The location of the peak was used as an estimate of onset time ($\hat{o}$), and the area under the curve of the deconvolved trace between −0.4 ms and +0.8 ms from $\hat{o}$ was used to estimate the height ($\hat{h}$) of the event.

(4) Clustering. Detected EPSCs were clustered such that two consecutive events were in the same cluster if the second event started before the first had decayed back to baseline. To achieve this, we took all segments of the denoised trace where it was higher than $1.8\ \widehat{SD}$. We extended each segment by 10 ms before its beginning and 20 ms after its end, and further extended it by 10 ms around events. We merged overlapping segments, such that all events within the same segment were included in the same cluster.

(5) Fitting. To accurately determine the time and the shape of synaptic events, we used two steps of curve fitting to refine our parameter estimates iteratively. For this, the zero-baseline trace for each cluster of events was fit to the sum of synaptic kernels with separate parameters:

$$\widetilde{s} = c + \sum_{i=1}^{n} k\left(t; h, o, \tau_{decay}, \tau_{rise}\right) \quad (2)$$

where $k(t; h, o, \tau_{decay}, \tau_{rise})$ is a synaptic kernel of height $h$ at onset time $o$ with shape $k(t) = \exp(-t/\tau_{decay}) \times (1 - \exp(-t/\tau_{rise}))$ as Eq. (1), $c$ is the constant offset for each cluster, and $n$ is the number of events in the cluster (for simplicity, we assumed that in the soma, where the recordings are made, synaptic events are summed linearly).
We minimized

$$\frac{RMS(S - \widetilde{s})}{\widehat{SD}} + \frac{RMS(o - \hat{o})}{5} \quad (3)$$

where $RMS$ is the root mean square error, $S$ is the zero-baseline trace, and onset times of all the events are in milliseconds. The height and the two time constants (rise and decay) were optimized in the logarithmic scale to ensure the same relative accuracy across event heights and timescales. The search space of the parameters height $[\hat{h}/15, 3\hat{h}]$ and onset time $[\hat{o} - 10, \hat{o} + 10]$ was set relative to the estimate for each event, whereas the search space for the time constants ($0.5 < \tau_{decay} < 50$, $0.1 < \tau_{rise} < 10$, in ms, with a constraint $\tau_{decay} > \tau_{rise}$) was identical across all the events. The first round of minimization was done using Dual Annealing[90], a global minimization method based on simulated annealing in SciPy[91]. The number of iterations and function evaluations were increased with the number of events in the cluster. The second round of minimization was done using the SciPy implementation of Powell's method, a derivative-free local optimization method, with the starting point from the optimum found by dual annealing.

(6) Thresholding. After the fitting, only events with heights of more than $2.5\ \widehat{SD}$ were kept. A $2.5\ \widehat{SD}$ value corresponded to $4.55 \pm 0.06$ pA.

See Table S4 for all parameter used in the procedure.

### Modeling the distribution of EPSCs and assessing functional connectivity

The recorded (postsynaptic) cell receives spontaneous EPSCs from cells in the network that are independent of the stimulation of the targeted (candidate presynaptic) cells. Moreover, evoked EPSCs can have large jitter resulting from the jitter in stimulation-evoked presynaptic spikes (Fig. S3c) and in synaptic latency. These two factors served as motivation for developing a statistical model to determine which stimulated cell is connected to the recorded cell.

(1) Bayesian "rate-and-time" models. To this end, we combined two types of information using Bayesian models whose posteriors were sampled using a Markov chain Monte Carlo (MCMC) method. First, if a stimulated cell is connected to the recorded cell, a bump in the rate of EPSCs is expected between -5–25 ms from stimulation onset (see Fig. 2b–d). Second, if a cell is connected, evoked events appear in addition to spontaneous events, such that a higher rate of events is expected during the evoked time period (90 ms after the stimulation, disregarding the last 10 ms before the next stimulation) compared with reference 90 ms intervals where no stimulation occurred (spontaneous intervals). Model 1 assumes that the stimulated cell is not connected to the recorded cell, and thus expects the distribution of events in the

evoked time periods to be uniform and the same rate to explain the number of events in both spontaneous and evoked periods. Model 2 assumes that the cell is connected, and thus expects the distribution of events in the evoked time periods to have a bump following the stimulation time on top of a uniform distribution. These two models (1 and 2) are referred to as "rate-and-time" models. We sampled from the posteriors of these two models using the No-U-Turn Sampler (NUTS)[92] in PyMC3[93]. Then, using the Pareto-smoothed importance sampling leave-one-out (PSIS-LOO)[94] information criterion, we chose the hypothesis (connected or not connected) with a better fit for each cell. The priors for these models were based on either empirical data or biologically plausible values (Table S5).

(2) Data. The rate-and-time models fit the following data for each stimulated cell for each protocol repetition:
- Spont_time = duration of the given spontaneous segment (≤90 ms); these segments were within 5 seconds of the time of stimulation of the stimulated cell.
- Evoked_time = duration of the given evoked segment (=90 ms).
- Num_events_spont = total number of events in each spontaneous segment.
- Num_events_evoked = total number of events in each evoked segment.
- Event_times = exact times of all the events which occur in the evoked time segment (relative to stimulation onset).

(3) Parameters. We fit the aforementioned data to infer two parameters for each stimulated cell:
- Spont_rate = spontaneous event rate (inferred separately for each protocol repetition).
- Evoked_per_trial = number of events evoked per trial (shared across repetitions and inferred only if the model assumes the cell is connected).

(4) Mathematical formulation.
Model 1 (assuming no synaptic connection):
- Priors:
$$Spont\_rate \sim Gamma(rate\_mu, rate\_sigma)$$
- Distributions:
$$bump = Gamma(\mu = bump\_center, \sigma = bump\_width)$$
$$unif = Unif(lower = 0, upper = Evoked\_time)$$
- Likelihood:
$$Num\_events\_spont \sim Poisson(Spont\_rate \times Spont\_time)$$
$$Num\_events\_evoked \sim Poisson(Spont\_rate \times Evoked\_time)$$
$$Event\_times \sim Mixture([bump, unif], weights = [0, 1])$$
Model 2 (assuming synaptic connection):
- Priors:
$$Spont\_rate \sim Gamma(rate\_mu, rate\_sigma)$$
$$Evoked\_per\_trial \sim Gamma$$
$$(Evoked\_per\_trial\_mu, Evoked\_per\_trial\_sigma)$$
- Distributions:
$$bump = Gamma(\mu = bump\_center, \sigma = bump\_width)$$
$$unif = Unif(lower = 0, upper = Evoked\_time)$$
- Likelihood:
$$Num\_events\_spont \sim Poisson(Spont\_rate \times Spont\_time)$$
$$Num\_events\_evoked \sim Poisson(Spont\_rate \times Evoked\_time + Evoked\_per\_trial)$$
$$w = Evoked\_per\_trial / (Evoked\_per\_trial + Spont\_rate \times Evoked\_time)$$
$$Event\_times \sim Mixture([bump, unif], weights = [w, 1 - w])$$
These two models were fit in PyMC3 using NUTS.

(5) Model comparison. Since the two rate-and-time models use the same priors and fit the same likelihoods, we compared them using expected log pointwise predictive density (ELPD) using Pareto-smoothed importance sampling leave-one-out cross-validation (PSIS-LOO-CV)[94]. Specifically, the two models were combined into a single model using the Bayesian bootstrap-pseudo-Bayesian model averaging (BB-pseudo-BMA) method[95] applied to the PSIS-

LOO calculated using the "compare" function of the ArviZ package[96]. This procedure gives weights to the two models (connected and not connected) which sum up to 1. These weights can be interpreted as the probability of the particular model to be correct, assuming that one of the models is indeed correct. We used the weight of the rate-and-time connected model (model 2) as the output of the procedure ($w_{rt}$).

(6) Bayesian "rate-only" and "time-only" models. The rate-and-time models described above combine the information from both the rates and the times of EPSCs. However, for some cell pairs, the time information may indicate a bump implying a connection, while the rates may suggest no extra evoked events over the expected spontaneous rate (or the opposite: the rate information may imply a connection while the time information does not). Such borderline cell pairs may be misclassified by these models. To resolve this, we further fit two pairs of reduced versions of the rate-and-time models.

Models 3 and 4: A pair of "time-only" models where we only used the Mixture likelihood of the event times. As with the rate-and-time models, one of the time-only models assumes no synaptic connection (model 3), and the other assumes a connection (model 4). For this pair of time-only models we used priors with distribution $Beta(\alpha = 2, \beta = 2)$ on the weight of the bump directly. Note that the model which assumes no connection (model 3) has no priors.

Models 5 and 6: A pair of "rate-only" models where we only used the Poisson likelihoods of the rates. As before, one model assumes no connection (model 5) and the other assumes a connection (model 6). We used the same priors for spontaneous rate (Spont_rate) and evoked per trial (Evoked_per_trial).

The posteriors of these pairs of models (models 3–6) were similarly sampled using NUTS. ELPD was calculated and the relative weights of the connected and not connected hypotheses were calculated the same way as for models 1 and 2 (rate-and-time). We used the weights of the connected models (models 4 and 6) as the outputs of the procedure ($w_t$ and $w_r$).

(7) Connectivity determination. To determine whether a stimulated cell is connected to the corresponding recorded cell, we relied both on the models and on manual observation. Cells whose weights of the connected models (models 2, 4, and 6) crossed a threshold, namely cells satisfying $(w_{rt} \geq 0.5) \wedge (w_t \geq 0.4) \wedge (w_r \geq 0.4)$, were considered as putatively connected. The use of the "rate-only" ($w_r$) and "time only" ($w_t$) models in the criterion ensured that we only considered cells to be connected if both types of information agreed independently that the cell is likely to be connected, thus helping reduce false classifications. For manual identification of synaptic connections, traces recorded during stimulation of each cell were aligned to stimulation onset and observed in search of EPSCs that appear near the stimulation with high reliability and low jitter. Cases of disagreement between the model and the manual observation (n = 212 stimulated cells) were reexamined manually and settled by manual observation (leaving n = 177 cases of disagreement). Cases where manual observation was uncertain (n = 44 stimulated cells) were settled by the criterion above, namely $(w_{rt} \geq 0.5) \wedge (w_t \geq 0.4) \wedge (w_r \geq 0.4)$. Performance of the model (Fig. 2e, f) was measured after handling all cases of disagreement and uncertainty as described.

(8) Bump estimation. For the connected cells, we used another Bayesian model (model 7) to estimate the location and width of the bump in EPSCs. Since the bump represents the distribution of evoked EPSCs, we used this model in order to calculate the strength of synaptic connection (see below). This model was similar to the rate-and-time model assuming synaptic connection (model 2), except for the use of a normal distribution for the bump (which also makes calculations numerically more stable)

and that the center and the width of this normal distribution have prior distributions for which we infer the posteriors (for determining connectivity, these were fixed numbers and thus it was always the same bump).

Model 7 (assuming synaptic connection with variable bump):

- Priors:

$bump\_center \sim Gamma(\mu = bump\_center\_mu, \sigma = bump\_center\_sigma)$

$bump\_width \sim BoundedGamma(\mu = bump\_center\_mu, \sigma = bump\_center\_sigma, lower\_bound = 3.0, upper\_bound = 8.0)$

$Spont\_rate \sim Gamma(rate\_mu, rate\_sigma)$

$Evoked\_per\_trial \sim Gamma(Evoked\_per\_trial\_mu, Evoked\_per\_trial\_sigma)$

- Distributions:

$bump = Normal(bump\_center, bump\_width)$

$unif = Unif(lower = 0, upper = Evoked\_time)$

- Likelihood:

$Num\_events\_spont \sim Poisson(Spont\_rate \times Spont\_time)$

$Num\_events\_evoked \sim Poisson(Spont\_rate \times Evoked\_time + Evoked\_per\_trial)$

$w = Evoked\_per\_trial / (Evoked\_per\_trial + Spont\_rate \times Evoked\_time)$

$Event\_times \sim Mixture([bump, unif], weights = [w, 1 - w])$

## Measurement of synaptic connection strength

The timing of evoked EPSCs could not be accurately predicted due to the jitter in presynaptic spike and in postsynaptic response, such that spontaneous EPSCs occurring adjacent to stimulation could be mistaken for evoked EPSCs. To minimize this potential error, we used the normal distribution which describes the time distribution of evoked EPSCs (the bump) from model 7 described above. The strength of synaptic connection at each stimulation was thus taken as the weighted average of the EPSCs during a 2–30 ms time window following that stimulation, where the weight of each EPSC was determined by the probability density function of the fitted normal.

## Logistic regression model

To understand which features contribute to connectivity, we fit a logistic regression model from the features of the cell pairs to their binary connectivity. For this analysis we used all pairs of stimulated (candidate presynaptic) cells and their corresponding recorded (postsynaptic) cells in the entire mPFC. We removed cell pairs where the stimulated cell was excluded according to the criteria detailed below (removing 2376 pairs), as well as cell pairs where not all of the intrinsic electrophysiological properties of the postsynaptic cell could be measured (removing 689 pairs where a Cs-based internal solution was used for recording). This left $n = 7752$ cell pairs used for the model.

(1) Preprocessing. First, to have all the features on similar scales, we log-transformed features that were only positive or spanned many orders of magnitude (adaptation index, bursting index, membrane capacitance, output gain, input resistance, and spike half-width), and logit-transformed features that took values between 0 and 1 (sag ratio), so that the distribution of each feature was similar to a normal distribution with relatively low skewness and kurtosis. We included the cell type features ("mPFC-BLA to mPFC-BLA connection" and "mPFC-BLA to non-mPFC-BLA connection") as either 0 or 1 (where "random to random connection" was encoded as 0 for both cell type features). In order to compare the relative contribution of the features to connectivity, they must also be normalized, such that the relative magnitudes of the regression coefficients correspond to the information gained by a change in feature value in units of its standard deviation. However, several features in the data have strong correlations with each other, demonstrating multicollinearity (see Fig. S8b for correlations between electrophysiological properties). This makes the interpretation of the coefficients difficult, as they no longer correspond to the amount of information gained by a change of value in each feature. To resolve the multicollinearity, we applied a robust zero-phase components analysis (ZCA) transform to both whiten and normalize the data, so that all the pairwise correlations are removed, and the variance along each transformed feature is 1. We used the SciPy[91] implementation of the minimum covariance determinant estimator[97] (MinCovDet) to estimate the correlation matrix, which we eigen-decomposed to calculate the ZCA transform.

(2) Sparsity-inducing Horseshoe priors. We next performed feature selection to reduce noise and to capture the predictive features in the data. For this, we used a Bayesian generalized linear model whose posteriors of the model were sampled using a MCMC method. We applied Horseshoe priors[57] using the formulation in ref. [58] and based on its PyMC3[93] implementation in https://mellorjc.github.io/HorseshoePriorswithpymc3.html

- Priors:

$v = 3$

$r_{local} \sim Normal(\mu = 0, \sigma = 1)[\text{for each feature}]$

$r_{global} \sim Normal(\mu = 0, \sigma = 1)$

$\rho_{local} \sim InverseGamma(\alpha = 0.5 \times v, \beta = 0.5 \times v)[\text{for each feature}]$

$\rho_{global} \sim InverseGamma(\alpha = 0.5, \beta = 0.5)$

$z \sim Normal(\mu = 0, \sigma = 1)[\text{for each feature}]$

$\beta_0 \sim Normal(\mu = -3.67, \sigma = 1)$

$\tau = r_{global} \times \sqrt{\rho_{global}}$

$\lambda = r_{local} \times \sqrt{\rho_{local}}$

$\beta = z \times \lambda \times \tau$

$\mu_{logit} = X \cdot \beta + \beta_0$

where $X$ is the ZCA-transformed features (model input data).

- Likelihood:

$observed\_data \sim Bernoulli\left(p = InverseLogit\left(\mu_{logit}\right)\right)$

We sampled the posteriors from this model using NUTS[92] in PyMC3. We selected features whose median posterior coefficient was larger than 0.05 in absolute value as features that provide significant information about connectivity (Fig. 4h). The reason for this choice is that a ±0.05 coefficient value corresponds to a change of 0.05 in log odds (which in turn corresponds to a change of approximately 5% in odds) when the value of the feature moves by 1 standard deviation from its mean.

(3) Unregularized, cross-validated logistic regression and cross-entropy calculations. To quantify the information that the features have on connectivity, we used an unregularized, cross-validated logistic regression model implemented in Matlab using the glmfit function. We used the preprocessed features and $n = 191$ stratified cross-validations (equal to the number of connected cell pairs in the dataset to have one connected pair per fold). We calculated the cross-entropy between the model prediction $\hat{y}$ (continuous probability) and the true connections $y$ (binary) as

$$H = (1/n) \times (-y \cdot \log(\hat{y}) - (1 - y) \cdot \log(1 - \hat{y})) \quad (4)$$

where $n$ is the number of cell pairs in the test set (or the total number of cell pairs in the dataset in case of no cross-validation). We ran this model and calculation using all features, using only the selected features, and using the selected features minus one of them, and for each run we calculated the distribution of cross-entropies across the cross-validations as well as the cross-entropy when running the model without cross-validation (Fig. 4i). The same stratified folds were used for all cross-validations so that models using different features could be compared using a paired test (see Statistics below). As control, we ran the model using shuffled connections (without cross-validation) and calculated

the cross entropy between the prediction using shuffled connections and the true connections across 10,000 shuffles. As another control, we calculated the cross-entropy between the true connections and a constant connection probability of 0.025 (the mean connection probability across all cell pairs).

## Subtracting evoked photocurrents

While recording from an stCoChR-expressing cell, stimulation of some targeted cells can evoke unwanted photocurrents in the recorded cell. This is a result of imperfect restriction of the channelrhodopsin molecules to the soma, and can obscure EPSCs. To resolve this, we identified cells whose stimulation evoked direct photocurrent by manually observing the recording traces after aligning them to the stimulation pulse. Photocurrents were identified by their minimal latency and jitter, their reliability, and their reproducible waveform across consecutive stimulations of the same cell (Fig. 2c). Cells whose stimulation evoked photocurrent stronger than 20 pA at the soma of the recorded cell were excluded from analysis. Traces corresponding to cells whose stimulation evoked ≤ 20 pA photocurrent were manually examined for synaptic connections by searching the stimulation-aligned traces for reliable, low-jitter EPSC occurrences. Cases where stimulation evoked a compound response consisting of both a photocurrent and an EPSC were identified by the rapid rise constant of the EPSC as compared with the photocurrent (e.g., Fig. 2c, left, third cell). In order to remove the photocurrent so as to accurately measure these EPSCs, we identified stimulation-aligned traces where no EPSCs were evoked (within a time window of 30 ms after stimulation), and subtracted the mean of these photocurrent-containing traces from the rest of the traces. The resulting traces contained only EPSCs with approximately no photocurrent. Synaptic strength for each stimulation was then calculated by subtracting the mean of a baseline window preceding stimulation (−20 to +2 ms relative to pulse onset) from the minimum of a window following stimulation (+2 to +25 ms relative to pulse onset). In order to improve separation of evoked photocurrents from EPSCs, we repeated the stimulation protocol in several of the recorded cells, in the presence of APV (25 μM) and CNQX (10 μM) to block glutamate transmission (Fig. 2c). We then subtracted the mean of the traces obtained in presence of glutamate-receptor blockers from the corresponding traces without blockers.

## Quantification of intrinsic electrophysiological properties

At the beginning of recording from each cell (prior to the synaptic-mapping protocol), a series of square current pulses (~ −200 to +400 pA, interval 25–50 pA) were injected in current-clamp mode (unless the intracellular solution was Cs-based). Input resistance $R_{in}$ was calculated based on the peak hyperpolarization during injection of the smallest negative current. Membrane capacitance $C_m$ was calculated based on $R_{in}$ and on the membrane time constant $\tau_m$, using an exponential fit $[a \times (1 - e^{-t/\tau_m}) + c]$ of 10 to 90% of the peak hyperpolarization during injection of the smallest negative current. Maximal firing rate was taken as the maximal rate among all injected current pulses. Output gain was taken as the slope of the input-output curve in the range between the minimal and maximal firing rates. Bursting and adaptation indices were based on traces with closest to 80% of the maximal firing rate per cell. Bursting index was taken as the ratio between the second and the first ISIs. Adaptation index was taken as the ratio between the last and the second ISIs. Spike half-width, amplitude, and threshold were calculated based on the first spike in the trace with the lowest firing rate (that is, the first evoked spike). Sag ratio was based on the trace with the strongest hyperpolarizing current, and calculated as $(V_{min} - V_{ss})/(V_{min} - V_{bl})$, where $V_{min}$ is the minimal voltage during the first 30% of the hyperpolarizing pulse, $V_{ss}$ is the steady-state voltage during the pulse, and $V_{bl}$ is the baseline just before the pulse. In six of the connectivity-mapping experiments, the internal solution was Cs-based and therefore the intrinsic electrophysiological properties could not be measured.

## Inferring the brain-reference anatomical positions of the probed cells

In order to standardize the coordinates of all cells (pre- and post-synaptic) across all experiments and calculate their positions in the brain, we transformed the cells' coordinates into brain-reference anatomical positions. For this purpose, we defined three orthogonal planes for each slice. The anatomical position of each cell was based on its distance from each of these planes. (1) A slice-surface plane was defined by a collection of points ($n = 17.4 \pm 0.64$) on the surface of the scanned volume. Recorded slices typically had small curvatures on their surface, possibly caused by the harp that is used to hold them in place during recording, creating an angled volume surface relative to the image plane ($\theta = 11.2 \pm 0.7$ °). The points used to define this plane were imaged during the scanning of the volume for connectivity mapping. Distance from this plane defined a cell's depth inside the slice (referred to as AP position). (2) A midline plane was defined by passing through the dorsal end and the ventral end of the midline of the slice (both ends were recorded) and by being orthogonal to the slice-surface plane. Distance from this plane defined the ML position of the cell. (3) A dorsal-end plane was defined by passing through the dorsal end of the slice and by being orthogonal to the other two planes. The distance from this plane defined the DV position of the cell. The DV and ML positions of each cell were then projected on a reference coronal atlas image[54] (https://kimlab.io/brain-map/atlas/) that matched the slice's AP position relative to bregma, which was determined based on anatomical landmarks such as the shape of the corpus callosum. The anatomical region of the cell was determined by its projected coordinates on the corresponding intensity-labeled atlas image[54] (where anatomical regions were distinguished by their pixel intensity), using a custom Matlab script. The layer of the cell was determined based on Fig S6.

## The effect of spatial specificity of stimulation on measured probabilities of synaptic connection

Targeted optogenetic stimulation of one cell may lead to co-stimulation of adjacent cells, depending on the local density of opsin-expressing cells. If two (or more) adjacent cells respond with spiking and with elevated GCaMP6s fluorescence to stimulation of either one of them, and only one (or some) of them is synaptically connected to the recorded cell, it will seem as though both (or all) stimulated cells are connected to the recorded cell, introducing a connectivity overestimation bias. To address this scenario, we manually registered the positions of all labeled cells in four scanned brain-slice volumes. For each cell in each volume, we defined an ellipsoid centered at the center of mass of the cell. The primary axes of the ellipsoid were defined as the FWHM of the spiking probability at the corresponding axes (Fig. S4c–e). We calculated the number of cells that fall within this ellipsoid as a proxy for the probability to stimulate two (or more) cells together. The fraction of cells having at least one neighboring cell within their FWHM ellipsoid was 0.13 ± 0.02 (Fig. S4c, blue). Among the cells having adjacent within-ellipsoid neighbors, the number of neighbors was 1.06 ± 0.02 (Fig. S4d, blue). The fraction of cell pairs, among all possible pairs in a scanned volume, that are within each other's ellipsoid was $4.2 \times 10^{-4} \pm 3.2 \times 10^{-5}$ (Fig. S4e, blue). These data suggest that in ~13% of cells targeted for photostimulation, two cells might be co-stimulated instead of only one. In the case where all opsin-expressing cells in the volume are sequentially stimulated and respond with spiking, the probability of synaptic connection is therefore overestimated, on average, by 13%. This is because 13% of the connected cells are expected to lie adjacent to a cell that is being co-stimulated with them and is also targeted separately for stimulation. However, in our hands, only 58 ± 7% of the total opsin-expressing cells in a scanned volume are targeted for stimulation. When stimulating a subset of the opsin-expressing cells in a tissue volume during a mapping experiment, an additional 13%,

on average, are effectively being stimulated. Since the number of connections found in each experiment is not affected by these unintended stimulations, the actual connection probability may be lower by 11.5% than that calculated in our experiments. We did not correct for this bias. Importantly, it only affects the absolute probability of connections. It does not affect calculations involving connection strength or the identity of the connected and non-connected cells (such as laminar distribution of connections and prediction of connectivity using the regression model in Fig. 4).

### The effect of acute brain slice preparation on connectivity

One of the major drawbacks of our methodology is that the connections are measured in acute brain slices, and not in the live animal. During the slicing of the brain, neuronal projections are inevitably cut. We took several measures to minimize the loss of neuronal connections due to this cutting process: we recorded from coronal slices, which are cut parallel to the axis of the apical dendrites of pyramidal cells in the mPFC; we recorded from cells located relatively deep in the brain slice (depth in slice = 66.9 ± 2.0 μm, range 26.7–125.1 μm); and we stimulated presynaptic cells spanning the entire depth of the slice, thereby mapping inputs from cells that are otherwise inaccessible using the conventional multiple-patch configuration. By estimating the fraction of the anatomical axodendritic overlap volume between the pre- and postsynaptic neurons that is removed after slicing, a recent study suggested that the cutting process scales down connection rates globally, with little bias to specific connection types[98]. Other studies using similar anatomical simulations suggest that within the intersomatic displacement ranges used in our dataset, ≥60% of the synaptic contacts remain intact in 300 μm-thick brain slices[77,99]. Only a few studies have directly examined functional synaptic connectivity in vivo. Connection probabilities from L2/3 pyramidal neurons onto nearby interneurons in the barrel cortex seem similar in vivo and in vitro[100,101], and among pyramidal neurons the connections in vivo seem even sparser than in vitro[79,101]. This may be attributed to increase in synapse density after slicing, as observed in hippocampal slices[102]. To quantify the possible loss of synapses due to severing of projections, we performed the following analysis. For each connectivity map in our dataset, we considered two groups of presynaptic cells: one containing all the cells that are located in the volume between the slice surface and the depth of the recorded (postsynaptic) cell, and another containing all the cells located in the volume between the depth of the recorded cell down to twice its depth. Since these two groups reside in equally sized tissue volumes, and they differ only in their distance from the cut surface, the difference in connectivity features between them represents the downscaling of connections caused by slicing. We found that connection probability deeper in the slice was 1.77 times higher than close to the surface (0.038 vs. 0.021, reduction of 43.6%; Fig. S10). Mean connection strength did not differ between the two volumes (16.6 pA below vs. 16.2 pA above the recorded cell; Fig. S10). This downscaling of connections is expected to be approximately uniform across map types[98], allowing us to compare the connection probabilities and strengths between them. Therefore, the artefacts of the slicing process should not undermine the comparative qualities of our findings.

### Specificity of mPFC-BLA cell labeling

The representative images in Fig. S1a show retrogradely labeled cell bodies, as well as locally transduced cells in the BLA and their anterograde projections, resulting from injections of rAAV2-retro-Cre into the BLA of Ai9 mice. Fluorescent labeling can be seen outside of the BLA, mainly in the striatal region dorsal to the BLA. This viral spread most likely occurred during the withdrawal of the injection needle containing the virus. To determine whether the transduction spread to regions outside of the BLA can lead to retrograde labeling of mPFC cells that do not project to the BLA and therefore could be mistaken as mPFC-BLA cells, we injected wildtype mice with rAAV2-retro-Cre into

the BLA and Cre-dependent stCoChR and GCaMP6s into the mPFC. To visualize GCaMP6s-expressing axonal projections, we stained brain slices against GFP. We found that fluorescent projections are denser inside the BLA than outside of it (Fig. S2a). Furthermore, we surveyed the recently published database for brain-wide BLA connectivity[56] (https://mouseconnectomeproject.github.io/amygdalar/) and found that the infralimbic cortex only sparsely innervates the striatal region dorsal to the BLA, compared with its dense innervations in the BLA. Additionally, the BLA receives synaptic input from and sends projections to surrounding regions, such as the basomedial amygdala (ventromedial to the BLA) and the entorhinal cortex (lateral to the BLA), which could account for the labeling seen in these regions in Fig. S1a. These observations suggest that the retrogradely labeled cells in the mPFC are mostly mPFC-BLA cells, with a small minority of other cells projecting to areas that surround the BLA.

### Estimate of the total number of connected cells from the EPSC distribution

The distribution of stimulation-aligned EPSCs in our data (Fig. 2b) can be used to estimate the expected number of connected presynaptic cells in the entire dataset.

Let $E$ be the total number of synaptic events in Fig. 2b, f be the fraction of events in the evoked bump (and $1–f$ the fraction of events in the uniform part), $N$ be the number of stimulated cells, $c$ be the fraction of connected cells among all stimulated cells, $n$ be the average number of stimulations on each targeted cell (including protocol repetitions), and $e_p$ be the average number of evoked EPSCs per stimulation for a connected cell.

Then the number of evoked events can be expressed as

$$\#evoked\_events = E \times f = (N \times c) \times n \times e_p \qquad (5)$$

The fraction of connected cells can therefore be estimated as

$$\hat{c} = \frac{E \times f}{N \times n \times e_p} \qquad (6)$$

From the data used in Fig. 2b, E = 151797, $N$ = 10445, $n$ = 27.2 ± 0.14, and $f$ = 0.046 (the weight of the Gamma in the mixture model). Assuming $e_p$ = 0.5 (a combination of the number of evoked spikes per spiral from Fig. S3c with synaptic failure), we get

$$\hat{c} = \frac{151797 \times 0.046}{10445 \times 27.2 \times 0.5} = 0.049$$

This value roughly resembles the fraction of connected cells in our dataset (excluding stimulated cells that evoke a direct photocurrent in the recorded cell, and keeping all other cells regardless of their GCaMP6s signals): $c$ = 243/10445 = 0.023.

### Compensation for light power attenuation inside the tissue

In all experiments, we maintained light power constant on cell bodies located at different depths in the tissue by compensating for scattering with increased light power. To calculate the light power at the focal point as function of its depth in the brain slice, we designed an "inverse fiber" model. We used an existing model to calculate the spatial distribution of power in tissue for an optic fiber with given NA, tip radius and wavelength[103]. We modeled the power distribution for a point fiber (tip of 1 μm) with properties as in our system (1.0 NA and $\lambda$ = 940 nm). At a given distance $d$ below the fiber, the acceptance angle of the fiber, set by the NA, defines a circular plane centered at the fiber axis and perpendicular to the fiber axis. By symmetry, the integrated light density over this plane is equivalent to the light power at the focal point when an objective with the same NA focuses into depth $d$ in the tissue. With this model, we found that the dependence of light power on depth in the tissue fits monoexponential decay with attenuation length of $\tau$ = 147.6 μm. This value is consistent with the excitation attenuation length previously found for 920 nm in the mouse brain

in vivo ($\tau$ ~ 155 $\mu$m)[104]. We modulated the absolute light power accordingly throughout all experiments.

## Analysis of GCaMP6s fluorescence and validation of spiking in response to stimulation

The fluorescence trace recorded during a single spiral-pattern scan (as in Fig. 1d) was averaged such that a complete spiral was treated as one time point. To determine whether a targeted cell spiked in response to spiral scanning, we calculated the slope of GCaMP6s fluorescence using a linear fit over the raw GCaMP6s fluorescence trace obtained during the entire spiral-scan train (10 or 15 spirals). Notably, when scanning populations of cells in the presence of TTX, GCaMP6s fluorescence can still accumulate (Fig. 1h, middle), supposedly due to strong light-induced depolarization which can occur without spiking[44]. This spike-independent depolarization means that the GCaMP6s signal obtained while scanning in the presence of TTX is in itself not a reliable measure for determining spiking. However, we found that the slope of the raw GCaMP6s fluorescence trace across a spiral train generally decreases after application of TTX (Fig. 1g, bottom; Fig. 1h, right; and Fig. S3b). We therefore used the lower 95% confidence bound of the raw GCaMP6s fluorescence slope as a measure for spiking. Cells whose lower 95% confidence bound was negative were excluded from analysis. For calculation of GCaMP6s $\Delta F/F_0$, the first spiral in the train was taken as baseline ($F_0$), due to the slow onset kinetics of GCaMP6s relative to spiral duration.

## Exclusion of interneurons from connectivity analysis

The AAV vectors encoding for stCoChR and for GCaMP6s are both under the ubiquitous EF1α promoter. To express stCoChR and GCaMP6s in a sparse, random set of mPFC cells and map connectivity among random mPFC cells, we injected a low-titer AAV-Cre vector into the mPFC which was either under the EF1α promoter or under the more pyramidal cell-specific CaMKIIα promoter (see above, under *AAV expression plasmids*). This could lead to expression of stCoChR and GCaMP6s in interneurons. To minimize the resulting bias, we excluded from analysis any cell whose stimulation evoked hyperpolarizing currents in the recorded cell (under a −70 mV clamp). In some of the experiments, we repeated the sequential stimulation under a depolarized holding potential (−60 to −40 mV, and 0 mV in cases the intracellular solution was Cs-based) to facilitate identification of inhibitory connections. Since connections from interneurons to pyramidal neurons appear at very high probabilities, both in sensory cortex[6] and specifically in the mPFC[76,105], and since interneurons are estimated to amount to ~20% of cortical neurons[106], the remaining stimulated interneurons that were not connected to the recorded cell likely introduce a very small bias to connection probability.

## Summary of exclusion criteria

Stimulated (candidate presynaptic) cells were excluded from analysis if they met at least one of these conditions: Lower 95% confidence bound for the slope of the raw GCaMP6s fluorescence trace across the spiral train was negative; stimulation evoked a direct photocurrent larger than 20 pA in the recorded (postsynaptic) cell; stimulation evoked an IPSC; recording trace during stimulation was too noisy or leaky. To measure the performance of our connectivity determination model (Fig. 2e, f), we disregarded the GCaMP6s signal of the presynaptic cell (such that cell pairs where the presynaptic cell was not confirmed to have spiked were included in this particular analysis, unlike all other analyses), and we excluded from this analysis cell pairs where stimulation of the presynaptic cell induced any direct photocurrent (also <20 pA) in the recorded postsynaptic cell.

## Data analysis

Detection and modeling of synaptic events and analysis of synaptic connectivity were performed using custom scripts written in Matlab and in Python. Analysis of GCaMP6s fluorescence was performed using Matlab. Analysis of electrophysiological recordings for intrinsic properties was performed using Matlab and Clampfit (Molecular Devices). Image analysis was performed using Matlab and Fiji. Statistical analysis was performed in Matlab. Data are presented as mean ± SEM unless otherwise stated.

## Statistics

In comparisons of connectivity features between map types (Fig. S5a) and in comparisons of electrophysiological properties between cell types (Fig. 4d, e and Fig. S8a), we used the Kruskal-Wallis test with multiple post hoc comparisons using Tukey's Honestly Significant Difference (HSD) procedure. Two-way analysis of variance (ANOVA) was used to test interaction between connection type and layer (Fig. 3c). Wilcoxon signed-rank test was used for comparing weighted synaptic input along the mediolateral axis (Fig. S5f) and along the dorsoventral axis (Fig. S5g), for comparing connectivity measures above and below the recorded cell (Fig. S10), and for comparing spatial specificity (FWHM) of spiking vs. GCaMP6s $\Delta F/F_0$ (Fig. S4b). Two-sample Kolmogorov-Smirnov test was used for comparing properties of automatically vs. manually detected cells (Fig. S2e, f). To compare regression models to the model using all (and only) selected features (Fig. 4i), the same cross-validations were used for all models (except for the shuffled connections and mean connection prediction), and a paired-sample Student's t-test was performed. The model using shuffled connections did not use cross validation and therefore the comparison with the model using selected features was performed with a two-sample Student's t-test. Finally, to compare the mean probability prediction model, one-sample t-test was used.

## Markov chain Monte Carlo (MCMC) sampling parameters

The MCMC sampling in the cell connectivity determination was done using the NUTS algorithm in PyMC3 with 5000 steps for tuning for a 'target_accept' of 0.95 and 2000 steps after the tuning for sampling in 4 independent chains. We used the 'jitter+adapt_full' initialization. The MCMC sampling for the Horseshoe-prior logistic regression was done using the NUTS algorithm in PyMC3 with 10000 steps for tuning for a 'target_accept' of 0.95 and 2000 steps after the tuning for sampling in 4 independent chains. We used the 'advi+adapt_diag' initialization with a maximum of 50000 steps for initialization.

## Reporting summary

Further information on research design is available in the Nature Portfolio Reporting Summary linked to this article.

## Data availability

Source data are provided with this paper as spreadsheets for each of the figures. Raw data will be shared upon request.

## Code availability

All of the custom-written analysis code used in this study is available from the Yizhar lab Git repository at https://doi.org/10.5281/zenodo.7607227.

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

## Acknowledgements

We would like to thank all of the Yizhar lab members for critical comments and discussion of this manuscript. This work was supported by funding from the European Research Council (ERC CoG PrefrontalMap 819496) and by the European Union's Horizon 2020 research and innovation actions (H2020-RIA DEEPER 101016787). O.Y. is supported by the Joseph and Wolf Lebovic Charitable Foundation Chair for Research in Neuroscience.

## Author contributions

Y.P. and O.Y. designed the study and wrote the manuscript; Y.P. performed all experiments and data analyses; P.P. developed the automated event detection pipeline and performed statistical analysis of connectivity data; M.M. developed the initial automated two-photon stimulation paradigm; A.B. helped with statistical analyses; A.L. and R.L. performed histology and molecular biology; M.B. assisted with data analysis. All authors read and commented on the manuscript.

## Competing interests

The authors declare no competing interests.
