## [Peer Review File · Nature Communications]

Determinants of functional synaptic connectivity among amygdala-projecting prefrontal cortical neurons in male miceREVIEWER COMMENTS

Reviewer #1 (Remarks to the Author):

In this manuscript, Printz et al. describe a high-throughput optogenetic method for mapping large-scale (i. e. on a large number of cell-pairs) functional synaptic connectivity within medial prefrontal cortex (mPFC) neurons. The approach uses 2P-spiral scanning illumination of stCoChR-GCaMP6s co-expressing presynaptic cells and cell-attached recording of the postsynaptic cell. Near cellular resolution for presynaptic excitation is demonstrated by using the soma-restricted opsin stCoChR previously developed by Yizhar lab, and a volumetric analysis of mPFC-BLA cell distribution.

Photo-evoked activity on presynaptic cells is monitored by performing 2P Ca²⁺ imaging during cell stimulation. To this end, the authors demonstrated an elegant approach that only relies on one single laser to induce both presynaptic action potentials and simultaneously collect GCaMP fluorescence to verify successful spiking. Further they establish an extensive analysis pipeline on the one hand to map the tissue volume of interest and identify expressing, potentially presynaptic cells and on the other hand to analyse the resulting recordings.

The approach is very thoroughly done, all steps in the extensive protocol were calibrated carefully with appropriate controls done, such that the resulting data set is not only very impressive but also strongly reliable.

The system is used to investigate the connectivity and connectivity strength among mPFC-BLA cells, precisely among the cells in mPFC which project into the basolateral amygdala (BLA) cells.

This cell population is identified by expressing a Cre expressing vector into the BLA and Cre-dependent AAV vectors expressing stCoChR and GCaMP6s into the mPFC.

Connectivity mapping is performed and discussed within three groups of cells to probe connection of 1) mPFC-BLA onto mPFC-BLA 2) mPFC-BLA onto non-labeled (therefore likely non BLA projecting) cells 3) randomly chosen cells.

High throughput connectivity analysis using optogenetics is a timely and powerful approach and, in this paper, is for the first time used for the investigation of a specific neuronal circuit in acute slices. The manuscript is well written, the methodology well described and characterized. This also include a detailed description of all possible artifact that the approach could generate. The approach demonstrated is extendable to other brain circuits and as such I expect it to be of great interest for a large community of neuroscientists.

I have few suggestions to improve the clarity of the manuscript:

Page 5: the conditions chosen for the experiment: spiral duration 7.2ms, stimulation frequency 10Hz, light power 10 mW, are not shown in the Figure 1I or Figure 2SC, if the authors have these data, it would be better to include them in the figures.

Page 6: authors characterize the axial resolution looking at the spiking probability and DF/F signal; It would be important here to also give the optical axial resolution, how extended is the spot used for the

spiral scanning along the axial direction? Also could they give an explanation why the spatial specificity curve for GCaMP-DF/F is so much narrower than the spike probability? Is this because their imaging conditions are not sensitive enough to detect a single spike?

Figure 1: please complement the figure with some representative images (or a stack) of the recorded slices to show expression, co-expression, sparsity of the labelling etc. In the current manuscript there are no images of the actual slices that were used for the experiments.

From Figure 1E and in Fig. S2C, it seems that stimulation often gives rise to multiple spikes. Could the author comments whether this has an effect on the amplitude of the post-synaptic responses and how this is taken into account in the measurement of the synaptic strength?

Figure 1F, G show the DF/F and GCaMP fluoresce curve as a function of spiral number, respectively. This characterization is important as the amplitude and slope of the GCaMPs signal are used to select the post-synaptic cells for the connectivity experiments. Few points should be then explained more clearly. Specifically, Fig1F show largely varying DF/F values, from one cell to another, in one cell the signal is comparable to the recordings made in presence of TTX where one would expect no spike activity. Same in Fig.1G, multiple cells in presence of TTX show responses comparable, if not higher, to the aCSF case. The authors should discuss these curves and the diversity in the responses more extensively. For example, are all the curves taken at the same power? Or is the GcaMP imaging power chosen on the base of the opsin expression levels (higher power for low opsins expressing cells) so that the GCaMP fluoresce signal varied accordingly? Are their imaging conditions sufficient to detect single APs?

The authors mention that, even in presence of TTX (pag 30), GCaMP fluorescence can still accumulate “supposedly due to the strong light-induced depolarization”. Can the authors better clarify this statement and the nature of the effect? Also can they specify if cells that give these level of GCaMP transient also in presence of TTX, would pass the criterion to validate activation?

Page 6: For the automated detection of potential pre-synaptic cells the authors describe to use mScarlet and dTomato fluorescence, co-expressed with stCoChR and GCaMP respectively. I expect that the cells used for the connectivity mapping would all (most) express both the CoChR and GCaMP, however in the description of the automated detection it is not clear if this was based on the detection of co-expressing cells or if some of the detected cells only express the CoChR and no GCaMP or vice versa? Meaning the 209/316 cells detected in the volume (Page 6) correspond to cells expressing one of the two or co-expressing both?

What did the authors mean with observing the detections?

This also relates to my next point:

Does the cross-talk with GCaMP and mScarlet affect the cell mapping? Precisely, co-expressing cells should show GCaMP and mScarlet fluorescence in the cytosol, and dTomato fluorescence in the nucleus. Detecting cytosolic mScarlet informs about the presence of CoChR, but in case of cross talk, how do they

distinguish that from GCamP, which is also in the cytosol? This is important if they screen for co-expression directly, which is not clear (see previous comment). Maybe they just map the volume for any expression, stimulate all cells and discard the ones that are not responding?

Fig. 2D: The authors show the expected shape of the EPSC according to the model used for the automatic identification of connections. In the reported example, the onset of the response is perfectly aligned with the start of the stimulation, while in the experimental curve (Fig2C) the response is temporally delayed, could the author clarify this point and define what criterion is used to distinguish EPSC from direct photoactivation. Could the authors comment on that?

Discussion: In the discussion, the authors mention previous work achieving optogenetic based synaptic mapping saying that they are “restricted to measuring only unidirectional (in-degree) connectivity”. Could they better clarify this idea?

Discussion: in the discussion it would help to discuss (if any) the limitations of the presented technology and give an outlook on what could be improved (e.g. light delivering approach, detection, opsin expression and targeting....)

Pag 28/29: the authors give a detailed description on how they discriminated among direct photocurrents and EPSC and confirm that the criteria that they used work by comparing the currents recorded in presence of glutamate-receptor blockers. It would help if they could here also quantify how many cells they typically excluded for experiments. Also, supposing that the presence of artifactual photocurrent decreases with distance, they should comment on how this could affect the overall estimation of connectivity ratio and spatial distribution of connections.

Pag 28: “were manually examined for synaptic connections by searching the stimulation-aligned traces for reliable, low-jitter EPSC occurrences.” Authors should define here the meaning of low-jitter EPSC, as the above discussion they made to distinguish photoactivation from EPSC based on the fast responses of photoactivation could be otherwise confusing.

Figure S6: the authors discuss the possibility of short-term plasticity induced by repeated spiral stimulation, which gives rise to a greater EPSC amplitude for the first 3 or 4 stimulations. According to Figure 1E, it seems that the first 2/3 spirals can also generate double spikes which would probably also generate higher EPSC. Could the authors comment on how these two effects can be distinguished?

The authors report EPSC down to around 5 pA. In methods they mention a threshold of 2.5 SD. What does this value correspond to practically? Are connectivity rate values they observe potentially affected by this precision? I suggest the authors better discuss their detection sensitivity at the post-synaptic site.

Minors

It would be useful to supplement figS1D and E with some actual images from the stacks show examples of cells detected manually and automatically.

Fig1C: I presume, based on the color code of the first column, that second and third column have been recorded with a 15 μm spiral: it will help to give this information.

Page 5: Throughout the characterization of the stimulation, it is not very clear if the 7.2 ms spirals are always two 3.6 ms spirals (same dwell time)?

Page 55, Fig S7 A: Please include a legend to explain the three groups/colours or include it in the legend of this figure

FigS1 Panels A, B and C,D,E are describing different experiments. The cell detection was performed (and characterized) on the slices used for connectivity recording (and not on the Ai9 mice?). It could be confusing for the reader to have this in the same figure.

Discussion: "Here we presented a large-scale implementation of such an approach, combining it with calcium-based readout of activity using a single laser source." Better probably to repeat here the FOV and the achieved number of recorded pairs.

Methods page 30, considering that the τ_{decay} is used to define a scattering length it would probably better to call it this way and also to compare the found value (147.6 μm) with what is given in the literature.

Reviewer #2 (Remarks to the Author):

The study by Printz et al. entitled "High-throughput mapping of functional synaptic connectivity in the prefrontal cortex" employs a novel and promising method combining whole-cell patch-clamp recordings with 2-photon microscopic imaging and simultaneous optogenetic stimulation of projection-defined neurons in acute prefrontal brain slices to map the functional connectivity of the local excitatory circuit.

The authors characterize several features defining the probability of synaptic connections between retrogradely labelled BLA-projecting prefrontal neurons, mPFC-BLA neurons, and non-mPFC-BLA neurons, and between random mPFC neurons, respectively. The authors show that the major determinants of local connectivity within mPFC are the distance between the cell bodies and the anatomical position of cells, especially along the mediolateral axis of the mPFC.

The methods of the study are well described and we positively note the great care and detail the authors deploy in characterising e.g. the size and duration of scan parameters, and the clear highlighting

and discussion of limitations of the methodology.

While we are positive about the study, we find a need for clarifications and further discussion of particular issues, and a few experimental parameters needing to be addressed.

The viral strategy: while the calibration of expression of retrograde AAV vectors is well described, we fail to understand what exact viral strategy was actually used in experimental animals. We also find the images of BLA targeting suboptimal, as they don't allow evaluation of the specificity of the viral labelling. In line with this, it appears that the quantifications of labelling at the injection site focuses on the BLA specifically but say nothing about the degree of unspecific labelling in other amygdalar nuclei, or elsewhere. This is a central point, as a big part of the study regards mPFC-BLA neurons specifically. We find that the specificity needs to be appropriately addressed and accounted for. Overall we would welcome more images, including of cell body labelling in the mPFC (e.g. lamination) as to understand what neurons are interrogated, and the spatial distribution of the subnetwork identified.

mPFC anatomy: anatomy is central to both the study, and its findings, and great progress has been made in anatomical mapping of the mouse brain over the past decade. We find the anatomical delineations and nomenclature in the current study unclear. An, to us, unknown atlas has been used that does not appear to adhere to the most commonly used digital atlas (created by the Allen Institute for Brain Science). As the definition of the PFC varies among researchers, it is of importance that cell locations can be understood/re-mapped by colleagues in the field. We find the study would be stronger if it was better conveyed how the PFC subregions are delineated anatomically, i.e. if the reader could understand e.g. what part of the tissue the authors denote IL, DP, cingulate. This would help understanding and personal interpretation of the cell locations. There is only a quite small single image (Fig 1B; is this adhering to <https://kimlab.io/brain-map/atlas/>). We also recommend finding space in the main text to at least briefly state what atlas was used, why, and how.

Also related to anatomy:

DP - It is unclear why the DP is included in the study, and, thus, here included in the mPFC. This needs to be addressed/justified/discussed, including in relation to the BLA and topics addressed in the Discussion.

Division of prefrontal layers - we find it unclear how the layers were defined/identified, and why (and how) the authors separate layers 2 and 3? Furthermore, why are layers 5 and 6 pooled (e.g. Figure 3D); how can this be justified? The laminations applied goes against some conventions in the PFC field (and atlases), and particularly, it is known that L5 and L6 hold differential input and output patterns (e.g. Harris and Shepherd, 2015). In relation to this: in Fig 3D it is shown that for the mPFC-BLA-projecting subnetwork (red), L5/6 receives higher weighted input than L2. However, in Fig S5D it appears that L3 (not reported in Fig 3D) receives similar input as L5/6(?) We find that the partly unconventional handling of the layers together with the lack of L3 data in parts of analysis (Fig 3D) cause some confusion. Is it possible to make Fig 3D and S5D understandable together? The laminar aspect is important as a central claim is that 'the weighted output from mPFC-BLA cells onto other mPFC-BLA cells was stronger in deeper layers than in superficial ones'.

Furthermore, in Fig S5D it appears that non-mPFC-BLA projecting and random mPFC cells in layer 5/6 (3rd column, blue and black) do not receive any input from layer 2/3, which would be surprising considering assumed canonical cortical circuit models. This finding also needs to be incorporated into any conclusions.

We find it unclear if the regression in Fig 4H was performed on data from all 'pairs' of cells without regard to their class (mPFC-BLA - mPFC-BLA; mPFC-BLA - Non-mPFC-BLA etc.). It would be of interest to understand if the regression coefficients differ between the classes (and if no differences are found, how can then the demonstrated differences in connectivity between different mPFC subnetworks be understood?)

The majority of patched cells are located in IL (75% (69/92)). It would be of interest to know, if the authors hold the data, whether the connectivity between prefrontal subregions differs (e.g. PL vs. IL) or if the findings in IL can be generalized to other (also not prefrontal?) regions.

Minor comments:

The colors used in Fig S1B (IL, low/high titer) are difficult to distinguish, particularly for the circles.

Figure 2G shows a 'representative synaptic connectivity map' with a patch recorded cell in DP. As the majority of patched cells (69/92) were recorded in IL and only 6/92 in DP, why was that example chosen? See also comment about DP above.

The sentence 'Moreover, the intrinsic properties of the postsynaptic mPFC cell and anatomical position of both cells jointly account for...' in the abstract is difficult to follow (if one has not already read the article) - postsynaptic to what, and what two cells are meant ('both cells')? Also at other places it is at times hard to know what cell(s) is meant - perhaps consider to avoid only saying cell, and instead consistently state post-/presynaptic, mPFC-BLA cell etc...

Fig 3E, F: Can this data be translated into layers/mPFC subregions? It would allow interpretation of possible differences in weighted input between subregions.

The abstract ends with 'Our findings demonstrate a functional segregation of mPFC excitatory neuron subnetworks, and reveal the factors determining connectivity in the mPFC'. This appears a bit of an overstatement - the study is purely ex vivo and functional synapses are examined rather than the function of segregated subnetworks. Also, only a single specific subnetwork is investigated.

Fig S5D: while a summary figure is great for an overview, the array of scales on the y-axes makes it very hard to actually compare the connections - could perhaps fewer scales be used?

Reviewer #3 (Remarks to the Author):

This study explores the connectivity of neurons in the prefrontal cortex using a high-throughput single cell optogenetic mapping approach. They report a number of interesting properties related to local circuit connectivity, focusing on mPFC neurons that project to the BLA and how their connectivity relates to other cells in the network. The manuscript is well written, the technological approach is rigorous and the topic investigated is an interesting one that will be of relevance to the field. However, my main objections are the way that the data are presented and some issues around the way cells are sampled in different groups. Without more detailed analysis it is difficult to see how much of their findings depend on grouping heterogeneous populations into a single sample.

Methods

The authors adjusted laser power across depth based on theoretical considerations but was this effective, was p-spike similar across this depth based on their fluorescence data?

Their approach is highly dependent on the reliability of their spike and EPSC detection algorithms. If they run their connection detection model for traces where a presynaptic spike was not detected, what percentage of traces yield an input? This would be a useful control to confirm both their spike detection capability with GCaMP imaging and also their EPSC detection model.

Was there a difference in connectivity based on internal solution used, did using Cesium internal for a subset of recordings influence their ability to detect inputs?

Results

L5 and L6 have very different connectivity. Even within L5 descending input from superficial layers drops off significantly as a function of depth (for example Anderson et al., 2010 Nat Neurosci.). Do they observe this effect in mPFC and was there a difference in the M-L depth L5/6 cells across groups?

They should report p-Conn and EPSC amplitude across layers, not just for the sum input. This would help people trying to model the mPFC understand the synaptic properties within and across layers.

The M-L analysis is kind of meaningless without knowing the location of the postsynaptic cell. A cell at the border of L1/2 will always have lateral input, while a L6 cell will always have input that is medial, the question is which layers provide those inputs. Given we know that the cortex is organized into precise layers and that connectivity is tightly correlated to the laminar location of both pre and postsynaptic neurons, choosing to use an arbitrary M/L axis seems somewhat meaningless. This is all the more problematic when you look at the sampling of layers in each population. The BLA \diamond BLA sample is biased towards L5/6 (L5/6 = 50% of all recorded cells), the BLA \diamond non-BLA sample is biased towards L2/3 (L5/6 =

21% of all cells) while random to random is also slightly L2/3 biased (44% in L5/6). Even ignoring the fact that we know connectivity is driven by layer location, without equal sampling across the depths in each group this type of M-L comparison is almost impossible to interpret.

Figure 3B, do these maps represent the maps for all the recorded cells aligned to the location of the soma? What if you align them to the depth/layer they were recorded? See above, this is hard to interpret without knowing where the cells are located.

Why was an arbitrary 300 μ m radius chosen to analyze connectivity? For cells deep in L5 or L6 I presume this would mean that you aren't sampling most of the superficial layer input (where many BLA cells are located). In the ML (intralaminar) direction translaminar inputs can span many 100s of microns, see point above regarding drop off of input from superficial to deep layers based on depth. More detailed analysis is needed to see if this approach is actually warranted or if they are artificially introducing connectivity patterns to their data by only subsampling the most proximal inputs.

Discussion

The authors make a valid point about how most historical paired recording studies are restricted to the superficial sublayer of the slice (first 50-100 μ m) and perform some helpful analysis related to this point that is in the supplemental discussion. It is a shame that is not more clearly signposted in the main text.

Discussion of presynaptic release should mention that they use 2mM Ca²⁺ which is possibly high compared to in vivo levels (estimated to be 1mM, See Seeman et al 2018 eLife) and could bias towards depressing synapses.

Reviewer #4 (Remarks to the Author):

The authors present a comprehensively validated optical method to determine the presynaptic connectivity of electrophysiologically recorded neurons in mouse brain slices. They use this method to determine key factors regarding the fine-scale circuit structure of mPFC. This method builds on previous approaches by adding the ability to know the projection target of the recorded neuron and its presynaptic partners while at the same time maintaining sufficiently low expression to avoid off-target activation. Given the low likelihood of connection in the particular circuit under study here, a method such as this is effectively critical to find the specificity the authors show is extant in these circuits. Fig S3 in particular is a crucial, necessary, and difficult control that has been well performed. The careful & rigorous analysis of the cells that could have been inaccurately photostimulated -- which turns out to be low given the combined sparse expression and spatial resolution of the method -- is particularly laudable. This manuscript is suitable for immediate publication in Nature Communications following some minor revisions. No further experiments are required.

Major comments:

GCaMP underreports spiking as previously shown and also apparent in the authors Figure S3. Is it possible the low overall probability of connection in S5 (even lower than the normally low probability of cortical connections amongst excitatory cells) is due to this effect? Probably not, given their low probability of connection is very similar to that found with paired recordings in ref 70. However I found one of the authors statements in the text, " 117.6 ± 5.0 cells (of which 95.4 ± 5.1 are confirmed to respond to stimulation via the GCaMP6s signal) ...", confusing. Did they not ALWAYS use confirmation of the presynaptic neuron's spiking to confirm the presynaptic connectivity? This does not appear to be the case from the methods describing the connectivity assessment, which are wonderfully detailed (but I may have missed it). Or did the authors use this in the case of the manual assessment, which was used to validate the model-based assessments, perhaps? One more sentence in the methods to clearly disambiguate this situation would be appreciated, especially given this sentence found elsewhere: "Stimulated (candidate presynaptic) cells were excluded from analysis if they met at least one of these conditions: Lower 95% confidence bound for the slope of the raw GCaMP6s fluorescence trace across the spiral train was negative; ... "

If the authors might be able to quantify the co-expression of stCoChR and GCaMP6s in the mPFC->BLA cells, this would be helpful. It is not absolutely critical, but given they are expressing these proteins from two different viruses, it could be useful to know. Analysis of whatever images of expression from their experiments they already would be helpful if available.

Minor

10, 15, and 20 um spiral legends (in Figs 1i and S2c) are mildly confusing.

Diagram at the bottom right of Fig S3e is not described.

S5 is presented before S4.

Tukey's may be informative for Fig 3d.

Some experiments used K-based internal solution for patching while others used Cs-based. Perhaps the experiments that used Cs-based internal could not be used for analyses requiring the firing pattern of the patched cell, but they were used for all other analyses? A sentence in the methods clarifying this situation could be helpful here.

The authors may want to consider citing other papers using high precision circuit mapping such as Nikolenko et al (<https://www.nature.com/articles/nmeth1105>), Packer et al 2012 (<https://www.nature.com/articles/nmeth.2249>), Naka et al 2019 (<https://elifesciences.org/articles/43696>), and the very recent Hage et al 2022 (<https://elifesciences.org/articles/71103>).

REVIEWER COMMENTS

We would like to thank the reviewers for their thorough reading of our manuscript, and for their helpful comments. We have addressed all the comments raised by the reviewers, which we believe has made the manuscript clearer and better. Below is our point-by-point discussion of the reviewers' comments (our comments in **blue**).

Reviewer #1 (Remarks to the Author):

In this manuscript, Printz et al. describe a high-throughput optogenetic method for mapping large-scale (i. e. on a large number of cell-pairs) functional synaptic connectivity within medial prefrontal cortex (mPFC) neurons. The approach uses 2P-spiral scanning illumination of stCoChR-GCaMP6s co-expressing presynaptic cells and cell-attached recording of the postsynaptic cell. Near cellular resolution for presynaptic excitation is demonstrated by using the soma-restricted opsin stCoChR previously developed by Yizhar lab, and a volumetric analysis of mPFC-BLA cell distribution. Photo-evoked activity on presynaptic cells is monitored by performing 2P Ca²⁺ imaging during cell stimulation. To this end, the authors demonstrated an elegant approach that only relies on one single laser to induce both presynaptic action potentials and simultaneously collect GCaMP fluorescence to verify successful spiking. Further they establish an extensive analysis pipeline on the one hand to map the tissue volume of interest and identify expressing, potentially presynaptic cells and on the other hand to analyse the resulting recordings.

The approach is very thoroughly done, all steps in the extensive protocol were calibrated carefully with appropriate controls done, such that the resulting data set is not only very impressive but also strongly reliable.

The system is used to investigate the connectivity and connectivity strength among mPFC-BLA cells, precisely among the cells in mPFC which project into the basolateral amygdala (BLA) cells.

This cell population is identified by expressing a Cre expressing vector into the BLA and Cre-dependent AAV vectors expressing stCoChR and GCaMP6s into the mPFC.

Connectivity mapping is performed and discussed within three groups of cells to probe connection of 1) mPFC-BLA onto mPFC-BLA 2) mPFC-BLA onto non-labeled (therefore likely non BLA projecting) cells 3) randomly chosen cells.

High throughput connectivity analysis using optogenetics is a timely and powerful approach and, in this paper, is for the first time used for the investigation of a specific neuronal circuit in acute slices. The manuscript is well written, the methodology well described and characterized. This also include a detailed description of all possible artifact that the approach could generate. The approach demonstrated is extendable to other brain circuits and as such I expect it to be of great interest for a large community of neuroscientists.

I have few suggestions to improve the clarity of the manuscript:

Page 5: the conditions chosen for the experiment: spiral duration 7.2ms, stimulation frequency 10Hz, light power 10 mW, are not shown in the Figure 1I or Figure 2SC, if the authors have these data, it would be better to include them in the figures.

The reviewer is correct in that the combination of parameters used in the connectivity-mapping experiments – namely duration of 7.2 ms, frequency of 10 Hz, and power of 10 mW – does not appear in the dataset described in Figure 1I and Figure S2C (now changed to Figure S3C). We do not have similar calibration data for this exact parameter combination, since we performed calibration for each parameter separately and chose the optimal ones after their individual calibration.

Page 6: authors characterize the axial resolution looking at the spiking probability and DF/F signal; It would be important here to also give the optical axial resolution, how extended is the spot used for the spiral scanning along the axial direction? Also could they give an explanation why the spatial specificity curve for GCaMP-DF/F is so much narrower than the spike probability? Is this because their imaging conditions are not sensitive enough to detect a single spike?

To address this request, we have included the optical resolution of the scanning as PSF measurement in Figure S4. The reason that the GCaMP6s spatial specificity curve is narrower than that of stCoChR-induced spiking is the very high light sensitivity of stCoChR as compared with GCaMP6s, resulting in higher light power needed for a resolvable GCaMP6s signal.

Figure 1: please complement the figure with some representative images (or a stack) of the recorded slices to show expression, co-expression, sparsity of the labelling etc. In the current manuscript there are no images of the actual slices that were used for the experiments.

Thank you for the suggestion, which we completely agree with. We added a representative z-projection image to Figure S2C. It shows the density of labeled mPFC-BLA cells in an imaged volume used for mapping connectivity. Regarding co-expression of stCoChR and GCaMP6s, since both are labeled with a red fluorophore and the basal GCaMP6s fluorescence (without activity) is too low, we performed anti-GFP staining to enhance the GCaMP6s signal. Images and quantification of stCoChR and GCaMP6s appear in Figure S2B. Notably, cells without co-expression would be discarded in analysis due to lack of GCaMP6s signal, either because of lack of stCoChR (and therefore spiking) or lack of GCaMP6s.

From Figure 1E and in Fig. S2C, it seems that stimulation often gives rise to multiple spikes. Could the author comments whether this has an effect on the amplitude of the post-synaptic responses and how this is taken into account in the measurement of the synaptic strength?

A brief explanation and a reference to the relevant detailed Methods section were added to the main text. It now states on page 7: “Finally, we calculated the strength of synaptic connection at each stimulation as the weighted average of EPSCs within a time window following stimulation (see Methods), thus accommodating for jitter in synaptic latency

and the possibility for multiple evoked EPSCs (Figure S3C).” The weighted average used to calculate synaptic strength was based on a normal fit of the EPSC distribution, as described in Methods under “Measurement of synaptic connection strength”.

Figure 1F, G show the DF/F and GCaMP fluoresce curve as a function of spiral number, respectively.

This characterization is important as the amplitude and slope of the GCaMPs signal are used to select the post-synaptic cells for the connectivity experiments. Few points should be then explained more clearly. Specifically, Fig1F show largely varying DF/F values, from one cell to another, in one cell the signal is comparable to the recordings made in presence of TTX where one would expect no spike activity. Same in Fig.1G, multiple cells in presence of TTX show responses comparable, if not higher, to the aCSF case. The authors should discuss these curves and the diversity in the responses more extensively. For example, are all the curves taken at the same power? Or is the GcaMP imaging power chosen on the base of the opsin expression levels (higher power for low opsins expressing cells) so that the GCaMP fluoresce signal varied accordingly? Are their imaging conditions sufficient to detect single APs?

The variation in GCaMP6s responses between cells most likely stems from variability in expression levels of both GCaMP6s and stCoChR, resulting in varying spiking efficiency as well as varying GCaMP6s signals for the same spiking behavior. Another factor that could contribute to this variation is the shape of the cell relative to the spiral pattern scanned on it. The light power was consistent across cells and was not adjusted to expression level (but was adjusted to depth in the tissue as explained in the Methods in order to maintain constant power between cells). stCoChR evokes very strong depolarizations (as characterized in Forli et al., eLife 2021), such that even in the presence of TTX, photostimulation can result in calcium influx through voltage-gated calcium channels, which in turn can increase the GCaMP6s signal without the cells spiking. We believe that this is the reason that some of the cells show strong GCaMP6s signal increase in presence of TTX. We therefore did not use the GCaMP6s signal in presence of TTX to determine a threshold for spiking, as the strong depolarization confounds the interpretation of this signal. Finally, our imaging conditions are not sufficient to detect single spikes but rather tell us whether spikes occurred during a stimulation train.

The authors mention that, even in presence of TTX (pag 30), GCaMP fluorescence can still accumulate “supposedly due to the strong light-induced depolarization”. Can the authors better clarify this statement and the nature of the effect? Also can they specify if cells that give these level of GCaMP transient also in presence of TTX, would pass the criterion to validate activation?

Please see the answer to the previous comment. Additionally, a depolarization amplitude that increases GCaMP6 signals in the presence of TTX would undoubtedly trigger spiking in a natural setting with no TTX. Therefore, it is highly likely that cells showing large increase in GcaMP6s signal with TTX would spike without TTX and pass the inclusion criteria.

Page 6: For the automated detection of potential pre-synaptic cells the authors describe to use mScarlet and dTomato fluorescence, co-expressed with stCoChR and GcaMP respectively. I expect that the cells used for the connectivity mapping would all (most) express both the CoChR and GcaMP, however in the description of the automated detection it is not clear if this was based on the detection of co-expressing cells or if some of the detected cells only express the CoChR and no GcamP or vice versa? Meaning the 209/316 cells detected in the volume (Page 6) correspond to cells expressing one of the two or co-expressing both? What did the authors mean with observing the detections?

In our experiments, we cannot distinguish between stCoChR expression and GCaMP6s expression due to the significant overlap between the spectra of dTomato and mScarlet. We rely on the fact that targeted cells which express only one of the two will not show GCaMP6s increase, and these cells will therefore be discarded during analysis.

This also relates to my next point:

Does the cross-talk with GCaMP and mScarlet affect the cell mapping? Precisely, co-expressing cells should show GCaMP and mScarlet fluorescence in the cytosol, and dTomato fluorescence in the nucleus. Detecting cytosolic mScarlet informs about the presence of CoChR, but in case of cross talk, how do they distinguish that from GCaMP, which is also in the cytosol? This is important if they screen for co-expression directly, which is not clear (see previous comment). Maybe they just map the volume for any expression, stimulate all cells and discard the ones that are not responding?

This is precisely what we did – scan all cells that express a red fluorophore (without distinction between nuclear and cytosolic expression) and include only responsive ones.

Fig. 2D: The authors show the expected shape of the EPSC according to the model used for the automatic identification of connections. In the reported example, the onset of the response is perfectly aligned with the start of the stimulation, while in the experimental curve (Fig2C) the response is temporally delayed, could the author clarify this point and define what criterion is used to distinguish EPSC from direct photoactivation. Could the authors comment on that?

In all our data, we observed that direct photocurrents appear at ≤ 1.5 ms latency, minimal jitter, and consistent waveform with slow rise constant compared with EPSCs. This can be seen in Figure 2C. The seemingly zero-latency EPSC response in Figure 2D is due to the filtering effect of the KDE. Throughout our experimental dataset, EPSCs always appeared with at least few ms latencies.

Discussion: In the discussion, the authors mention previous work achieving optogenetic based synaptic mapping saying that they are “restricted to measuring only unidirectional (in-degree) connectivity”. Could they better clarify this idea?

A clarification was added to the Discussion text. The intension is to say that only connections from the stimulated cells onto the recorded cell can be probed, and not in the reverse direction. Notably, after mapping connections from stimulated cells onto the recorded cell, another experiment can be conducted where one of the stimulated cells is now recorded and the previously recorded cell is now stimulated, but the recorded cell usually does not stay viable after a mapping experiment to consistently allow such an approach.

Discussion: in the discussion it would help to discuss (if any) the limitations of the presented technology and give an outlook on what could be improved (e.g. light delivering approach, detection, opsin expression and targeting....)

The main limitation of our approach lies in the jitter of spiking and synaptic responses which stems from the spiral stimulation method. This is inherent to spiral stimulation, but our analysis is still able to take the jitter into account using the modeling approach we describe in Fig. 2. Nevertheless, future work could utilize more advanced holographic methods to improve spike time precision and reduce jitter. A relevant text was added to the Discussion.

Pag 28/29: the authors give a detailed description on how they discriminated among direct photocurrents and EPSC and confirm that the criteria that they used work by comparing the currents recorded in presence of glutamate-receptor blockers. It would help if they could here also quantify how many cells they typically excluded for experiments. Also, supposing that the presence of artifactual photocurrent decreases with distance, they should comment on how this could affect the overall estimation of connectivity ratio and spatial distribution of connections.

The exact numbers of cells that were excluded in each experiment can be found in Table 2: it details, for each experiment, the number of stimulated cells, of which the number of cells inferred to have spiked in response to stimulation, and the number of cells included in analysis, such that the difference between the number of spiking cells and the number of included cells reflects the number of cells that were excluded due to large evoked photocurrents (or noisy recording, in some cases).

To address the second point concerning the dependence of evoked/artefactual photocurrents on distance from the recorded/postsynaptic cell, we have explicitly calculated this dependence (Rebuttal Figure 1). We found that as the reviewer suggested, the density of cells that evoke large photocurrents (PC) in the recorded cell, and are therefore excluded from analysis, decreases with distance from the recorded cell, whereas the distance of cells that do not evoke large photocurrents in the recorded cell (no PC) distributes normally. This could introduce a bias in estimation of connection probabilities, since relatively more close cells are excluded than farther cells. However, the absolute numbers of excluded cells, as seen in the bottom histogram, is small. Moreover, Figure S5B shows that at short distances (and also long distances), the mPFC-BLA to mPFC-BLA connections – where we can find PC-evoking cells – are not sparser than the other connection types – where no PC-evoking cells are found. Therefore, although a bias is theoretically predicted, we did not find it in the relevant dataset.

Rebuttal Figure 1. Quantification of cells whose stimulation evokes direct photocurrent in the recorded cell by distance from the recorded cell. Top, fraction of cells; bottom, number of cells.

Pag 28: "were manually examined for synaptic connections by searching the stimulation-aligned traces for reliable, low-jitter EPSC occurrences." Authors should define here the meaning of low-jitter EPSC, as the above discussion they made to distinguish photoactivation from EPSC based on the fast responses of photoactivation could be otherwise confusing.

The use of "low-jitter" here comes to differentiate such cells from randomly distributed EPSCs, which can sometimes appear to occur near the time of stimulation, but are more noisy and have higher "jitter". As can be seen in the third trace in Figure 2C, EPSCs can be distinguished from direct photocurrents by their rapid rise time as compared with photocurrent rise time, by their longer latency, and by the variance in their waveform (especially amplitude) between repetitions. We used these characteristics to identify cells whose stimulation evokes photocurrent and cells whose stimulation evokes EPSCs 'riding' on top of a photocurrent.

Figure S6: the authors discuss the possibility of short-term plasticity induced by repeated spiral stimulation, which gives rise to a greater EPSC amplitude for the first 3 or 4 stimulations. According to Figure 1E, it seems that the first 2/3 spirals can also generate double spikes which would probably also generate higher EPSC. Could the authors comment on how these two effects can be distinguished?

Our method does not allow complete control over the number of presynaptic spikes per stimulation, as shown in Figure S3C. Therefore, as the reviewer observed, calculation of short-term plasticity is inaccurate, and we mentioned this limitation in the Discussion. However, since this same limitation applies to all connection types that we have measured, we assume that differences in this plasticity estimation would reflect true differences between the connection types, and therefore chose to include this measurement in a Supplementary Figure. [As another reviewer pointed out, since these experiments are done in aCSF which has relatively high calcium concentration, the short-term plasticity features of synaptic connections are quite uniform and tend to be biased to strong adaptation, occluding any potential variation among populations].

The authors report EPSC down to around 5 pA. In methods they mention a threshold of 2.5 SD. What does this value correspond to practically? Are connectivity rate values they observe potentially affected by this precision? I suggest the authors better discuss their detection sensitivity at the post-synaptic site.

The value of 2.5 SD was added to the text under the relevant Methods section. As with any measurement technique, our ability to detect small synaptic events is limited by the sensitivity of our system. The 2.5 SD value represents the sensitivity of our system as the minimal EPSC amplitude which can be detected, and it is therefore the minimal strength of synaptic connection which we can detect. Strength of synaptic connections in the cortex is skewed, with many weak connections and few strong ones for each cell (for example, Cossell... Mrcic-Flogel, Nature 2015). Since our method might miss the weakest connections, we focused in our measurements on the weighted input that each cell receives, which represents the overall effect that the network has on the activity and excitability of the post-synaptic cell.

Minors

It would be useful to supplement figS1D and E with some actual images from the stacks show examples of cells detected manually and automatically.

Agreed. These images were added as Figure S2C.

Fig1C: I presume, based on the color code of the first column, that second and third column have been recorded with a 15 um spiral: it will help to give this information.

Assuming that the intension is to Figure S2C (now changed to Figure S3C) where there was a color code for the spiral size at the first column, we have changed the legend of this Figure to resemble that of Figure 1I which presents a similar dataset and where the spiral size color code appears on top of all columns.

Page 5: Throughout the characterization of the stimulation, it is not very clear if the 7.2 ms spirals are always two 3.6 ms spirals (same dwell time)?

The reviewer is correct, the dwell time is constant and the change in duration was achieved by concatenating spirals. A 7.2 stimulation is two 3.6 ms spirals (whereas one spiral is one round of in-and-out scan, as shown in Figure 1C, right image). A clarification was added to the Results text in page 5 where the surveying of scan parameters is described.

Page 55, Fig S7 A: Please include a legend to explain the three groups/colours or include it in the legend of this figure

A legend was added to the Figure. Please note that the figure numbering was changed and this figure now appears as Figure S8.

FigS1 Panels A, B and C,D,E are describing different experiments. The cell detection was performed (and characterized) on the slices used for connectivity recording (and not on the Ai9 mice?). It could be confusing for the reader to have this in the same figure.

We split Figure S1 into two separate figures and added illustrations to facilitate distinction between the different experiments presented in these figures.

Discussion: "Here we presented a large-scale implementation of such an approach, combining it with calcium-based readout of activity using a single laser source." Better probably to repeat here the FOV and the achieved number of recorded pairs.

Thank you for the suggestion. These details are now explicitly mentioned in this section of the Discussion: "Our semi-automated approach for cell detection and for sequential stimulation and calcium recording allowed us to probe the input from 95.4 ± 5.1 cells (mean \pm s.e.m), whose spiking in response to stimulation was validated using the GCaMP6s signal, onto each recorded cell in three dimensions within a volume of $\sim 420 \times 420 \times 300 \mu\text{m}^3$ ".

Methods page 30, considering that the τ decay is used to define a scattering length it

would probably better to call it this way and also to compare the found value (147.6 μm) with what is given in the literature.

We have changed the name of the constant to 'attenuation length' (since we believe it reflects a combination of scattering and absorption) and added a comparison to the reported literature.

Reviewer #2 (Remarks to the Author):

The study by Printz et al. entitled "High-throughput mapping of functional synaptic connectivity in the prefrontal cortex" employs a novel and promising method combining whole-cell patch-clamp recordings with 2-photon microscopic imaging and simultaneous optogenetic stimulation of projection-defined neurons in acute prefrontal brain slices to map the functional connectivity of the local excitatory circuit.

The authors characterize several features defining the probability of synaptic connections between retrogradely labelled BLA-projecting prefrontal neurons, mPFC-BLA neurons, and non-mPFC-BLA neurons, and between random mPFC neurons, respectively. The authors show that the major determinants of local connectivity within mPFC are the distance between the cell bodies and the anatomical position of cells, especially along the mediolateral axis of the mPFC.

The methods of the study are well described and we positively note the great care and detail the authors deploy in characterising e.g. the size and duration of scan parameters, and the clear highlighting and discussion of limitations of the methodology.

While we are positive about the study, we find a need for clarifications and further discussion of particular issues, and a few experimental parameters needing to be addressed.

The viral strategy: while the calibration of expression of retrograde AAV vectors is well described, we fail to understand what exact viral strategy was actually used in experimental animals. We also find the images of BLA targeting suboptimal, as they don't allow evaluation of the specificity of the viral labelling. In line with this, it appears that the quantifications of labelling at the injection site focuses on the BLA specifically but say nothing about the degree of unspecific labelling in other amygdalar nuclei, or elsewhere. This is a central point, as a big part of the study regards mPFC-BLA neurons specifically. We find that the specificity needs to be appropriately addressed and accounted for. Overall we would welcome more images, including of cell body labelling in the mPFC (e.g. lamination) as to understand what neurons are interrogated, and the spatial distribution of the subnetwork identified.

The specificity of mPFC-BLA cell labeling is indeed central to this study, and we value the reviewer's comments on this issue. To address this, we have included in Figure S2

confocal images of the BLA and the mPFC of wildtype mice injected with rAAV2-retro-Cre into the BLA and DIO-stCoChR+GCaMP6s into the mPFC. To enhance visibility of axonal projections labeled with stCoChR and GCaMP6s, we stained slices for GFP. We show that the densest axonal projections from labeled mPFC cells are inside the BLA. These are now shown as Figure S2A,B. We have further included a Supplementary Discussion section (subtitled “Specificity of mPFC-BLA cell labeling”) detailing the known projections from the infralimbic cortex to the region surrounding the BLA, to conclude that the likelihood of unspecific labeling exists, but is low. Briefly, the majority of labeling outside of the BLA (as seen in Figure S1A) was in the striatal region dorsal to the BLA, presumably due to viral leak from the injection needle during its withdrawal. However, based on the extensive characterization of the mouse brain-wide BLA connectivity (Hintiryan... Dong, Nat Comm 2021) and on the Allen Institute connectivity database, projections from the infralimbic cortex into this striatal region are very sparse compared with its projections into the BLA. Moreover, retrograde and anterograde labeling from the BLA result in labeling patterns similar to those we show in Figure S1A, and to a laminar distribution similar to that shown in Figure S2B. These observations suggest that non-specific retrograde labeling in the mPFC is minor compared with BLA-projecting cell labeling. Finally, ~55% of our mPFC-BLA cell mapping experiments were performed after injection of 0.1 μ l of rAAV2-retro-Cre into the BLA to minimize viral spread outside of the BLA, and the rest were performed after injection of 0.3–0.4 μ l of rAAV2-retro-Cre into the BLA. These are rather conservative amounts and are similar with previous studies targeting the BLA with stereotactic AAV injections.

mPFC anatomy: anatomy is central to both the study, and its findings, and great progress has been made in anatomical mapping of the mouse brain over the past decade. We find the anatomical delineations and nomenclature in the current study unclear. An, to us, unknown atlas has been used that does not appear to adhere to the most commonly used digital atlas (created by the Allen Institute for Brain Science). As the definition of the PFC varies among researchers, it is of importance that cell locations can be understood/re-mapped by colleagues in the field. We find the study would be stronger if it was better conveyed how the PFC subregions are delineated anatomically, i.e. if the reader could understand e.g. what part of the tissue the authors denote IL, DP, cingulate. This would help understanding and personal interpretation of the cell locations. There is only a quite small single image (Fig 1B; is this adhering to <https://kimlab.io/brain-map/atlas/>). We also recommend finding space in the main text to at least briefly state what atlas was used, why, and how.

This is a good point. As the reviewer correctly concluded, the atlas we used is the one shown in <https://kimlab.io/brain-map/atlas/>. This reference was added to the manuscript in the places where it was used (Chon... Kim Nat Comm 2019). We now mention specifically which atlas was used at each relevant place and figure in the manuscript. An explanation for the choices was added both to the main text and to the relevant Methods section that describes these choices (“*Inferring the brain-reference anatomical positions of the probed cells*”), and this section was further detailed for disambiguation. Generally, the aforementioned atlas was used for all anatomical analyses, as it integrates the two most commonly used atlases (Allen Institute and Franklin & Paxinos) and has an accessible interface for image analysis. For presentations in figures, we used the

Franklin & Paxinos, 2008 atlas for its outline presentation that is suitable for overlay on microscopy images.

Also related to anatomy:

DP - It is unclear why the DP is included in the study, and, thus, here included in the mPFC. This needs to be addressed/justified/discussed, including in relation to the BLA and topics addressed in the Discussion.

We appreciate the comment, but we also took into account the lack of sharp anatomical boundaries between regions in the mPFC. The borders drawn in atlases are somewhat arbitrary, and it is highly likely that similar functions are performed by BLA-projecting neurons in the ventral IL and dorsal DP. To address this comment, however, we repeated the analyses from Figure 3, Figure 4A–F, Figure S5A–C, Figure S6 (now appears as Figure S7), and Figure S7 (now appears as Figure S8) – but this time including only cells in the IL (and not in the DP). The main effects (and lack of effects) persisted, except for the significance level of the weighted input along the mediolateral axis for mPFC-BLA cells (Figure 3E, left), which reduced to $p = 0.06$. The conclusions from the data are unchanged after removing the DP. This may be due to the small number of DP connectivity maps in our dataset compared with the number of IL maps, as appears in Table 2 of the manuscript: 69 IL maps and only 6 DP maps. Apart from this, we justify the inclusion of the DP in our dataset and analyses based on the facts that the DP contains cells projecting to the BLA (Hintiryan... Dong, Nat Comm 2021; and our manuscript), and that some studies include it within the ventromedial PFC region without subdivisions and show that it is implicated in fear extinction which is typically attributed to the IL (for example, Bukalo... Holmes Sci Adv 2015). We believe that these findings suggest that both IL-BLA cells and DP-BLA cells are involved in similar learning-related cognitive processes.

Division of prefrontal layers - we find it unclear how the layers were defined/identified, and why (and how) the authors separate layers 2 and 3? Furthermore, why are layers 5 and 6 pooled (e.g. Figure 3D); how can this be justified? The laminations applied goes against some conventions in the PFC field (and atlases), and particularly, it is known that L5 and L6 hold differential input and output patterns (e.g. Harris and Shepherd, 2015). In relation to this: in Fig 3D it is shown that for the mPFC-BLA-projecting subnetwork (red), L5/6 receives higher weighted input than L2. However, in Fig S5D it appears that L3 (not reported in Fig 3D) receives similar input as L5/6(?) We find that the partly unconventional handling of the layers together with the lack of L3 data in parts of analysis (Fig 3D) cause some confusion. Is it possible to make Fig 3D and S5D understandable together? The laminar aspect is important as a central claim is that 'the weighted output from mPFC-BLA cells onto other mPFC-BLA cells was stronger in deeper layers than in superficial ones'.

We believe that complete division of the data into layers allows for most information to be extracted from the data, and that pooling can sometimes obscure interesting effects. For this reason, we divided the mPFC into layers in places where we could do so based on the data. For layers 2 and 3, we saw a pattern of cell density that resembles the known trajectory of cell density in the PFC (as in Van De Werd... Uylings, Brain Struct

Funct 2010). This subdivision was consistent with DeNardo... Luo, Nat Neurosci 2015 (Supplementary Figure 6 therein). As for layers 5 and 6, we agree with the reviewer that subdivision into these two layers would have been informative. Unfortunately, we had no molecular markers that would allow us to distinguish between these layers and therefore pooled them together. Due to this incomplete division into layers, we included in the manuscript analyses that are blind to layer but only consider mediolateral position, such as in Figure 3E and Figure 4G–I.

Regarding the handling of L3 in Figure 3D and Figure S5D: the relationship between these two figures may be more accessible and understandable if we present L3 data in Figure 3D (where it is now lacking since we have no cells of the random class in L3). We are attaching here the plot of Figure 3D after inclusion of L3 (Rebuttal Figure 2), and we have replaced Figure 3D with this new version to facilitate clarity.

Rebuttal Figure 2. Weighted input by connection type and layer, including L3.

Furthermore, in Fig S5D it appears that non-mPFC-BLA projecting and random mPFC cells in layer 5/6 (3rd column, blue and black) do not receive any input from layer 2/3, which would be surprising considering assumed canonical cortical circuit models. This finding also needs to be incorporated into any conclusions.

This result is now explicitly incorporated in the Results section of the manuscript.

We find it unclear if the regression in Fig 4H was performed on data from all 'pairs' of cells without regard to their class (mPFC-BLA - mPFC-BLA; mPFC-BLA - Non-mPFC-BLA etc.). It would be of interest to understand if the regression coefficients differ between the classes (and if no differences are found, how can then the demonstrated differences in connectivity between different mPFC subnetworks be understood?)

The regression was performed when two of the predictors were the connection type (one was mPFC-BLA and the other was non-mPFC-BLA, such that 0 for both indicated random mPFC). This gives explicit weight to the connection type and can account for interactions between it and the other predictors, and therefore is more informative than running the regression separately for each connection type. Indeed, we found that

mPFC-BLA to mPFC-BLA connection type had a significant positive coefficient indicating its weight in accounting for variability in connection probability.

The majority of patched cells are located in IL (75% (69/92)). It would be of interest to know, if the authors hold the data, whether the connectivity between prefrontal subregions differs (e.g. PL vs. IL) or if the findings in IL can be generalized to other (also not prefrontal?) regions.

We have repeated the analyses in Figure 3 using only the maps obtained from the PL cortex. Due to the small number of cells, conclusions were hard to draw for most analyses. Below is Table 2, where only the dorsal mPFC (PL and Cg) maps are mentioned (Rebuttal Table 1).

Map type	Recorded (postsynaptic) cell		Stimulated cells			
	Region	Layer	# total	# spiking	# included *	% spiking
mPFC-BLA to mPFC-BLA	PL	5/6	245	215	112	87.8
	PL	5/6	169	72	70	42.6
	PL	5/6	79	67	65	84.8
	PL	3	63	21	20	33.3
	PL	2	89	49	48	55.1
	PL	5/6	41	21	21	51.2
	PL	3	117	97	95	82.9
	PL	2	77	61	61	79.2
	PL	3	93	56	54	60.2
mPFC-BLA to non-mPFC-BLA	Cg	5/6	118	88	88	74.6
	PL	3	92	62	62	67.4
Random to random	Cg	5/6	91	78	78	85.7
	PL	5/6	109	63	63	57.8
	PL	5/6	142	129	129	90.8
	PL	5/6	50	43	42	86.0
	PL	5/6	126	115	114	91.3
	Cg	2	69	46	46	66.7

Rebuttal Table 1: Cell numbers by connection type in the dorsal mPFC.

Despite the small cell numbers, we could find an indication that the laminar connectivity pattern observed in IL-BLA cells (Figure 3D) could hold for PL-BLA cells as well (Rebuttal Figure 3A). We also found that the adaptation index in PL-BLA cells tends to be higher than in random PL cells (Rebuttal Figure 3B), consistent with our IL data (Figure 4E). We could not, however, find directional input patterns in the PL dataset (Rebuttal Figure 3C).

Rebuttal Figure 3. Connectivity properties and adaptation index in the dorsal mPFC. A, Weighted input by layer and connection type. B, Adaptation index by cell type. C, Weighted input directionality.

We note, however, that the regression analysis in Figure 4 was performed on the entire mPFC connectivity dataset, including the PL and Cg. We found that the dorsoventral anatomical position of the presynaptic cell accounted for some of the connectivity, whereby ventral position was associated with a reduction in connection probability. This could indicate a difference in overall connection rates between the dorsal and ventral mPFC subregions.

Minor comments:

The colors used in Fig S1B (IL, low/high titer) are difficult to distinguish, particularly for the circles.

Agreed. We changed the contrast and the stroke.

Figure 2G shows a 'representative synaptic connectivity map' with a patch recorded cell in DP. As the majority of patched cells (69/92) were recorded in IL and only 6/92 in DP, why was that example chosen? See also comment about DP above.

We replaced the DP map with an IL map in Figure 2G.

The sentence 'Moreover, the intrinsic properties of the postsynaptic mPFC cell and anatomical position of both cells jointly account for...' in the abstract is difficult to follow (if one has not already read the article) - postsynaptic to what, and what two cells are meant ('both cells')? Also at other places it is at times hard to know what cell(s) is meant - perhaps consider to avoid only saying cell, and instead consistently state post-/presynaptic, mPFC-BLA cell etc...

Thank you for pointing this out. We rephrased the Abstract to make it clearer.

Fig 3E, F: Can this data be translated into layers/mPFC subregions? It would allow interpretation of possible differences in weighted input between subregions.

We show the same data for the PL in our response to the reviewer's final major comment above. However, due to the small number of maps in the dorsal mPFC region, we did not include this separate analysis in the manuscript.

The abstract ends with 'Our findings demonstrate a functional segregation of mPFC excitatory neuron subnetworks, and reveal the factors determining connectivity in the mPFC'. This appears a bit of an overstatement - the study is purely ex vivo and functional synapses are examined rather than the function of segregated subnetworks. Also, only a single specific subnetwork is investigated.

We rephrased the Abstract.

Fig S5D: while a summary figure is great for an overview, the array of scales on the y-axes makes it very hard to actually compare the connections - could perhaps fewer scales be used?

We understand the reviewer's point, and have debated about this point. There is a tradeoff here between matching scales to allow comparisons, and the ability to visualize the data in each plot more clearly. We now generated the same figure with a fixed y-scale across all plots (Rebuttal Figure 4) and placed it instead of the previous one.

Rebuttal Figure 4. Cross-layer connectivity with constant y-scale.

Reviewer #3 (Remarks to the Author):

This study explores the connectivity of neurons in the prefrontal cortex using a high-throughput single cell optogenetic mapping approach. They report a number of interesting properties related to local circuit connectivity, focusing on mPFC neurons that project to the BLA and how their connectivity relates to other cells in the network. The manuscript is well written, the technological approach is rigorous and the topic investigated is an interesting one that will be of relevance to the field. However, my main objections are the way that the data are presented and some issues around the way cells are sampled in different groups. Without more detailed analysis it is difficult to see how much of their findings depend on grouping heterogeneous populations into a single sample.

Methods

The authors adjusted laser power across depth based on theoretical considerations but was this effective, was p-spike similar across this depth based on their fluorescence data?

To answer this, we measured the correlation between the slope of the raw GCaMP6s fluorescence trace across a train of spiral scans (as used to infer spiking) and the depth of the scanned cell in the slice, for all stimulated cells in our dataset (Rebuttal Figure 5). We found a small negative correlation, suggesting that depth within the slice either impedes induction of spikes or impedes detection of spiking. It is therefore likely that this negative correlation would have been stronger without the power compensation, making spike validation difficult. We cannot tell whether this negative correlation stems from less spikes in deeper cells (due to weaker excitation light) or from reduced detectability of spikes (due to reduced excitation and emission light from the GCaMP6s).

Rebuttal Figure 5. Depth of stimulated cells vs. their GCaMP6s response, for all stimulated cells in the dataset.

Their approach is highly dependent on the reliability of their spike and EPSC detection algorithms. If they run their connection detection model for traces where a presynaptic spike was not detected, what percentage of traces yield an input? This would be a useful control to confirm both their spike detection capability with GCaMP imaging and also their EPSC detection model.

The proportion of cells that our model determined to be connected is similar between the pool of responsive/spiking cells (whose spiking in response to stimulation was validated using GCaMP data) and the non-spiking cells. For this calculation, we considered 9566 stimulated cells out of the total 10817 stimulated cells in our dataset. Out of 1855 cells that were determined not to have spiked based on GCaMP data, the model determined 64 to be connected (3.45%). Out of 7711 cells that were determined to have spiked based on GCaMP data, the model determined 302 to be connected (3.92%; $\chi^2 = 0.76$, $p = 0.38$). This could be a result of the possibility that during stimulations that did not result in sufficient GCaMP signal, one or more off-target cells were stimulated and spiked, since the spiking spatial specificity is wider than that of the GCaMP specificity (Figure S4A,B). This would mean that our spike validation acts to filter out the cases where specifically the targeted cell did not spike, and possibly excludes false-positive connections from our dataset.

Was there a difference in connectivity based on internal solution used, did using Cesium internal for a subset of recordings influence their ability to detect inputs?

We recorded 7 connectivity maps using a Cs-based internal solution (one of these maps was not used in the manuscript since it could not be properly aligned anatomically).

Pooling all maps where each internal solution type was used, the connection rates were 0.026 ± 0.003 for K and 0.011 ± 0.007 for Cs. This difference was not significant due to the large variance and small number of Cs maps ($p = 0.11$ for χ^2 test when pooling all map proportions and $p = 0.20$ for t-test when taking individual map proportions).

Importantly, the Cs maps were inconsistent in their connection type (5 were mPFC-BLA to mPFC-BLA connections and 2 were random mPFC connections) and in their mPFC subregion and layer, so we believe that no informed conclusion can be drawn from this comparison. Notably, we used K-based solution for the majority of experiments in order to extract the electrophysiological properties of the recorded cell – both passive and active properties – which would not be possible using Cs-based solution due to blockage of K conductances.

Results

L5 and L6 have very different connectivity. Even within L5 descending input from superficial layers drops off significantly as a function of depth (for example Anderson et al., 2010 Nat Neurosci.). Do they observe this effect in mPFC and was there a difference in the M-L depth L5/6 cells across groups?

Within L5/6, the great majority of cells were within $500 \mu\text{m}$ distance from the midline. We show below (Rebuttal Figure 6) the connection rates and strength as function of ML position for all cells in all groups. We did not find a trend such as the one the reviewer is referring to within L5/6. The color code is as the rest of the figures – red for mPFC-BLA to mPFC-BLA, blue for mPFC-BLA to non-mPFC-BLA, and black for random.

Rebuttal Figure 6. Connection probability (top), connection strength (middle), and number of recorded cells as function of mediolateral position, for all connection types.

They should report p-Conn and EPSC amplitude across layers, not just for the sum input. This would help people trying to model the mPFC understand the synaptic properties within and across layers.

We are attaching below (Rebuttal Figure 7 and Rebuttal Figure 8) similar data as in Figure S5D, but for connection probability and amplitude. We will make the entire dataset from these experiments, as well as our analysis code, freely available upon request.

Rebuttal Figure 7. Across-layer connection probabilities.

Rebuttal Figure 8. Across-layer connection strengths.

The M-L analysis is kind of meaningless without knowing the location of the postsynaptic cell. A cell at the border of L1/2 will always have lateral input, while a L6 cell will always have input that is medial, the question is which layers provide those inputs. Given we

know that the cortex is organized into precise layers and that connectivity is tightly correlated to the laminar location of both pre and postsynaptic neurons, choosing to use an arbitrary M/L axis seems somewhat meaningless. This is all the more problematic when you look at the sampling of layers in each population. The BLA↔BLA sample is biased towards L5/6 (L5/6 = 50% of all recorded cells), the BLA↔non-BLA sample is biased towards L2/3 (L5/6 = 21% of all cells) while random to random is also slightly L2/3 biased (44% in L5/6). Even ignoring the fact that we know connectivity is driven by layer location, without equal sampling across the depths in each group this type of M-L comparison is almost impossible to interpret.

This ML-axis analysis was meant to complement the laminar analysis provided in Figure 3D and Figure S5D. While the effect of layer location on connectivity is indeed well-characterized in granular cortical regions, much less is known about it in the mPFC, and the laminar division in the mPFC is also not as clear (see DeNardo et al., Nat Neurosci 18, 2015, and Supplementary Figure 6b therein for variance in layer borders of the mPFC). We therefore sought to analyze the effect of ML position independent of categorical laminar position, and test the assumptions of the effect of layer position on connectivity specifically in the mPFC. Looking at the ML position distribution, although mPFC-BLA cell position does not distribute uniformly (as seen in Rebuttal Figure 6), it does span the entire sampled range (the bias for L5/6 is due to its relative thickness). When we perform the ML analysis only on mPFC-BLA cells where at least 20% of the presynaptic cells are on either side (that is, no more than 80% of the stimulated cells can be concentrated on one anatomical side of the recorded cell, in order to avoid having cells that receive input from one side only), the results we presented in Figure 3E persist. Regarding the other cell populations, we agree with the reviewer that non-mPFC-BLA cells are underrepresented in deeper regions and random mPFC cells are underrepresented at intermediate depths, making the results hard to interpret. This could also account for the fact that we did not find a reversal in the connectivity pattern between mPFC-BLA cells and non-mPFC-BLA cells in this analysis, as we found in the laminar analysis in Figure 3D.

Figure 3B, do these maps represent the maps for all the recorded cells aligned to the location of the soma? What if you align them to the depth/layer they were recorded? See above, this is hard to interpret without knowing where the cells are located.

We show below the connectivity maps when they are aligned to the recorded cell position, as presented in Figure 3B (“Cell reference”, left plots), and aside to this we show the same maps when they are overlaid and presented in anatomical space (“Brain reference”, right plots), for each connection type (Rebuttal Figure 9, Rebuttal Figure 10, and Rebuttal Figure 11). The connection strength scale and the legend are identical for all plots.

Rebuttal Figure 9. mPFC-BLA to mPFC-BLA connections.

Rebuttal Figure 10. mPFC-BLA to non-mPFC-BLA connections.

Rebuttal Figure 11. Random mPFC to random mPFC connections.

Why was an arbitrary 300μm radius chosen to analyze connectivity? For cells deep in L5 or L6 I presume this would mean that you aren't sampling most of the superficial layer input (where many BLA cells are located). In the ML (intralaminar) direction translaminar inputs can span many 100s of microns, see point above regarding drop off of input from superficial to deep layers based on depth. More detailed analysis is needed to see if this approach is actually warranted or if they are artificially introducing connectivity patterns to their data by only subsampling the most proximal inputs.

We repeated the analyses that used the 300 μm radius restriction: Figure 3C (only the probability of connection was restricted by distance, not the connection strength), Figure 3D, Figure 4F, and Figure S5A (also here the connection strength was not restricted by distance). To see if the effects were specific to the 300 μm radius, we used a radius of 150 μm and also no distance restriction (Rebuttal Table 2). All the effects we observed for the 300 μm radius persisted for the other distance restrictions.

Rebuttal Table 2. Connectivity properties by distance restriction from the recorded cell.

Radius (μm)	Effect		
	150	300	450 (max)
Figure 3C: conn strength vs P(conn)	No corr	No corr	No corr
Figure 3D: weighted input for layers and types	Conn type×layer interact	Conn type×layer interact	Conn type×layer interact

Figure 4F: weighted input vs Rin	Neg corr (weak with R=-0.26, p=0.03)	Neg corr (R=-0.3, p=0.009)	Neg corr (R=-0.3, p=0.009)
Figure S5A: conn props for map types	No diff	No diff	No diff

Discussion

The authors make a valid point about how most historical paired recording studies are restricted to the superficial sublayer of the slice (first 50-100 μ m) and perform some helpful analysis related to this point that is in the supplemental discussion. It is a shame that is not more clearly signposted in the main text.

Thank you for this suggestion. We now mention this point more explicitly in the Discussion.

Discussion of presynaptic release should mention that they use 2mM Ca²⁺ which is possibly high compared to in vivo levels (estimated to be 1mM, See Seeman et al 2018 eLife) and could bias towards depressing synapses.

We agree with this observation, and we now mention this possibility in the Discussion.

Reviewer #4 (Remarks to the Author):

The authors present a comprehensively validated optical method to determine the presynaptic connectivity of electrophysiologically recorded neurons in mouse brain slices. They use this method to determine key factors regarding the fine-scale circuit structure of mPFC. This method builds on previous approaches by adding the ability to know the projection target of the recorded neuron and its presynaptic partners while at the same time maintaining sufficiently low expression to avoid off-target activation. Given the low likelihood of connection in the particular circuit under study here, a method such as this is effectively critical to find the specificity the authors show is extant in these circuits. Fig S3 in particular is a crucial, necessary, and difficult control that has been well performed. The careful & rigorous analysis of the cells that could have been inaccurately photostimulated -- which turns out to be low given the combined sparse expression and spatial resolution of the method -- is particularly laudable. This manuscript is suitable for immediate publication in Nature Communications following some minor revisions. No further experiments are required.

We thank the reviewer for these supportive comments.

Major comments:

GCaMP underreports spiking as previously shown and also apparent in the authors

Figure S3. Is it possible the low overall probability of connection in S5 (even lower than the normally low probability of cortical connections amongst excitatory cells) is due to this effect? Probably not, given their low probability of connection is very similar to that found with paired recordings in ref 70. However I found one of the authors statements in the text, " 117.6 ± 5.0 cells (of which 95.4 ± 5.1 are confirmed to respond to stimulation via the GCaMP6s signal) ...", confusing. Did they not ALWAYS use confirmation of the presynaptic neuron's spiking to confirm the presynaptic connectivity? This does not appear to be the case from the methods describing the connectivity assessment, which are wonderfully detailed (but I may have missed it). Or did the authors use this in the case of the manual assessment, which was used to validate the model-based assessments, perhaps? One more sentence in the methods to clearly disambiguate this situation would be appreciated, especially given this sentence found elsewhere: "Stimulated (candidate presynaptic) cells were excluded from analysis if they met at least one of these conditions: Lower 95% confidence bound for the slope of the raw GCaMP6s fluorescence trace across the spiral train was negative; ... "

We have indeed always confirmed presynaptic spiking and excluded the non-spiking cells from all analyses. The only case where we did use the non-spiking cells was the analysis of the performance of the connectivity model (Figure 2E,F), but otherwise, these cells were discarded. We have changed the phrasing in the Results and in the Discussion to clarify these points, and we added a sentence in the Methods (under exclusion criteria) to explicitly mention the only case where we used the non-spiking cells.

Regarding the spike reporting by GCaMP: We think that the overall low probability of connection is not due to GCaMP sensitivity in spike detection, since underreporting of spiking is probably equally likely for connected stimulated cells and for non-connected stimulated cells, such that the underreporting just scales down the presynaptic cell sampling without biasing it toward non-connected cells.

If the authors might be able to quantify the co-expression of stCoChR and GCaMP6s in the mPFC->BLA cells, this would be helpful. It is not absolutely critical, but given they are expressing these proteins from two different viruses, it could be useful to know. Analysis of whatever images of expression from their experiments they already would be helpful if available.

Co-expression is indeed a difficult issue in such experiments. To maximize co-expression of the two vectors, we always use serotype- and titer-matched AAVs for both transgenes, which leads in our hands to high levels of co-expression. To address this comment, we quantified the co-expression of stCoChR and GCaMP6s. Since both stCoChR and GCaMP6s are labeled with a red fluorophore and the basal GCaMP6s fluorescence is typically too low to visualize in fixed slices, we performed anti-GFP immunostaining to enhance the GCaMP6s signal and measure its colocalization with stCoChR. This data now appears in Figure S2B.

Minor

10, 15, and 20 um spiral legends (in Figs 1i and S2c) are mildly confusing.

Agreed. We changed the legend in Figure S2C (now appearing as Figure S3C) to resemble that in Figure 1I, so that it is clearer that the legend applies to all plots.

Diagram at the bottom right of Fig S3e is not described.

Thank you. We added a description to the figure legend.

S5 is presented before S4.

Thank you for bringing this to our attention. We have switched the order of the figures.

Tukey's may be informative for Fig 3d.

We agree that post hoc comparisons are informative for this analysis, but we also think that the important measure for this analysis is the interaction effect between layer and connection type, as we reported, because it points to different trends of connectivity across layers depending on the connection type. Individual comparisons would not fully capture this effect. For completeness, we report below the individual comparisons. Since this figure discusses the differences in output patterns from mPFC-BLA cells depending on the postsynaptic cell type, we repeated the analysis including only connections from mPFC-BLA cells onto either mPFC-BLA or non-mPFC-BLA cells (omitting the random connections), and included layer 3 in the analysis (which was not sampled for the random mPFC cell class; Rebuttal Figure 12 and Rebuttal Table 3).

Rebuttal Figure 12. Weighted input by layer and connection type, including L3 and omitting random-connection type.

Test	Effect/comparison	p value
------	-------------------	---------

Two-way ANOVA	Connection type main effect	0.899
	Layer main effect	0.399
	Connection type×layer interaction	0.004
Post hoc comparisons with Tukey's HSD	mPFC-BLA L2 vs non-mPFC-BLA L2	0.062
	mPFC-BLA L2 vs mPFC-BLA L3	0.312
	mPFC-BLA L2 vs non-mPFC-BLA L3	0.973
	mPFC-BLA L2 vs mPFC-BLA L5/6	0.471
	mPFC-BLA L2 vs non-mPFC-BLA L5/6	0.999
	Non-mPFC-BLA L2 vs mPFC-BLA L3	0.943
	Non-mPFC-BLA L2 vs non-mPFC-BLA L3	0.334
	Non-mPFC-BLA L2 vs mPFC-BLA L5/6	0.599
	Non-mPFC-BLA L2 vs non-mPFC-BLA L5/6	0.112
	mPFC-BLA L3 vs non-mPFC-BLA L3	0.819
	mPFC-BLA L3 vs mPFC-BLA L5/6	0.989
	mPFC-BLA L3 vs non-mPFC-BLA L5/6	0.376
	Non-mPFC-BLA L3 vs mPFC-BLA L5/6	0.962
	Non-mPFC-BLA L3 vs non-mPFC-BLA L5/6	0.931
mPFC-BLA L5/6 vs non-mPFC-BLA L5/6	0.545	

Rebuttal Table 3. Full statistics and post hoc comparisons for Rebuttal Figure 12.

Some experiments used K-based internal solution for patching while others used Cs-based. Perhaps the experiments that used Cs-based internal could not be used for analyses requiring the firing pattern of the patched cell, but they were used for all other analyses? A sentence in the methods clarifying this situation could be helpful here.

This is correct, we did not use the Cs-based recordings for spiking characterization (six experiments used Cs solution). This is now explicitly stated both under the electrophysiological properties section and the regression model section (where cell pairs without full electrophysiological characterization were not included) of the Methods.

The authors may want to consider citing other papers using high precision circuit mapping such as Nikolenko et al (<https://www.nature.com/articles/nmeth1105>), Packer et al 2012 (<https://www.nature.com/articles/nmeth.2249>), Naka et al 2019 (<https://elifesciences.org/articles/43696>), and the very recent Hage et al 2022 (<https://elifesciences.org/articles/71103>).

We thank the reviewer for these suggestions. We decided to cite several additional papers that used two-photon optogenetics to perform connectivity mapping, including two of the studies mentioned above. We believe that the two-photon uncaging study is less relevant here, as is the Naka et al. paper which used single-photon DMD-based stimulation. However, if the reviewer believes that these should still be cited, we would be happy to include them as well.

REVIEWER COMMENTS

Reviewer #1 (Remarks to the Author):

I have few suggestions to improve the clarity of the manuscript:

Page 5: the conditions chosen for the experiment: spiral duration 7.2ms, stimulation frequency 10Hz, light power 10 mW, are not shown in the Figure 1I or Figure 2SC, if the authors have these data, it would be better to include them in the figures.

The reviewer is correct in that the combination of parameters used in the connectivity mapping experiments – namely duration of 7.2 ms, frequency of 10 Hz, and power of 10 mW – does not appear in the dataset described in Figure 1I and Figure S2C (now changed to Figure S3C). We do not have similar calibration data for this exact parameter combination, since we performed calibration for each parameter separately and chose the optimal ones after their individual calibration.

OK

Page 6: authors characterize the axial resolution looking at the spiking probability and DF/F signal; It would be important here to also give the optical axial resolution, how extended is the spot used for the spiral scanning along the axial direction? Also could they give an explanation why the spatial specificity curve for GCaMP-DF/F is so much narrower than the spike probability? Is this because their imaging conditions are not sensitive enough to detect a single spike?

To address this request, we have included the optical resolution of the scanning as PSF measurement in Figure S4. The reason that the GCaMP6s spatial specificity curve is narrower than that of stCoChR-induced spiking is the very high light sensitivity of stCoChR as compared with GCaMP6s, resulting in higher light power needed for a resolvable GCaMP6s signal.

Thanks for this explanation which seems to confirm our understanding: the power used for the GcaMP6s spatial specificity curve is too low to permit to reliable resolve a single spike. In this case we should suggest specifying the working power also in the caption and add a comment on the reason for this difference of the axial resolution also in the text (e.g. at the end of the first paragraph at pag. 5).

Figure 1: please complement the figure with some representative images (or a stack) of the recorded slices to show expression, co-expression, sparsity of the labelling etc. In the current manuscript there are no images of the actual slices that were used for the experiments.

Thank you for the suggestion, which we completely agree with. We added a representative z-projection image to Figure S2C. It shows the density of labeled mPFCBLA cells in an imaged volume used for mapping connectivity. Regarding co-expression of stCoChR and GCaMP6s, since both are labeled with a red fluorophore and the basal GCaMP6s fluorescence (without activity) is too low, we performed anti-

GFP staining to enhance the GCaMP6s signal. Images and quantification of stCoChR and GCaMP6s appear in Figure S2B. Notably, cells without co-expression would be discarded in analysis due to lack of GCaMP6s signal, either because of lack of stCoChR (and therefore spiking) or lack of GCaMP6s.

OK

From Figure 1E and in Fig. S2C, it seems that stimulation often gives rise to multiple spikes. Could the author comment whether this has an effect on the amplitude of the post-synaptic responses and how this is taken into account in the measurement of the synaptic strength?

A brief explanation and a reference to the relevant detailed Methods section were added to the main text. It now states on page 7: "Finally, we calculated the strength of synaptic connection at each stimulation as the weighted average of EPSCs within a time window following stimulation (see Methods), thus accommodating for jitter in synaptic latency and the possibility for multiple evoked EPSCs (Figure S3C)." The weighted average used to calculate synaptic strength was based on a normal fit of the EPSC distribution, as described in Methods under "Measurement of synaptic connection strength".

OK

Figure 1F, G show the DF/F and GCaMP fluorescence curve as a function of spiral number, respectively. This characterization is important as the amplitude and slope of the GCaMPs signal are used to select the post-synaptic cells for the connectivity experiments. Few points should be then explained more clearly. Specifically, Fig1F show largely varying DF/F values, from one cell to another, in one cell the signal is comparable to the recordings made in presence of TTX where one would expect no spike activity. Same in Fig.1G, multiple cells in presence of TTX show responses comparable, if not higher, to the aCSF case. The authors should discuss these curves and the diversity in the responses more extensively. For example, are all the curves taken at the same power? Or is the GcaMP imaging power chosen on the base of the opsin expression levels (higher power for low opsins expressing cells) so that the GCaMP fluorescence signal varied accordingly? Are their imaging conditions sufficient to detect single APs?

The variation in GCaMP6s responses between cells most likely stems from variability in expression levels of both GCaMP6s and stCoChR, resulting in varying spiking efficiency as well as varying GCaMP6s signals for the same spiking behavior. Another factor that could contribute to this variation is the shape of the cell relative to the spiral pattern scanned on it. The light power was consistent across cells and was not adjusted to expression level (but was adjusted to depth in the tissue as explained in the Methods in order to maintain constant power between cells). stCoChR evokes very strong depolarizations (as characterized in Forli et al., eLife 2021), such that even in the presence of TTX, photostimulation can result in calcium influx through voltage-gated calcium channels, which in turn can increase the GCaMP6s signal without the cells spiking. We believe that this is the reason that some of the cells show strong GCaMP6s signal increase in presence of TTX. We therefore did not use the GCaMP6s signal in presence of TTX to determine a threshold for spiking, as the strong depolarization confounds the interpretation of

this signal. Finally, our imaging conditions are not sufficient to detect single spikes but rather tell us whether spikes occurred during a stimulation train.

We appreciate the explanation of the authors and agree. The authors should include these considerations into the manuscript. Especially the possibility of calcium influx without action potential firing might be something not every reader is aware of.

The authors mention that, even in presence of TTX (pag 30), GCaMP fluorescence can still accumulate “supposedly due to the strong light-induced depolarization”. Can the authors better clarify this statement and the nature of the effect? Also can they specify if cells that give these level of GCaMP transient also in presence of TTX, would pass the criterion to validate activation?

Please see the answer to the previous comment. Additionally, a depolarization amplitude that increases GCaMP6 signals in the presence of TTX would undoubtedly trigger spiking in a natural setting with no TTX. Therefore, it is highly likely that cells showing large increase in GcaMP6s signal with TTX would spike without TTX and pass the inclusion criteria.

Ok

Page 6: For the automated detection of potential pre-synaptic cells the authors describe to use mScarlet and dTomato fluorescence, co-expressed with stCoChR and GcaMP respectively. I expect that the cells used for the connectivity mapping would all (most) express both the CoChR and GcaMP, however in the description of the automated detection it is not clear if this was based on the detection of co-expressing cells or if some of the detected cells only express the CoChR and no GcamP or vice versa? Meaning the 209/316 cells detected in the volume (Page 6) correspond to cells expressing one of the two or co-expressing both? What did the authors mean with observing the detections?

In our experiments, we cannot distinguish between stCoChR expression and GCaMP6s expression due to the significant overlap between the spectra of dTomato and mScarlet. We rely on the fact that targeted cells which express only one of the two will not show GCaMP6s increase, and these cells will therefore be discarded during analysis.

Again, we appreciate the explanation of the authors, but it is really important that these aspects are incorporated in the text; as of now it is difficult to understand this from the manuscript.

This also relates to my next point:

Does the cross-talk with GCaMP and mScarlet affect the cell mapping? Precisely, coexpressing cells should show GCaMP and mScarlet fluorescence in the cytosol, and dTomato fluorescence in the nucleus. Detecting cytosolic mScarlet informs about the presence of CoChR, but in case of cross talk, how do they distinguish that from GCaMP, which is also in the cytosol? This is important if they screen for co-expression directly, which is not clear (see previous comment). Maybe they just map the volume for any expression, stimulate all cells and discard the ones that are not responding?

This is precisely what we did – scan all cells that express a red fluorophore (without distinction between nuclear and cytosolic expression) and include only responsive ones.

Ok

Fig. 2D: The authors show the expected shape of the EPSC according to the model used for the automatic identification of connections. In the reported example, the onset of the response is perfectly aligned with the start of the stimulation, while in the experimental curve (Fig2C) the response is temporally delayed, could the author clarify this point and define what criterion is used to distinguish EPSC from direct photoactivation. Could the authors comment on that?

In all our data, we observed that direct photocurrents appear at ≤ 1.5 ms latency, minimal jitter, and consistent waveform with slow rise constant compared with EPSCs. This can be seen in Figure 2C. The seemingly zero-latency EPSC response in Figure 2D is due to the filtering effect of the KDE. Throughout our experimental dataset, EPSCs always appeared with at least few ms latencies.

The contribution of direct stimulation of the post-synaptic cell is not trivial (the rise time can depend on power/distance, expression level, illumination method etc.) and can be an important factor in connectivity mapping. We appreciate the explanation of the authors, which is very reasonable, but not easy to find in the manuscript. May the authors could add a sentence in the main text or at least a reference to their (very detailed) section in materials and methods, where they explain how they deal with the direct stimulation?

Discussion: In the discussion, the authors mention previous work achieving optogenetic based synaptic mapping saying that they are “restricted to measuring only unidirectional (in-degree) connectivity”. Could they better clarify this idea?

A clarification was added to the Discussion text. The intension is to say that only connections from the stimulated cells onto the recorded cell can be probed, and not in the reverse direction. Notably, after mapping connections from stimulated cells onto the recorded cell, another experiment can be conducted where one of the stimulated cells is now recorded and the previously recorded cell is now stimulated, but the recorded cell usually does not stay viable after a mapping experiment to consistently allow such an approach.

Discussion: in the discussion it would help to discuss (if any) the limitations of the presented technology and give an outlook on what could be improved (e.g. light delivering approach, detection, opsin expression and targeting....)

The main limitation of our approach lies in the jitter of spiking and synaptic responses which stems from the spiral stimulation method. This is inherent to spiral stimulation, but our analysis is still able to take the jitter into account using the modeling approach we describe in Fig. 2. Nevertheless, future work

could utilize more advanced holographic methods to improve spike time precision and reduce jitter. A relevant text was added to the Discussion.

Ok

Pag 28/29: the authors give a detailed description on how they discriminated among direct photocurrents and EPSC and confirm that the criteria that they used work by comparing the currents recorded in presence of glutamate-receptor blockers. It would help if they could here also quantify how many cells they typically excluded for experiments. Also, supposing that the presence of artifactual photocurrent decreases with distance, they should comment on how this could affect the overall estimation of connectivity ratio and spatial distribution of connections.

The exact numbers of cells that were excluded in each experiment can be found in Table 2: it details, for each experiment, the number of stimulated cells, of which the number of cells inferred to have spiked in response to stimulation, and the number of cells included in analysis, such that the difference between the number of spiking cells and the number of included cells reflects the number of cells that were excluded due to large evoked photocurrents (or noisy recording, in some cases).

To address the second point concerning the dependence of evoked/artefactual photocurrents on distance from the recorded/postsynaptic cell, we have explicitly calculated this dependence (Rebuttal Figure 1). We found that as the reviewer suggested, the density of cells that evoke large photocurrents (PC) in the recorded cell, and are therefore excluded from analysis, decreases with distance from the recorded cell, whereas the distance of cells that do not evoke large photocurrents in the recorded cell (no PC) distributes normally. This could introduce a bias in estimation of connection probabilities, since relatively more close cells are excluded than farther cells. However, the absolute numbers of excluded cells, as seen in the bottom histogram, is small. Moreover, Figure S5B shows that at short distances (and also long distances), the mPFC-BLA to mPFC-BLA connections – where we can find PC-evoking cells – are not sparser than the other connection types – where no PC-evoking cells are found. Therefore, although a bias is theoretically predicted, we did not find it in the relevant dataset.

Rebuttal Figure 1. Quantification of cells whose stimulation evokes direct photocurrent in the recorded cell by distance from the recorded cell. Top, fraction of cells; bottom, number of cells.

Pag 28: “were manually examined for synaptic connections by searching the stimulation aligned traces for reliable, low-jitter EPSC occurrences.” Authors should define here the meaning of low-jitter EPSC, as the above discussion they made to distinguish photoactivation from EPSC based on the fast responses of photoactivation could be otherwise confusing.

The use of “low-jitter” here comes to differentiate such cells from randomly distributed EPSCs, which can sometimes appear to occur near the time of stimulation, but are more noisy and have higher “jitter”. As can be seen in the third trace in Figure 2C, EPSCs can be distinguished from direct photocurrents by their rapid rise time as compared with photocurrent rise time, by their longer latency, and by the variance in their waveform (especially amplitude) between repetitions. We used these

characteristics to identify cells whose stimulation evokes photocurrent and cells whose stimulation evokes EPSCs 'riding' on top of a photocurrent.

Ok, the explanation is satisfactory. But please keep in mind that the slow rise of directly evoked currents that is observed may arise from the spiral stimulation and exclusion criteria used here may not be generalizable.

Figure S6: the authors discuss the possibility of short-term plasticity induced by repeated spiral stimulation, which gives rise to a greater EPSC amplitude for the first 3 or 4 stimulations. According to Figure 1E, it seems that the first 2/3 spirals can also generate double spikes which would probably also generate higher EPSC. Could the authors comment on how these two effects can be distinguished?

Our method does not allow complete control over the number of presynaptic spikes per stimulation, as shown in Figure S3C. Therefore, as the reviewer observed, calculation of short-term plasticity is inaccurate, and we mentioned this limitation in the Discussion. However, since this same limitation applies to all connection types that we have measured, we assume that differences in this plasticity estimation would reflect true differences between the connection types, and therefore chose to include this measurement in a Supplementary Figure. [As another reviewer pointed out, since these experiments are done in aCSF which has relatively high calcium concentration, the short-term plasticity features of synaptic connections are quite uniform and tend to be biased to strong adaptation, occluding any potential variation among populations].

Ok

The authors report EPSC down to around 5 pA. In methods they mention a threshold of 2.5 SD. What does this value correspond to practically? Are connectivity rate values they observe potentially affected by this precision? I suggest the authors better discuss their detection sensitivity at the post-synaptic site.

The value of 2.5 SD was added to the text under the relevant Methods section. As with any measurement technique, our ability to detect small synaptic events is limited by the sensitivity of our system. The 2.5 SD value represents the sensitivity of our system as the minimal EPSC amplitude which can be detected, and it is therefore the minimal strength of synaptic connection which we can detect. Strength of synaptic connections in the cortex is skewed, with many weak connections and few strong ones for each cell (for example, Cossell... Mrsic-Flogel, Nature 2015). Since our method might miss the weakest connections, we focused in our measurements on the weighted input that each cell receives, which represents the overall effect that the network has on the activity and excitability of the post-synaptic cell.

Ok

Minors

Ok

It would be useful to supplement figS1D and E with some actual images from the stacks show examples of cells detected manually and automatically.

Agreed. These images were added as Figure S2C.

Fig1C: I presume, based on the color code of the first column, that second and third column have been recorded with a 15 um spiral: it will help to give this information.

Assuming that the intension is to Figure S2C (now changed to Figure S3C) where there was a color code for the spiral size at the first column, we have changed the legend of this Figure to resemble that of Figure 1I which presents a similar dataset and where the spiral size color code appears on top of all columns.

Page 5: Throughout the characterization of the stimulation, it is not very clear if the 7.2 ms spirals are always two 3.6 ms spirals (same dwell time)?

The reviewer is correct, the dwell time is constant and the change in duration was achieved by concatenating spirals. A 7.2 stimulation is two 3.6 ms spirals (whereas one spiral is one round of in-and-out scan, as shown in Figure 1C, right image). A clarification was added to the Results text in page 5 where the surveying of scan parameters is described.

Page 55, Fig S7 A: Please include a legend to explain the three groups/colours or include it in the legend of this figure

A legend was added to the Figure. Please note that the figure numbering was changed and this figure now appears as Figure S8.

FigS1 Panels A, B and C,D,E are describing different experiments. The cell detection was performed (and characterized) on the slices used for connectivity recording (and not on the Ai9 mice?). It could be confusing for the reader to have this in the same figure.

We split Figure S1 into two separate figures and added illustrations to facilitate distinction between the different experiments presented in these figures.

Discussion: "Here we presented a large-scale implementation of such an approach, combining it with calcium-based readout of activity using a single laser source." Better probably to repeat here the FOV and the achieved number of recorded pairs.

Thank you for the suggestion. These details are now explicitly mentioned in this section of the Discussion: "Our semi-automated approach for cell detection and for sequential stimulation and calcium recording allowed us to probe the input from 95.4 ± 5.1 cells (mean \pm s.e.m), whose spiking in response

to stimulation was validated using the GCaMP6s signal, onto each recorded cell in three dimensions within a volume of $\sim 420 \times 420 \times 300 \mu\text{m}^3$.

Methods page 30, considering that the t_{decay} is used to define a scattering length it would probably better to call it this way and also to compare the found value (147.6 μm) with what is given in the literature.

We have changed the name of the constant to 'attenuation length' (since we believe it reflects a combination of scattering and absorption) and added a comparison to the reported literature.

Reviewer #2 (Remarks to the Author):

Reviewer #2 (Remarks to the Author):

The study by Printz et al. entitled "High-throughput mapping of functional synaptic connectivity in the prefrontal cortex" employs a novel and promising method combining whole-cell patch-clamp recordings with 2-photon microscopic imaging and simultaneous optogenetic stimulation of projection-defined neurons in acute prefrontal brain slices to map the functional connectivity of the local excitatory circuit.

The authors characterize several features defining the probability of synaptic connections between retrogradely labelled BLA-projecting prefrontal neurons, mPFC-BLA neurons, and non-mPFC-BLA neurons, and between random mPFC neurons, respectively. The authors show that the major determinants of local connectivity within mPFC are the distance between the cell bodies and the anatomical position of cells, especially along the mediolateral axis of the mPFC.

The methods of the study are well described and we positively note the great care and detail the authors deploy in characterising e.g. the size and duration of scan parameters, and the clear highlighting and discussion of limitations of the methodology.

While we are positive about the study, we find a need for clarifications and further discussion of particular issues, and a few experimental parameters needing to be addressed.

The viral strategy: while the calibration of expression of retrograde AAV vectors is well described, we fail to understand what exact viral strategy was actually used in experimental animals. We also find the images of BLA targeting suboptimal, as they don't allow evaluation of the specificity of the viral labelling. In line with this, it appears that the quantifications of labelling at the injection site focuses on the BLA specifically but say nothing about the degree of unspecific labelling in other amygdalar nuclei, or

elsewhere. This is a central point, as a big part of the study regards mPFC-BLA neurons specifically. We find that the specificity needs to be appropriately addressed and accounted for. Overall we would welcome more images, including of cell body labelling in the mPFC (e.g. lamination) as to understand what neurons are interrogated, and the spatial distribution of the subnetwork identified.

The specificity of mPFC-BLA cell labeling is indeed central to this study, and we value the reviewer's comments on this issue. To address this, we have included in Figure S2 confocal images of the BLA and the mPFC of wildtype mice injected with rAAV2-retro- Cre into the BLA and DIO-stCoChR+GCaMP6s into the mPFC. To enhance visibility of axonal projections labeled with stCoChR and GCaMP6s, we stained slices for GFP. We show that the densest axonal projections from labeled mPFC cells are inside the BLA. These are now shown as Figure S2A,B. We have further included a Supplementary Discussion section (subtitled "Specificity of mPFC-BLA cell labeling") detailing the known projections from the infralimbic cortex to the region surrounding the BLA, to conclude that the likelihood of unspecific labeling exists, but is low. Briefly, the majority of labeling outside of the BLA (as seen in Figure S1A) was in the striatal region dorsal to the BLA, presumably due to viral leak from the injection needle during its withdrawal. However, based on the extensive characterization of the mouse brain-wide BLA connectivity (Hintiryan... Dong, Nat Comm 2021) and on the Allen Institute connectivity database, projections from the infralimbic cortex into this striatal region are very sparse compared with its projections into the BLA. Moreover, retrograde and anterograde labeling from the BLA result in labeling patterns similar to those we show in Figure S1A, and to a laminar distribution similar to that shown in Figure S2B. These observations suggest that non-specific retrograde labeling in the mPFC is minor compared with BLA-projecting cell labeling. Finally, ~55% of our mPFC-BLA cell mapping experiments were performed after injection of 0.1 μ l of rAAV2-retro-Cre into the BLA to minimize viral spread outside of the BLA, and the rest were performed after injection of 0.3–0.4 μ l of rAAV2-retro-Cre into the BLA. These are rather conservative amounts and are similar with previous studies targeting the BLA with stereotactic AAV injections.

We appreciate the added information and images related to the specificity of the viral labelling. The rAAV + 2 AAVs viral strategy was performed in WT MICE (not mentioned in results now) and the calibration experiments in Ai9 mice - we recommend that the authors in the text make it clear that the Cre recombination gives very different fluorescent labelling in the 2 sets of experiments. Particularly, TdTomato fills the neurons, labelling axons from recombined neurons in the injection site, ie give rise to anterograde labelling although a rAVV was injected. At least that is how we interpret the authors' descriptions. It is currently quite difficult to understand the text in eg "Specificity of mPFC-BLA cell labeling" and reconcile the labelling results in the two sets of experiments. We also find it suboptimal that the calibration experiments are mentioned after the actual experiments that used the findings from the calibration experiments. These are minor things but sources of confusion and misinterpretations.

mPFC anatomy: anatomy is central to both the study, and its findings, and great progress has been made in anatomical mapping of the mouse brain over the past decade. We find the anatomical delineations and nomenclature in the current study unclear. An, to us, unknown atlas has been used that does not appear to adhere to the most commonly used digital atlas (created by the Allen Institute for Brain

Science). As the definition of the PFC varies among researchers, it is of importance that cell locations can be understood/re-mapped by colleagues in the field. We find the study would be stronger if it was better conveyed how the PFC subregions are delineated anatomically, i.e. if the reader could understand e.g. what part of the tissue the authors denote IL, DP, cingulate. This would help understanding and personal interpretation of the cell locations. There is only a quite small single image (Fig 1B; is this adhering to <https://kimlab.io/brain-map/atlas/>). We also recommend finding space in the main text to at least briefly state what atlas was used, why, and how.

This is a good point. As the reviewer correctly concluded, the atlas we used is the one shown in <https://kimlab.io/brain-map/atlas/>. This reference was added to the manuscript in the places where it was used (Chon... Kim Nat Comm 2019). We now mention specifically which atlas was used at each relevant place and figure in the manuscript. An explanation for the choices was added both to the main text and to the relevant Methods section that describes these choices (“Inferring the brain-reference anatomical positions of the probed cells”), and this section was further detailed for disambiguation. Generally, the aforementioned atlas was used for all anatomical analyses, as it integrates the two most commonly used atlases (Allen Institute and Franklin & Paxinos) and has an accessible interface for image analysis. For presentations in figures, we used the Franklin & Paxinos, 2008 atlas for its outline presentation that is suitable for overlay on microscopy images.

We understand the reply as that the position of cells and the delineation of the mPFC adhere to the <https://kimlab.io/brain-map/atlas>. However, the figures does not necessarily reflect the data as a different atlas was used. It is our view that the figures should adhere to the analysis, i.e., the same atlas should be used, particularly as the authors say below, borders in atlases are somewhat arbitrary drawn, ie the atlases differ. In line with this, we notice that the DP appears to be expanded in the DV plane in the <https://kimlab.io/brain-map/atlas> compared to the Allen v2. We cannot see any good reason why two different atlases should be used.

Also related to anatomy:

DP - It is unclear why the DP is included in the study, and, thus, here included in the mPFC. This needs to be addressed/justified/discussed, including in relation to the BLA and topics addressed in the Discussion.

We appreciate the comment, but we also took into account the lack of sharp anatomical boundaries between regions in the mPFC. The borders drawn in atlases are somewhat arbitrary, and it is highly likely that similar functions are performed by BLA-projecting neurons in the ventral IL and dorsal DP. To address this comment, however, we repeated the analyses from Figure 3, Figure 4A–F, Figure S5A–C, Figure S6 (now appears as Figure S7), and Figure S7 (now appears as Figure S8) – but this time including only cells in the IL (and not in the DP). The main effects (and lack of effects) persisted, except for the significance level of the weighted input along the mediolateral axis for mPFC-BLA cells (Figure 3E, left), which reduced to $p = 0.06$. The conclusions from the data are unchanged after removing the DP. This may be due to the small number of DP connectivity maps in our dataset compared with the number of IL maps, as appears in Table 2 of the manuscript: 69 IL maps and only 6 DP maps. Apart from this, we justify the inclusion of the DP in our dataset and analyses based on the facts that the DP contains cells

projecting to the BLA (Hintiryan... Dong, Nat Comm 2021; and our manuscript), and that some studies include it within the ventromedial PFC region without subdivisions and show that it is implicated in fear extinction which is typically attributed to the IL (for example, Bukalo... Holmes Sci Adv 2015). We believe that these findings suggest that both IL-BLA cells and DP-BLA cells are involved in similar learning-related cognitive processes.

This reply contains some handwaving. More important, this point has not been addressed in the revised manuscript, and we again suggest that the authors discuss the inclusion of DP in the analysis. We think the authors can agree that DP as a rule is not included in the mPFC and if the authors wish to do so, the scientific reasons should be clarified to the readers. Contrary to the regions traditionally included in the PFC, the DP does not have a L6b. Inclusion of DP will raise questions and without scientific reasoning its inclusion smells of p hacking given the result presented in the reply above. We also notice that medial orbital cortex is not mentioned in the ms(?), although it in the atlas used is wedged between IL and DP within the AP coordinates stated by the authors (1.2 to 2.0 from Bregma). Overall the anatomy - subregions and layering, is a weakness of the study, which is a pity and seems unnecessary given the high quality of other parts of the study.

Division of prefrontal layers - we find it unclear how the layers were defined/identified, and why (and how) the authors separate layers 2 and 3? Furthermore, why are layers 5 and 6 pooled (e.g. Figure 3D); how can this be justified? The laminations applied goes against some conventions in the PFC field (and atlases), and particularly, it is known that L5 and L6 hold differential input and output patterns (e.g. Harris and Shepherd, 2015). In relation to this: in Fig 3D it is shown that for the mPFC-BLA-projecting subnetwork (red), L5/6 receives higher weighted input than L2. However, in Fig S5D it appears that L3 (not reported in Fig 3D) receives similar input as L5/6(?) We find that the partly unconventional handling of the layers together with the lack of L3 data in parts of analysis (Fig 3D) cause some confusion. Is it possible to make Fig 3D and S5D understandable together? The laminar aspect is important as a central claim is that 'the weighted output from mPFC-BLA cells onto other mPFC-BLA cells was stronger in deeper layers than in superficial ones'.

We believe that complete division of the data into layers allows for most information to be extracted from the data, and that pooling can sometimes obscure interesting effects. For this reason, we divided the mPFC into layers in places where we could do so based on the data. For layers 2 and 3, we saw a pattern of cell density that resembles the known trajectory of cell density in the PFC (as in Van De Werd... Uylings, Brain StructFunct 2010). This subdivision was consistent with DeNardo... Luo, Nat Neurosci 2015 (Supplementary Figure 6 therein). As for layers 5 and 6, we agree with the reviewer that subdivision into these two layers would have been informative. Unfortunately, we had no molecular markers that would allow us to distinguish between these layers and therefore pooled them together. Due to this incomplete division into layers, we included in the manuscript analyses that are blind to layer but only consider mediolateral position, such as in Figure 3E and Figure 4G-I.

Regarding the handling of L3 in Figure 3D and Figure S5D: the relationship between these two figures may be more accessible and understandable if we present L3 data in Figure 3D (where it is now lacking since we have no cells of the random class in L3). We are attaching here the plot of Figure 3D after

inclusion of L3 (Rebuttal Figure 2), and we have replaced Figure 3D with this new version to facilitate clarity.

This reply is a bit confusing - the authors argue for division of data into layers and then take the approach to pool two highly specific layers (L5 and L6). The response to why L5 and 6 were not separated is quite unscientific - layer specific markers are just a googling away, and eg scRNA and ST datasets are freely available and can be used for immune or in situ. For example, FoxP2 might be used as a marker of L6 within mPFC, as shown in the preprint Babiczky et al., 2021 (this preprint might be of general interest to the authors since it investigates molecular markers within mPFC and also includes DP. There it is shown, for example, that L6, as marked by FoxP2, is thinner in DP in comparison to IL, while Calb1 labeling L2/3 across PL and IL, in DP is present across almost the entire region, making a point on DP dissimilarity to the conventional mPFC). We did not point this in the first review as we were quite sure the authors would address the points on DP and layering.

Digital atlases also allow basic mapping, particular as the neurons' coordinates were mapped in detail in the current ms. Deep layers are compared to superficial layers in the ms, and while Fig S6 clearly indicate investigation of neurons in L5 (ie neurons directly below L2/3), is the study investigating deep layer neurons, or only L5 neurons? As a bare minimum, it should be shown that the dataset include data from both L5 and L6.

Furthermore, in Fig S5D it appears that non-mPFC-BLA projecting and random mPFC cells in layer 5/6 (3rd column, blue and black) do not receive any input from layer 2/3, which would be surprising considering assumed canonical cortical circuit models. This finding also needs to be incorporated into any conclusions.

This result is now explicitly incorporated in the Results section of the manuscript.

We appreciate this.

We find it unclear if the regression in Fig 4H was performed on data from all 'pairs' of cells without regard to their class (mPFC-BLA - mPFC-BLA; mPFC-BLA - Non-mPFC-BLA etc.). It would be of interest to understand if the regression coefficients differ between the classes (and if no differences are found, how can then the demonstrated differences in connectivity between different mPFC subnetworks be understood?)

The regression was performed when two of the predictors were the connection type (one was mPFC-BLA and the other was non-mPFC-BLA, such that 0 for both indicated random mPFC). This gives explicit weight to the connection type and can account for interactions between it and the other predictors, and therefore is more informative than running the regression separately for each connection type. Indeed, we found that mPFC-BLA to mPFC-BLA connection type had a significant positive coefficient indicating its weight in accounting for variability in connection probability.

Thank you for clarifying this question, we appreciate it.

The majority of patched cells are located in IL (75% (69/92)). It would be of interest to know, if the authors hold the data, whether the connectivity between prefrontal subregions differs (e.g. PL vs. IL) or if the findings in IL can be generalized to other (also not prefrontal?) regions.

We have repeated the analyses in Figure 3 using only the maps obtained from the PL cortex. Due to the small number of cells, conclusions were hard to draw for most analyses. Below is Table 2, where only the dorsal mPFC (PL and Cg) maps are mentioned (Rebuttal Table 1).

Despite the small cell numbers, we could find an indication that the laminar connectivity pattern observed in IL-BLA cells (Figure 3D) could hold for PL-BLA cells as well (Rebuttal Figure 3A). We also found that the adaptation index in PL-BLA cells tends to be higher than in random PL cells (Rebuttal Figure 3B), consistent with our IL data (Figure 4E). We could not, however, find directional input patterns in the PL dataset (Rebuttal Figure 3C).

We note, however, that the regression analysis in Figure 4 was performed on the entire mPFC connectivity dataset, including the PL and Cg. We found that the dorsoventral anatomical position of the presynaptic cell accounted for some of the connectivity, whereby ventral position was associated with a reduction in connection probability. This could indicate a difference in overall connection rates between the dorsal and ventral mPFC subregions.

The last results are highly interesting and tap into the notion that functional properties are more relevant than cytoarchitectural delineations for understanding of how the PFC is built and functions. In line with this, biological and functional parameters should dictate what part of the brain could be considered part of the PFC, eg for the DP.

Minor comments:

The colors used in Fig S1B (IL, low/high titer) are difficult to distinguish, particularly for the circles.

Agreed. We changed the contrast and the stroke.

Figure 2G shows a 'representative synaptic connectivity map' with a patch recorded cell in DP. As the majority of patched cells (69/92) were recorded in IL and only 6/92 in DP, why was that example chosen? See also comment about DP above.

We replaced the DP map with an IL map in Figure 2G.

The sentence 'Moreover, the intrinsic properties of the postsynaptic mPFC cell and anatomical position of both cells jointly account for...' in the abstract is difficult to follow (if one has not already read the article) - postsynaptic to what, and what two cells are meant ('both cells')? Also at other places it is at times hard to know what cell(s) is meant - perhaps consider to avoid only saying cell, and instead

consistently state post- /presynaptic, mPFC-BLA cell etc...

Thank you for pointing this out. We rephrased the Abstract to make it clearer.

Fig 3E, F: Can this data be translated into layers/mPFC subregions? It would allow interpretation of possible differences in weighted input between subregions.

We show the same data for the PL in our response to the reviewer's final major comment above. However, due to the small number of maps in the dorsal mPFC region, we did not include this separate analysis in the manuscript.

The abstract ends with 'Our findings demonstrate a functional segregation of mPFC excitatory neuron subnetworks, and reveal the factors determining connectivity in the mPFC'. This appears a bit of an overstatement - the study is purely *ex vivo* and functional synapses are examined rather than the function of segregated subnetworks. Also, only a single specific subnetwork is investigated.

We rephrased the Abstract.

Fig S5D: while a summary figure is great for an overview, the array of scales on the y- axes makes it very hard to actually compare the connections - could perhaps fewer scales be used?

We understand the reviewer's point, and have debated about this point. There is a tradeoff here between matching scales to allow comparisons, and the ability to visualize the data in each plot more clearly. We now generated the same figure with a fixed y- scale across all plots (Rebuttal Figure 4) and placed it instead of the previous one.

We understand the authors' debating, both plot versions indeed have their pros and cons. Perhaps let the purpose of the plot guide the layout - if comparison is the major goal, the new version is actually informative.

Reviewer #3 (Remarks to the Author):

The authors have made a great effort to provide very thorough rebuttals to my previous comments, which is greatly appreciated. Overall the manuscript is improved but there are still one or two small concerns which I feel need to be addressed.

Depth analysis Rebuttal Figure 6 – This isn't really what I was getting at, unless I misunderstand their plot, the idea here is that the spatial properties of the connections (i.e the presynaptic location of the inputs) will change as a function of depth. So as one transitions from superficial or deep L5 or down into L6 then the inputs shift from being descending to local. The parameters you provide just show that cells

remain connected to the local network, which is unsurprising, but does not provide any spatial information. That being said, the fact that very few cells seem to be recorded in L6 (>600 μ m) seems to suggest this is unlikely to be a major concern though and the sparsity of cells in each population probably would make any detailed analysis of this point challenging. However, it would be important to add this point to the discussion and perhaps reference the work of Anderson et al., 2010 Nat Neurosci. as a possible explanation for the lack of L2/3 to 5 input, which as the authors note is in disagreement with almost every region of cortex where this connection has been assayed.

Rebuttal figures 7 and 8. Could these please be included in the manuscript. I think they are very interesting and useful data for the field. Far more useful than the data in Figure S5D which as a rather arbitrary and non-standard metric (see also below).

ML Analysis in Figure 3. Given there are issues with this analysis based on both the sampling across layers and the overall interpretation without knowing the location of the recorded cell. May I suggest that they remove this analysis, or push it to supplemental, and replace it with the data from rebuttal figure 7 & 8 which are far more easy to interpret and extend the data in Figure 3D in a useful and meaningful way, which the current ML analysis does not.

Reviewer #4 (Remarks to the Author):

The authors have addressed all my concerns.

(A minor point: Naka et al does use 2P opto mapping in Fig 3. But they do not target individual neurons, so it is a subtly but crucially different experiment.)

Printz et al. - Response to reviewer comments on the revised manuscript (26/12/2022):

We wish to thank the reviewers once again for the thorough reading of our work, and for their helpful and constructive criticism. We have thoroughly addressed the remaining comments on our revised manuscript, which we believe substantially improved its readability and clarity. Please see below our responses to the remaining issues raised by the reviewers. The text is color coded as follows: our original responses to reviewer questions are in **blue**; the reviewers' responses to our previous answers are in **bold black**, and our current answers to those comments (where required) are in **red**.

Reviewer #1 (Remarks to the Author):

I have few suggestions to improve the clarity of the manuscript:

Page 5: the conditions chosen for the experiment: spiral duration 7.2ms, stimulation frequency 10Hz, light power 10 mW, are not shown in the Figure 1I or Figure 2SC, if the authors have these data, it would be better to include them in the figures.

The reviewer is correct in that the combination of parameters used in the connectivity mapping experiments – namely duration of 7.2 ms, frequency of 10 Hz, and power of 10 mW – does not appear in the dataset described in Figure 1I and Figure S2C (now changed to Figure S3C). We do not have similar calibration data for this exact parameter combination, since we performed calibration for each parameter separately and chose the optimal ones after their individual calibration.

OK

Page 6: authors characterize the axial resolution looking at the spiking probability and DF/F signal; It would be important here to also give the optical axial resolution, how extended is the spot used for the spiral scanning along the axial direction? Also could they give an explanation why the spatial specificity curve for GCaMP6s is so much narrower than the spike probability? Is this because their imaging conditions are not sensitive enough to detect a single spike?

To address this request, we have included the optical resolution of the scanning as PSF measurement in Figure S4. The reason that the GCaMP6s spatial specificity curve is narrower than that of stCoChR-induced spiking is the very high light sensitivity of stCoChR as compared with GCaMP6s, resulting in higher light power needed for a resolvable GCaMP6s signal.

Thanks for this explanation which seems to confirm our understanding: the power used for the GCaMP6s spatial specificity curve is too low to permit to reliably resolve a single spike. In this case we should suggest specifying the working power also in the caption and add a comment on the reason for this difference of the axial resolution also in the text (e.g. at the end of the first paragraph at pag. 5).

We added an explanation to the main text and the power used to the relevant figure caption.

Figure 1: please complement the figure with some representative images (or a stack) of the recorded slices to show expression, co-expression, sparsity of the labelling etc. In the current manuscript there are no images of the actual slices that were used for the experiments.

Thank you for the suggestion, which we completely agree with. We added a representative z-projection image to Figure S2C. It shows the density of labeled mPFCBLA cells in an imaged volume used for mapping connectivity. Regarding co-expression of stCoChR and GCaMP6s, since both are labeled with a red fluorophore and the basal GCaMP6s fluorescence (without activity) is too low, we performed anti-GFP staining to enhance the GCaMP6s signal. Images and quantification of stCoChR and GCaMP6s appear in Figure S2B. Notably, cells without co-expression would be discarded in analysis due to lack of GCaMP6s signal, either because of lack of stCoChR (and therefore spiking) or lack of GCaMP6s.

OK

From Figure 1E and in Fig. S2C, it seems that stimulation often gives rise to multiple spikes. Could the author comment whether this has an effect on the amplitude of the post-synaptic responses and how this is taken into account in the measurement of the synaptic strength?

A brief explanation and a reference to the relevant detailed Methods section were added to the main text. It now states on page 7: "Finally, we calculated the strength of synaptic connection at each stimulation as the weighted average of EPSCs within a time window following stimulation (see Methods), thus accommodating for jitter in synaptic latency and the possibility for multiple evoked EPSCs (Figure S3C)." The weighted average used to

calculate synaptic strength was based on a normal fit of the EPSC distribution, as described in Methods under "Measurement of synaptic connection strength".

OK

Figure 1F, G show the DF/F and GCaMP fluorescence curve as a function of spiral number, respectively. This characterization is important as the amplitude and slope of the GCaMPs signal are used to select the post-synaptic cells for the connectivity experiments. Few points should be then explained more clearly. Specifically, Fig1F show largely varying DF/F values, from one cell to another, in one cell the signal is comparable to the recordings made in presence of TTX where one would expect no spike activity. Same in Fig.1G, multiple cells in presence of TTX show responses comparable, if not higher, to the aCSF case. The authors should discuss these curves and the diversity in the responses more extensively. For example, are all the curves taken at the same power? Or is the GCaMP imaging power chosen on the base of the opsin expression levels (higher power for low opsins expressing cells) so that the GCaMP fluorescence signal varied accordingly? Are their imaging conditions sufficient to detect single APs?

The variation in GCaMP6s responses between cells most likely stems from variability in expression levels of both GCaMP6s and stCoChR, resulting in varying spiking efficiency as well as varying GCaMP6s signals for the same spiking behavior. Another factor that could contribute to this variation is the shape of the cell relative to the spiral pattern scanned on it. The light power was consistent across cells and was not adjusted to expression level (but was adjusted to depth in the tissue as explained in the Methods in order to maintain constant power between cells). stCoChR evokes very strong depolarizations (as characterized in Forli et al., eLife 2021), such that even in the presence of TTX, photostimulation can result in calcium influx through voltage-gated calcium channels, which in turn can increase the GCaMP6s signal without the cells spiking. We believe that this is the reason that some of the cells show strong GCaMP6s signal increase in presence of TTX. We therefore did not use the GCaMP6s signal in presence of TTX to determine a threshold for spiking, as the strong depolarization confounds the interpretation of this signal. Finally, our imaging conditions are not sufficient to detect single spikes but rather tell us whether spikes occurred during a stimulation train.

We appreciate the explanation of the authors and agree. The authors should include these considerations into the manuscript. Especially the possibility of calcium influx without action potential firing might be something not every reader is aware of.

The main text refers to the Methods which contain a detailed explanation for this matter.

The authors mention that, even in presence of TTX (pag 30), GCaMP fluorescence can still accumulate "supposedly due to the strong light-induced depolarization". Can the authors better clarify this statement and the nature of the effect? Also can they specify if cells that give these level of GCaMP transient also in presence of TTX, would pass the criterion to validate activation?

Please see the answer to the previous comment. Additionally, a depolarization amplitude that increases GCaMP6 signals in the presence of TTX would undoubtedly trigger spiking in a natural setting with no TTX. Therefore, it is highly likely that cells showing large increase in GcaMP6s signal with TTX would spike without TTX and pass the inclusion criteria.

Ok

Page 6: For the automated detection of potential pre-synaptic cells the authors describe to use mScarlet and dTomato fluorescence, co-expressed with stCoChR and GcaMP respectively. I expect that the cells used for the connectivity mapping would all (most) express both the CoChR and GcaMP, however in the description of the automated detection it is not clear if this was based on the detection of co-expressing cells or if some of the detected cells only express the CoChR and no GcamP or vice versa? Meaning the 209/316 cells detected in the volume (Page 6) correspond to cells expressing one of the two or co-expressing both? What did the authors mean with observing the detections?

In our experiments, we cannot distinguish between stCoChR expression and GCaMP6s expression due to the significant overlap between the spectra of dTomato and mScarlet. We rely on the fact that targeted cells which express only one of the two will not show GCaMP6s increase, and these cells will therefore be discarded during analysis.

Again, we appreciate the explanation of the authors, but it is really important that these aspects are incorporated in the text; as of now it is difficult to understand this from the manuscript.

An explicit explanation is now incorporated in the Results: "Detection was based on mScarlet and nuclear dTomato co-expressed with stCoChR and GCaMP6s, respectively (since these two markers could not be distinguished, co-expression of both stCoChR and GCaMP6s was validated later in analysis based on GCaMP6s

fluorescence in response to stimulation; see below and Methods).”

This also relates to my next point:

Does the cross-talk with GCaMP and mScarlet affect the cell mapping? Precisely, coexpressing cells should show GCaMP and mScarlet fluorescence in the cytosol, and dTomato fluorescence in the nucleus. Detecting cytosolic mScarlet informs about the presence of CoChR, but in case of cross talk, how do they distinguish that from GCaMP, which is also in the cytosol? This is important if they screen for co-expression directly, which is not clear (see previous comment). Maybe they just map the volume for any expression, stimulate all cells and discard the ones that are not responding?

This is precisely what we did – scan all cells that express a red fluorophore (without distinction between nuclear and cytosolic expression) and include only responsive ones.

Ok

Fig. 2D: The authors show the expected shape of the EPSC according to the model used for the automatic identification of connections. In the reported example, the onset of the response is perfectly aligned with the start of the stimulation, while in the experimental curve (Fig2C) the response is temporally delayed, could the author clarify this point and define what criterion is used to distinguish EPSC from direct photoactivation. Could the authors comment on that?

In all our data, we observed that direct photocurrents appear at ≤ 1.5 ms latency, minimal jitter, and consistent waveform with slow rise constant compared with EPSCs. This can be seen in Figure 2C. The seemingly zero-latency EPSC response in Figure 2D is due to the filtering effect of the KDE. Throughout our experimental dataset, EPSCs always appeared with at least few ms latencies.

The contribution of direct stimulation of the post-synaptic cell is not trivial (the rise time can depend on power/distance, expression level, illumination method etc.) and can be an important factor in connectivity mapping. We appreciate the explanation of the authors, which is very reasonable, but not easy to find in the manuscript. May the authors could add a sentence in the main text or at least a reference to their (very detailed) section in materials and methods, where they explain how they deal with the direct stimulation?

We now refer to the specific Methods section that explains the direct photocurrents in the caption of the relevant figure – Figure 2C: “Note the compound photocurrent+EPSC response in the third cell (see Methods, under *Subtracting evoked photocurrents*).” We believe that referring to this issue in the main Results section would further complicate the already methods-intensive manuscript, and might confuse the reader.

Discussion: In the discussion, the authors mention previous work achieving optogenetic based synaptic mapping saying that they are “restricted to measuring only unidirectional (in-degree) connectivity”. Could they better clarify this idea?

A clarification was added to the Discussion text. The intension is to say that only connections from the stimulated cells onto the recorded cell can be probed, and not in the reverse direction. Notably, after mapping connections from stimulated cells onto the recorded cell, another experiment can be conducted where one of the stimulated cells is now recorded and the previously recorded cell is now stimulated, but the recorded cell usually does not stay viable after a mapping experiment to consistently allow such an approach.

Discussion: in the discussion it would help to discuss (if any) the limitations of the presented technology and give an outlook on what could be improved (e.g. light delivering approach, detection, opsin expression and targeting....)

The main limitation of our approach lies in the jitter of spiking and synaptic responses which stems from the spiral stimulation method. This is inherent to spiral stimulation, but our analysis is still able to take the jitter into account using the modeling approach we describe in Fig. 2. Nevertheless, future work could utilize more advanced holographic methods to improve spike time precision and reduce jitter. A relevant text was added to the Discussion.

Ok

Pag 28/29: the authors give a detailed description on how they discriminated among direct photocurrents and EPSC and confirm that the criteria that they used work by comparing the currents recorded in presence of glutamate-receptor blockers. It would help if they could here also quantify how many cells they typically excluded for experiments. Also, supposing that the presence of artifactual photocurrent decreases with distance, they should comment on how this could affect the overall estimation of connectivity ratio and spatial distribution of connections.

The exact numbers of cells that were excluded in each experiment can be found in Table 2: it details, for each experiment, the number of stimulated cells, of which the number of cells inferred to have spiked in response to stimulation, and the number of cells included in analysis, such that the difference between the number of spiking cells and the number of included cells reflects the number of cells that were excluded due to large evoked photocurrents (or noisy recording, in some cases).

To address the second point concerning the dependence of evoked/artefactual photocurrents on distance from the recorded/postsynaptic cell, we have explicitly calculated this dependence (Rebuttal Figure 1). We found that as the reviewer suggested, the density of cells that evoke large photocurrents (PC) in the recorded cell, and are therefore excluded from analysis, decreases with distance from the recorded cell, whereas the distance of cells that do not evoke large photocurrents in the recorded cell (no PC) distributes normally. This could introduce a bias in estimation of connection probabilities, since relatively more close cells are excluded than farther cells. However, the absolute numbers of excluded cells, as seen in the bottom histogram, is small. Moreover, Figure S5B shows that at short distances (and also long distances), the mPFC-BLA to mPFC-BLA connections – where we can find PC-evoking cells – are not sparser than the other connection types – where no PC-evoking cells are found. Therefore, although a bias is theoretically predicted, we did not find it in the relevant dataset.

Rebuttal Figure 1. Quantification of cells whose stimulation evokes direct photocurrent in the recorded cell by distance from the recorded cell. Top, fraction of cells; bottom, number of cells.

Pag 28: “were manually examined for synaptic connections by searching the stimulation aligned traces for reliable, low-jitter EPSC occurrences.” Authors should define here the meaning of low-jitter EPSC, as the above discussion they made to distinguish photoactivation from EPSC based on the fast responses of photoactivation could be otherwise confusing.

The use of “low-jitter” here comes to differentiate such cells from randomly distributed EPSCs, which can sometimes appear to occur near the time of stimulation, but are more noisy and have higher “jitter”. As can be seen in the third trace in Figure 2C, EPSCs can be distinguished from direct photocurrents by their rapid rise time as compared with photocurrent rise time, by their longer latency, and by the variance in their waveform (especially amplitude) between repetitions. We used these characteristics to identify cells whose stimulation evokes photocurrent and cells whose stimulation evokes EPSCs ‘riding’ on top of a photocurrent.

Ok, the explanation is satisfactory. But please keep in mind that the slow rise of directly evoked currents that is observed may arise from the spiral stimulation and exclusion criteria used here may not be generalizable.

Figure S6: the authors discuss the possibility of short-term plasticity induced by repeated spiral stimulation, which gives rise to a greater EPSC amplitude for the first 3 or 4 stimulations. According to Figure 1E, it seems that the first 2/3 spirals can also generate double spikes which would probably also generate higher EPSC. Could the authors comment on how these two effects can be distinguished?

Our method does not allow complete control over the number of presynaptic spikes per stimulation, as shown in Figure S3C. Therefore, as the reviewer observed, calculation of short-term plasticity is inaccurate, and we mentioned this limitation in the Discussion. However, since this same limitation applies to all connection types that we have measured, we assume that differences in this plasticity estimation would reflect true differences between the connection types, and therefore chose to include this measurement in a Supplementary Figure. [As another reviewer pointed out, since these experiments are done in aCSF which has relatively high calcium concentration, the short-term plasticity features of synaptic connections are quite uniform and tend to be biased to strong adaptation, occluding any potential variation among populations].

Ok

The authors report EPSC down to around 5 pA. In methods they mention a threshold of 2.5 SD. What does this value correspond to practically? Are connectivity rate values they observe potentially affected by this precision? I suggest the authors better discuss their detection sensitivity at the post-synaptic site.

The value of 2.5 SD was added to the text under the relevant Methods section. As with any measurement technique, our ability to detect small synaptic events is limited by the sensitivity of our system. The 2.5 SD value represents the sensitivity of our system as the minimal EPSC amplitude which can be detected, and it is therefore the minimal strength of synaptic connection which we can detect. Strength of synaptic connections in the cortex is skewed, with many weak connections and few strong ones for each cell (for example, Cossell... Mrsic-Flogel, Nature 2015). Since our method might miss the weakest connections, we focused in our measurements on the weighted input that each cell receives, which represents the overall effect that the network has on the activity and excitability of the post-synaptic cell.

Ok

Minors

Ok

It would be useful to supplement figS1D and E with some actual images from the stacks show examples of cells detected manually and automatically.

Agreed. These images were added as Figure S2C.

Fig1C: I presume, based on the color code of the first column, that second and third column have been recorded with a 15 um spiral: it will help to give this information.

Assuming that the intention is to Figure S2C (now changed to Figure S3C) where there was a color code for the spiral size at the first column, we have changed the legend of this Figure to resemble that of Figure 11 which presents a similar dataset and where the spiral size color code appears on top of all columns.

Page 5: Throughout the characterization of the stimulation, it is not very clear if the 7.2 ms spirals are always two 3.6 ms spirals (same dwell time)?

The reviewer is correct, the dwell time is constant and the change in duration was achieved by concatenating spirals. A 7.2 stimulation is two 3.6 ms spirals (whereas one spiral is one round of in-and-out scan, as shown in Figure 1C, right image). A clarification was added to the Results text in page 5 where the surveying of scan parameters is described.

Page 55, Fig S7 A: Please include a legend to explain the three groups/colours or include it in the legend of this figure

A legend was added to the Figure. Please note that the figure numbering was changed and this figure now appears as Figure S8.

FigS1 Panels A, B and C,D,E are describing different experiments. The cell detection was performed (and characterized) on the slices used for connectivity recording (and not on the Ai9 mice?). It could be confusing for the reader to have this in the same figure.

We split Figure S1 into two separate figures and added illustrations to facilitate distinction between the different experiments presented in these figures.

Discussion: "Here we presented a large-scale implementation of such an approach, combining it with calcium-based readout of activity using a single laser source." Better probably to repeat here the FOV and the achieved number of recorded pairs.

Thank you for the suggestion. These details are now explicitly mentioned in this section of the Discussion: "Our semi-automated approach for cell detection and for sequential stimulation and calcium recording allowed us to probe the input from 95.4 ± 5.1 cells (mean \pm s.e.m), whose spiking in response to stimulation was validated using the GCaMP6s signal, onto each recorded cell in three dimensions within a volume of $\sim 420 \times 420 \times 300 \mu\text{m}^3$ ".

Methods page 30, considering that the tdecay is used to define a scattering length it would probably better to call it this way and also to compare the found value (147.6 um) with what is given in the literature.

We have changed the name of the constant to 'attenuation length' (since we believe it reflects a combination of scattering and absorption) and added a comparison to the reported literature.

Reviewer #2 (Remarks to the Author):

The study by Printz et al. entitled "High-throughput mapping of functional synaptic connectivity in the prefrontal cortex" employs a novel and promising method combining whole-cell patch-clamp recordings with 2-photon microscopic imaging and simultaneous optogenetic stimulation of projection-defined neurons in acute prefrontal brain slices to map the functional connectivity of the local excitatory circuit.

The authors characterize several features defining the probability of synaptic connections between retrogradely labelled BLA-projecting prefrontal neurons, mPFC-BLA neurons, and non-mPFC-BLA neurons, and between random mPFC neurons, respectively. The authors show that the major determinants of local connectivity within mPFC are the distance between the cell bodies and the anatomical position of cells, especially along the mediolateral axis of the mPFC.

The methods of the study are well described and we positively note the great care and detail the authors deploy

in characterising e.g. the size and duration of scan parameters, and the clear highlighting and discussion of limitations of the methodology.

While we are positive about the study, we find a need for clarifications and further discussion of particular issues, and a few experimental parameters needing to be addressed.

The viral strategy: while the calibration of expression of retrograde AAV vectors is well described, we fail to understand what exact viral strategy was actually used in experimental animals. We also find the images of BLA targeting suboptimal, as they don't allow evaluation of the specificity of the viral labelling. In line with this, it appears that the quantifications of labelling at the injection site focuses on the BLA specifically but say nothing about the degree of unspecific labelling in other amygdalar nuclei, or elsewhere. This is a central point, as a big part of the study regards mPFC-BLA neurons specifically. We find that the specificity needs to be appropriately addressed and accounted for. Overall we would welcome more images, including of cell body labelling in the mPFC (e.g. lamination) as to understand what neurons are interrogated, and the spatial distribution of the subnetwork identified.

The specificity of mPFC-BLA cell labeling is indeed central to this study, and we value the reviewer's comments on this issue. To address this, we have included in Figure S2 confocal images of the BLA and the mPFC of wildtype mice injected with rAAV2-retro- Cre into the BLA and DIO-stCoChR+GCaMP6s into the mPFC. To enhance visibility of axonal projections labeled with stCoChR and GCaMP6s, we stained slices for GFP. We show that the densest axonal projections from labeled mPFC cells are inside the BLA. These are now shown as Figure S2A,B. We have further included a Supplementary Discussion section (subtitled "Specificity of mPFC-BLA cell labeling") detailing the known projections from the infralimbic cortex to the region surrounding the BLA, to conclude that the likelihood of unspecific labeling exists, but is low. Briefly, the majority of labeling outside of the BLA (as seen in Figure S1A) was in the striatal region dorsal to the BLA, presumably due to viral leak from the injection needle during its withdrawal. However, based on the extensive characterization of the mouse brain-wide BLA connectivity (Hintiryan... Dong, Nat Comm 2021) and on the Allen Institute connectivity database, projections from the infralimbic cortex into this striatal region are very sparse compared with its projections into the BLA. Moreover, retrograde and anterograde labeling from the BLA result in labeling patterns similar to those we show in Figure S1A, and to a laminar distribution similar to that shown in Figure S2B. These observations suggest that non-specific retrograde labeling in the mPFC is minor compared with BLA-projecting cell labeling. Finally, ~55% of our mPFC-BLA cell mapping experiments were performed after injection of 0.1 µl of rAAV2-retro-Cre into the BLA to minimize viral spread outside of the BLA, and the rest were performed after injection of 0.3–0.4 µl of rAAV2-retro-Cre into the BLA. These are rather conservative amounts and are similar with previous studies targeting the BLA with stereotactic AAV injections.

We appreciate the added information and images related to the specificity of the viral labelling. The rAAV + 2 AAVs viral strategy was performed in WT MICE (not mentioned in results now) and the calibration experiments in Ai9 mice - we recommend that the authors in the text make it clear that the Cre recombination gives very different fluorescent labelling in the 2 sets of experiments. Particularly, TdTomato fills the neurons, labelling axons from recombined neurons in the injection site, ie give rise to anterograde labelling although a rAVV was injected. At least that is how we interpret the authors' descriptions. It is currently quite difficult to understand the text in eg "Specificity of mPFC-BLA cell labeling" and reconcile the labelling results in the two sets of experiments. We also find it suboptimal that the calibration experiments are mentioned after the actual experiments that used the findings from the calibration experiments. These are minor things but sources of confusion and misinterpretations.

Thank you for the attention to these details and helping us improve the clarity of the manuscript. We changed the order of the description of the experiments to first mention the Ai9 calibration and then the use of wild-type mice, and also clarified in the Results which mouse line was used for each experiment. Regarding the anterograde labeling from AAVretro, Figure S1 caption discusses the labeling of cell bodies at the injection site. We are concerned that incorporating this discussion into the Results at the beginning would throw the reader off into technical detail and make an already method-intensive section harder to absorb.

mPFC anatomy: anatomy is central to both the study, and its findings, and great progress has been made in anatomical mapping of the mouse brain over the past decade. We find the anatomical delineations and nomenclature in the current study unclear. An, to us, unknown atlas has been used that does not appear to adhere to the most commonly used digital atlas (created by the Allen Institute for Brain Science). As the definition of the PFC varies among researchers, it is of importance that cell locations can be understood/re-mapped by colleagues in the field. We find the study would be stronger if it was better conveyed how the PFC subregions are delineated anatomically, i.e. if the reader could understand e.g. what part of the tissue the authors denote IL, DP, cingulate. This would help understanding and personal interpretation of the cell locations. There is only a quite small single image (Fig 1B; is this adhering to <https://kimlab.io/brain-map/atlas/>?). We also recommend finding space in the main text to at least briefly state what atlas was used, why, and how.

This is a good point. As the reviewer correctly concluded, the atlas we used is the one shown

in <https://kimlab.io/brain-map/atlas/>. This reference was added to the manuscript in the places where it was used (Chon... Kim Nat Comm 2019). We now mention specifically which atlas was used at each relevant place and figure in the manuscript. An explanation for the choices was added both to the main text and to the relevant Methods section that describes these choices ("Inferring the brain-reference anatomical positions of the probed cells"), and this section was further detailed for disambiguation. Generally, the aforementioned atlas was used for all anatomical analyses, as it integrates the two most commonly used atlases (Allen Institute and Franklin & Paxinos) and has an accessible interface for image analysis. For presentations in figures, we used the Franklin & Paxinos, 2008 atlas for its outline presentation that is suitable for overlay on microscopy images.

We understand the reply as that the position of cells and the delineation of the mPFC adhere to the <https://kimlab.io/brain-map/atlas/>. However, the figures does not necessarily reflect the data as a different atlas was used. It is our view that the figures should adhere to the analysis, i.e., the same atlas should be used, particularly as the authors say below, borders in atlases are somewhat arbitrary drawn, ie the atlases differ. In line with this, we notice that the DP appears to be expanded in the DV plane in the <https://kimlab.io/brain-map/atlas/> compared to the Allen v2. We cannot see any good reason why two different atlases should be used.

The reviewer is correct. For consistency, we have changed the reference atlas images in Figure 1B,J and Figure 2G and used the atlas in <https://kimlab.io/brain-map/atlas/>, such that this atlas is used all throughout the manuscript, both for analysis and for presentation.

Also related to anatomy:

DP - It is unclear why the DP is included in the study, and, thus, here included in the mPFC. This needs to be addressed/justified/discussed, including in relation to the BLA and topics addressed in the Discussion.

We appreciate the comment, but we also took into account the lack of sharp anatomical boundaries between regions in the mPFC. The borders drawn in atlases are somewhat arbitrary, and it is highly likely that similar functions are performed by BLA-projecting neurons in the ventral IL and dorsal DP. To address this comment, however, we repeated the analyses from Figure 3, Figure 4A–F, Figure S5A–C, Figure S6 (now appears as Figure S7), and Figure S7 (now appears as Figure S8) – but this time including only cells in the IL (and not in the DP). The main effects (and lack of effects) persisted, except for the significance level of the weighted input along the mediolateral axis for mPFC-BLA cells (Figure 3E, left), which reduced to $p = 0.06$. The conclusions from the data are unchanged after removing the DP. This may be due to the small number of DP connectivity maps in our dataset compared with the number of IL maps, as appears in Table 2 of the manuscript: 69 IL maps and only 6 DP maps. Apart from this, we justify the inclusion of the DP in our dataset and analyses based on the facts that the DP contains cells projecting to the BLA (Hintiryan... Dong, Nat Comm 2021; and our manuscript), and that some studies include it within the ventromedial PFC region without subdivisions and show that it is implicated in fear extinction which is typically attributed to the IL (for example, Bukalo... Holmes Sci Adv 2015). We believe that these findings suggest that both IL-BLA cells and DP-BLA cells are involved in similar learning- related cognitive processes.

This reply contains some handwaving. More important, this point has not been addressed in the revised manuscript, and we again suggest that the authors discuss the inclusion of DP in the analysis. We think the authors can agree that DP as a rule is not included in the mPFC and if the authors wish to do so, the scientific reasons should be clarified to the readers. Contrary to the regions traditionally included in the PFC, the DP does not have a L6b. Inclusion of DP will raise questions and without scientific reasoning its inclusion smells of p hacking given the result presented in the reply above. We also notice that medial orbital cortex is not mentioned in the ms(?), although it in the atlas used is wedged between IL and DP within the AP coordinates stated by the authors (1.2 to 2.0 from Bregma). Overall the anatomy - subregions and layering, is a weakness of the study, which is a pity and seems unnecessary given the high quality of other parts of the study.

We have added to the Results section a reasoning for our view of the inclusion of the DP in our dataset. It now reads: "... In these two experiments we observed that in the dorsal mPFC (prelimbic and cingulate cortices), mPFC-BLA cells were largely absent from layer 3, where as in the ventral infralimbic and dorsal peduncular cortices, mPFC-BLA cells distributed densely across all layers (Figure 1B, Figure S1A, and Figure S6)... Among the 75 maps in the ventral mPFC, six maps (8%) were in the dorsal peduncular cortex (DP). Since the DP contained cells projecting to the BLA with a laminar distribution similar to that of the infralimbic cortex (Figure 1B, Figure S1A, and Figure S6; see also ref. 56), and since the DP-BLA cells have been implicated in fear extinction, similar to infralimbic neurons³⁹ – we have included the DP maps in our analysis of ventral mPFC connectivity." Regarding the medial orbital cortex: Our dataset indeed includes an AP coordinate range of 1.2–2 mm from bregma. Within this range, the MO is wedged between the IL and DP only between 1.9 and 2 mm, and within this very narrow range, the maps we have sampled were too dorsal to reach the MO or the DP. Finally, we can assure the reviewer that we have included the DP maps in our analysis from the very beginning. We performed the IL-only analysis only after revision, so that the DP was not included for the purpose of statistical significance.

Division of prefrontal layers - we find it unclear how the layers were defined/identified, and why (and how) the authors separate layers 2 and 3? Furthermore, why are layers 5 and 6 pooled (e.g. Figure 3D); how can this be justified? The laminations applied goes against some conventions in the PFC field (and atlases), and particularly, it is known that L5 and L6 hold differential input and output patterns (e.g. Harris and Shepherd, 2015). In relation to this: in Fig 3D it is shown that for the mPFC-BLA-projecting subnetwork (red), L5/6 receives higher weighted input than L2. However, in Fig S5D it appears that L3 (not reported in Fig 3D) receives similar input as L5/6(?) We find that the partly unconventional handling of the layers together with the lack of L3 data in parts of analysis (Fig 3D) cause some confusion. Is it possible to make Fig 3D and S5D understandable together? The laminar aspect is important as a central claim is that 'the weighted output from mPFC-BLA cells onto other mPFC-BLA cells was stronger in deeper layers than in superficial ones'.

We believe that complete division of the data into layers allows for most information to be extracted from the data, and that pooling can sometimes obscure interesting effects. For this reason, we divided the mPFC into layers in places where we could do so based on the data. For layers 2 and 3, we saw a pattern of cell density that resembles the known trajectory of cell density in the PFC (as in Van De Werd... Uylings, Brain StructFunct 2010). This subdivision was consistent with DeNardo... Luo, Nat Neurosci 2015 (Supplementary Figure 6 therein). As for layers 5 and 6, we agree with the reviewer that subdivision into these two layers would have been informative. Unfortunately, we had no molecular markers that would allow us to distinguish between these layers and therefore pooled them together. Due to this incomplete division into layers, we included in the manuscript analyses that are blind to layer but only consider mediolateral position, such as in Figure 3E and Figure 4G-I. Regarding the handling of L3 in Figure 3D and Figure S5D: the relationship between these two figures may be more accessible and understandable if we present L3 data in Figure 3D (where it is now lacking since we have no cells of the random class in L3). We are attaching here the plot of Figure 3D after inclusion of L3 (Rebuttal Figure 2), and we have replaced Figure 3D with this new version to facilitate clarity.

This reply is a bit confusing - the authors argue for division of data into layers and then take the approach to pool two highly specific layers (L5 and L6). The response to why L5 and 6 were not separated is quite unscientific - layer specific markers are just a googling away, and eg scRNA and ST datasets are freely available and can be used for immune or in situ. For example, FoxP2 might be used as a marker of L6 within mPFC, as shown in the preprint Babiczky et al., 2021 (this preprint might be of general interest to the authors since it investigates molecular markers within mPFC and also includes DP. There it is shown, for example, that L6, as marked by FoxP2, is thinner in DP in comparison to IL, while Calb1 labeling L2/3 across PL and IL, in DP is present across almost the entire region, making a point on DP dissimilarity to the conventional mPFC). We did not point this in the first review as we were quite sure the authors would address the points on DP and layering. Digital atlases also allow basic mapping, particular as the neurons' coordinates were mapped in detail in the current ms. Deep layers are compared to superficial layers in the ms, and while Fig S6 clearly indicate investigation of neurons in L5 (ie neurons directly below L2/3), is the study investigating deep layer neurons, or only L5 neurons? As a bare minimum, it should be shown that the dataset include data from both L5 and L6.

For a more complete division into layers, we closely examined the division between layers 5 and 6 in the literature and found that a common cutoff for separating L5 from L6 is a depth of ~640 μm from the midline (for example, the same immunostaining performed in DeNardo... Luo, Nat Neurosci 2015 that we referred to above). With this border, we found that all of the recorded cells in our dataset, that were labeled as positioned in L5/6, are in fact in L5 (maximum depth of 581 μm). We have updated the text and figures throughout the manuscript to reflect the fact that only cells in L5 were sampled, and not in L6. We very much appreciate the discussion with the reviewer that has led to this improvement in the manuscript. It is noteworthy that a small fraction of the stimulated cells (stimulated while recording from L5 cells) are located in L6 (27 mPFC-BLA cells in L6 compared with 3780 in L5, and 98 randomly labeled cells in L6 compared with 1645 in L5), and so for the cross-laminar connectivity presented in Figure 3D,E and in Figure S5E, the terminology "L5/6" is still used for the source of synaptic input.

Furthermore, in Fig S5D it appears that non-mPFC-BLA projecting and random mPFC cells in layer 5/6 (3rd column, blue and black) do not receive any input from layer 2/3, which would be surprising considering assumed canonical cortical circuit models. This finding also needs to be incorporated into any conclusions.

This result is now explicitly incorporated in the Results section of the manuscript.

We appreciate this.

We find it unclear if the regression in Fig 4H was performed on data from all 'pairs' of cells without regard to their class (mPFC-BLA - mPFC-BLA; mPFC-BLA - Non-mPFC-BLA etc.). It would be of interest to understand if the regression coefficients differ between the classes (and if no differences are found, how can then the demonstrated differences in connectivity between different mPFC subnetworks be understood?)

The regression was performed when two of the predictors were the connection type (one was mPFC-BLA and the other was non-mPFC-BLA, such that 0 for both indicated random mPFC). This gives explicit weight to the connection type and can account for interactions between it and the other predictors, and therefore is more informative than running the regression separately for each connection type. Indeed, we found that mPFC-BLA to mPFC-BLA connection type had a significant positive coefficient indicating its weight in accounting for variability in connection probability.

Thank you for clarifying this question, we appreciate it.

The majority of patched cells are located in IL (75% (69/92)). It would be of interest to know, if the authors hold the data, whether the connectivity between prefrontal subregions differs (e.g. PL vs. IL) or if the findings in IL can be generalized to other (also not prefrontal?) regions.

We have repeated the analyses in Figure 3 using only the maps obtained from the PL cortex. Due to the small number of cells, conclusions were hard to draw for most analyses. Below is Table 2, where only the dorsal mPFC (PL and Cg) maps are mentioned (Rebuttal Table 1).

Despite the small cell numbers, we could find an indication that the laminar connectivity pattern observed in IL-BLA cells (Figure 3D) could hold for PL-BLA cells as well (Rebuttal Figure 3A). We also found that the adaptation index in PL-BLA cells tends to be higher than in random PL cells (Rebuttal Figure 3B), consistent with our IL data (Figure 4E). We could not, however, find directional input patterns in the PL dataset (Rebuttal Figure 3C).

We note, however, that the regression analysis in Figure 4 was performed on the entire mPFC connectivity dataset, including the PL and Cg. We found that the dorsoventral anatomical position of the presynaptic cell accounted for some of the connectivity, whereby ventral position was associated with a reduction in connection probability. This could indicate a difference in overall connection rates between the dorsal and ventral mPFC subregions.

The last results are highly interesting and tap into the notion that functional properties are more relevant than cytoarchitectural delineations for understanding of how the PFC is built and functions. In line with this, biological and functional parameters should dictate what part of the brain could be considered part of the PFC, eg for the DP.

Thank you for this comment. We completely agree with this point, and this is indeed one of the reasons that motivated us to retain the DP data in the manuscript.

Minor comments:

The colors used in Fig S1B (IL, low/high titer) are difficult to distinguish, particularly for the circles.

Agreed. We changed the contrast and the stroke.

Figure 2G shows a 'representative synaptic connectivity map' with a patch recorded cell in DP. As the majority of patched cells (69/92) were recorded in IL and only 6/92 in DP, why was that example chosen? See also comment about DP above.

We replaced the DP map with an IL map in Figure 2G.

The sentence 'Moreover, the intrinsic properties of the postsynaptic mPFC cell and anatomical position of both cells jointly account for...' in the abstract is difficult to follow (if one has not already read the article) - postsynaptic to what, and what two cells are meant ('both cells')? Also at other places it is at times hard to know what cell(s) is meant - perhaps consider to avoid only saying cell, and instead consistently state post- /presynaptic, mPFC-BLA cell etc...

Thank you for pointing this out. We rephrased the Abstract to make it clearer.

Fig 3E, F: Can this data be translated into layers/mPFC subregions? It would allow interpretation of possible differences in weighted input between subregions.

We show the same data for the PL in our response to the reviewer's final major comment above. However, due to the small number of maps in the dorsal mPFC region, we did not include this separate analysis in the manuscript.

The abstract ends with 'Our findings demonstrate a functional segregation of mPFC excitatory neuron subnetworks, and reveal the factors determining connectivity in the mPFC'. This appears a bit of an overstatement - the study is purely ex vivo and functional synapses are examined rather than the function of segregated subnetworks. Also, only a single specific subnetwork is investigated.

We rephrased the Abstract.

Fig S5D: while a summary figure is great for an overview, the array of scales on the y- axes makes it very hard to actually compare the connections - could perhaps fewer scales be used?

We understand the reviewer's point, and have debated about this point. There is a tradeoff here between matching scales to allow comparisons, and the ability to visualize the data in each plot more clearly. We now generated the same figure with a fixed y- scale across all plots (Rebuttal Figure 4) and placed it instead of the previous one.

We understand the authors' debating, both plot versions indeed have their pros and cons. Perhaps let the purpose of the plot guide the layout - if comparison is the major goal, the new version is actually informative.

Thank you for these remarks. We have decided to keep the new format where a uniform scale is used for all plots.

Reviewer #3 (Remarks to the Author):

The authors have made a great effort to provide very thorough rebuttals to my previous comments, which is greatly appreciated. Overall the manuscript is improved but there are still one or two small concerns which I feel need to be addressed.

Depth analysis Rebuttal Figure 6 – This isn't really what I was getting at, unless I misunderstand their plot, the idea here is that the spatial properties of the connections (i.e the presynaptic location of the inputs) will change as a function of depth. So as one transitions from superficial or deep L5 or down into L6 then the inputs shift from being descending to local. The parameters you provide just show that cells remain connected to the local network, which is unsurprising, but does not provide any spatial information. That being said, the fact that very few cells seem to be recorded in L6 (>600µm) seems to suggest this is unlikely to be a major concern though and the sparsity of cells in each population probably would make any detailed analysis of this point challenging. However, it would be important to add this point to the discussion and perhaps reference the work of Anderson et al., 2010 Nat Neurosci. as a possible explanation for the lack of L2/3 to 5 input, which as the authors note is in disagreement with almost every region of cortex where this connection has been assayed.

After close examination of the depths of recorded cells and the known laminar organization of the mPFC, we concluded that no cells in our dataset were recorded in L6. We changed our manuscript to reflect the fact that all recorded L5/6 cells are, in fact, L5 cells. Within L5, all of the recorded mPFC-BLA cells, except for two, were in superficial L5 (down to depth 457 µm, see Rebuttal Figure 6), so it is not possible for us to compare the type of input that superficial cells receive vs. deeper cells. As for random cells in L5, none of these cells received input from cells in L2/3 (in the case of the two random cells that were deeper than 500 µm, no stimulated presynaptic cells were located in L2/3 to begin with), but we agree with the reviewer that the small number of cells in this category precludes any meaningful conclusion from this analysis. For the same reasons, we did not refer to the Anderson et al. study.

Rebuttal figures 7 and 8. Could these please be included in the manuscript. I think they are very interesting and useful data for the field. Far more useful than the data in Figure S5D which as a rather arbitrary and non-standard metric (see also below).

We have included these plots as Figure 3D,E (see also our reply to the next comment).

ML Analysis in Figure 3. Given there are issues with this analysis based on both the sampling across layers and the overall interpretation without knowing the location of the recorded cell. May I suggest that they remove this analysis, or push it to supplemental, and replace it with the data from rebuttal figure 7 & 8 which are far more easy to interpret and extend the data in Figure 3D in a useful and meaningful way, which the current ML analysis does not.

Based on the two reviewer's comments above, we have rearranged Figure 3 and Figure S5, such that the cross-layers probability of connection as well as connection strength are now inside main Figure 3, whereas the rest of the data that was in Figure 3 has been moved to Figure S5.

Reviewer #4 (Remarks to the Author):

The authors have addressed all my concerns.

(A minor point: Naka et al does use 2P opto mapping in Fig 3. But they do not target individual neurons, so it is a subtly but crucially different experiment.)

Thank you for pointing this out.